# On the Topological Structure of Nonlocal Continuum Field Theories

**Said Mikki** [1,2]

1 International Campus, Zhejiang University/University of Illinois at Urbana-Champaign (ZJU-UIUC) Institute, Zhejiang University, 718 East Haizhou Road, Haining 314400, China; said.m.mikki@gmail.com
2 Electrical and Computer Engineering Department, University of Illinois at Urbana-Champaign, Engineering Hall MC 266, 1308 West Green Street, Champaign, IL 61820, USA

**Abstract:** An alternative to conventional spacetime is proposed and rigorously formulated for nonlocal continuum field theories through the deployment of a fiber bundle-based superspace extension method. We develop, in increasing complexity, the concept of nonlocality starting from general considerations, going through spatial dispersion, and ending up with a broad formulation that unveils the link between general topology and nonlocality in generic material media. It is shown that nonlocality naturally leads to a Banach (vector) bundle structure serving as an enlarged space (superspace) inside which physical processes, such as the electromagnetic ones, take place. The added structures, essentially fibered spaces, model the topological microdomains of physics-based nonlocality and provide a fine-grained geometrical picture of field–matter interactions in nonlocal metamaterials. We utilize standard techniques in the theory of smooth manifolds to construct the Banach bundle structure by paying careful attention to the relevant physics. The electromagnetic response tensor is then reformulated as a superspace bundle homomorphism and the various tools needed to proceed from the local topology of microdomains to global domains are developed. For concreteness and simplicity, our presentations of both the fundamental theory and the examples given to illustrate the mathematics all emphasize the case of electromagnetic field theory, but the superspace formalism developed here is quite general and can be easily extended to other types of nonlocal continuum field theories. An application to fundamental theory is given, which consists of utilizing the proposed superspace theory of nonlocal metamaterials in order to explain why nonlocal electromagnetic materials often require additional boundary conditions or extra input from microscopic theory relative to local electromagnetism, where in the latter case such extra input is not needed. Real-life case studies quantitatively illustrating the microdomain structure in nonlocal semiconductors are provided. Moreover, in a series of connected appendices, we outline a new broad view of the emerging field of nonlocal electromagnetism in material domains, which, together with the main superspace formalism introduced in the main text, may be considered a new unified general introduction to the physics and methods of nonlocal metamaterials.

**Keywords:** nonlocal metamaterials; multiscale structures; fiber bundles; superspace; mathematical methods; mathematical physics; nonlocal continuum field theory; semiconductor materials

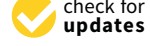



## 1. Introduction

Numerous research studies point toward a basic fact: topology and physics are destined to come closer to each other in the following decades [1–4]. This in itself is not totally new because several authors, for example, Henri Poincare, E. Cartan, and Hermann Weyl, had already advocated topological thinking in physics [5–7]. However, a salient feature of this convergence is the focus on material engineering applications, for example, metamaterials and topology-based devices. In this paper, we look into the general and rigorous foundations of the discipline behind these applications, namely the framework of *nonlocal continuum field theories* [8,9], with focus on explicating the generic multiscale topological

structure of continua studied by such theories. We propose that in addition to the now mainstream approach to topological materials [10,11], where the emphasis is often laid on exploiting the global dependence of the wave function on momentum (Fourier) space, there is a need to consider how materials can be assigned an indirect structure indexed by parameters taken directly from the *spatial* side of the configuration space, i.e., either space–time or space–frequency.

Our key observation is that arriving at an adequate understanding and characterization of nonlocality in generic scenarios would naturally require gathering information at the microtopological level of what we dub *nonlocal microdomains* (the topological level of small regions around every point where the response is nonlocal), then collectively aggregating these microdomains in order to obtain the global topological structure (the macro-topological level). The fundamental insight coming from topology is precisely how this process of "moving from the local to the global" can be enacted. We have found that a very efficient method to do this is the natural formulation of the entire problem in terms of a fiber bundle superspace, where conventional spacetime or space–frequency are here understood as nothing but "index spaces" embedded into a larger (in our opinion more fundamental) fibered superspace characteristic of nonlocal continuum field theories. In other words, and in contrast to existing approaches to local field theories and topological materials, our strategy is not to first solve Maxwell's equations in order to find the state function as expressed within the Fourier **k**-space, after which one proceeds to study topology over momentum space; instead, we start in spacetime (or space–frequency), and then formulate the *extended* or superspace structure of a topology over a fiber bundle where the conventional position space of the nonlocal continuum, e.g., Euclidean space, would manifest itself merely as the index space of the fiber bundle superspace.

The principal conceptual and philosophical message behind this work is that spacetime (or space–frequency) is not adequate for formulating nonlocal continuum field theories, and that a more appropriate natural approach is the superspace formalism proposed below, which, in our case, is based on a specific fiber bundle construction taking into account the intricate physics-based microdomain structure of the generic nonlocal continuum. It is the hope of the author that by helping scientists generate new insights into their physics and models, this formalism may provide a rigorous approach complementing some of the exciting theories and researches currently addressing various topics in continuum field theories, nonlocal metamaterials, and topological materials, while possibly stimulating the creation of novel algorithms for the computation of suitable topological invariant characterizing complex material domains. Due to the wide scope and complexity of this work, we first provide in Section 2 a relatively lengthy overview on the our contribution, where high-level information about this work, in addition to a guide to the literature and how to read the present paper, are outlined before moving to the more technical treatments of the subsequent sections and appendices.

## 2. Preliminary Considerations

While the essential idea of the superspace formalism introduced here will be valid for a generic nonlocal continuum field theory, it is much easier sometimes to work with a concrete example, especially in explaining what nonlocality is for someone who is coming to the subject for the first time. Therefore, in this preliminary section, we emphasize the special but very important case of *electromagnetic* nonlocality.

### 2.1. What Is Nonlocality?

In classical electromagnetic (EM) theory, it is currently widely held that there are no nonlocal interactions or phenomena in vacuum because Maxwell's equations, which capture the ultimate content of the physics of electromagnetic fields, are essentially local differential equations [12]. In other words, an effect applied at point **r** in space will first be felt at the same location but then spread or propagate slowly into the *infinitesimally* immediate neighborhood. Long-term disturbances, such as electromagnetic waves, propagate through

both vacuum and material media by cascading these infinitesimal perturbations in outward directions (rays or propagation paths) emanating from the time-varying point source that originated the whole process. However, if we leave behind vacuum electromagnetism and move into electromagnetically-responsive matter-filled space, then we note that *nonlocal* interactions in material domains differ fundamentally from the de facto local vacuum-like picture in allowing fields applied at position $\mathbf{r}'$ to influence the medium at *different* location $\mathbf{r}$ [13]. That is, in the nonlocal material system, a location not *infinitesimally* close to the source position $\mathbf{r}'$ can experience a nonvanishing effect emanating from the source location. While the "nonlocality scale" $|\mathbf{r} - \mathbf{r}'|$ tends to be quite small in most natural media (and certainly zero in vacuum), in some types of materials, the so-called *nonlocal media*, observable response can be found such that this "radius of nonlocality" $|\mathbf{r} - \mathbf{r}'|$ becomes appreciably different from zero [14–16].

The existence of multiple scales in the fundamental physics of nature is not really new. The scaling properties are important in Yang–Mills fields, the non-abelian field theory, and it has been recently used to propose the presence of fractal structures in the dynamical evolution of the fields. For example, one may consider the fractal structure of Yang–Mills fields [17] as an example of a multiple-scale effect in fundamental field theories.[1] In a more familiar setting, it is generally accepted that Aharonov–Bohm-type effects, which lead to observable nonlocal electrodynamic effects [18], have their origin in quantum physics. Bringing quantum physics into field theory can be shown to lead to intrinsically nonlocal effects since quantum field theory may be considered a fundamentally nonlocal theory due to, for example, entanglement effects [19,20]. However, in this paper, we focus on *classical* field theory realized through *phenomenological* models of the electromagnetic response of the material domain. The phenomenological model itself (the constitutive relations [9]) may have as its ultimate origin a purely quantum effect. For example, the main example considered in this paper, the nonlocal semiconductor material domain, has as its "origin of nonlocality" the essentially *quantum* process of exciton polariton coupling in solids (Section 7). It should be noted that in recent years some authors suggested that *classical* electromagnetism, under certain conditions, *may* induce nonlocal effects [21–23]; nevertheless, such scenarios are *outside* the scope of the physical paradigm treated in the present paper.

On the other hand, and interestingly enough for our purposes, Cvijanovich proposed several decades ago a theoretical model in which *vacuum* itself is modeled as a nonlocal constitutive *non*-material domain, where the standard Lorentzian spacetime manifold of general relativity is assumed here to play the role of the "medium" transmitting nonlocal actions [24]. Such proposal might be linked to field–matter interaction regimes where there is a strong coupling between gravitational fields and electromagnetic degrees of freedom. For flat spacetime, however, we already know from experiments that classical electromagnetism is strictly local. Nevertheless, it was discovered recently that classical electromagnetism can be made nonlocal if the photon mass is nonzero. More precisely, classical massive electromagnetism can be shown to arise in certain nonlocal (spatially dispersive) homogeneous domains [25]. Therefore, the statement that "classical electromagnetism is strictly local" should be qualified by allowing for the possibility that the photon mass might be proved experimentally to be non-vanishing, say in a future empirical research. In spite of all these interesting proposals on how to modify classical electromagnetic theory in order to make it compatible with nonlocality at the very fundamental level, the system of field theory treated in this paper is mainly classical, and the underlying spacetime structure is flat (the gravitational degrees of freedom are ignored).

The research field concerned with the study of the classical electromagnetism of nonlocal material domains is called *nonlocal electromagnetism/electromagnetics/electrodynamics.* This paper introduces a comprehensive general approach to this emerging discipline together with a series of selected applications. An extensive literature survey on past researches into nonlocal electromagnetism is given in Appendixes A.1 and A.2. The subject of nonlocal electromagnetism, here understood as the electromagnetism of nonlocal material

domains, is presently treated as a subdomain of the science of metamaterials. Historically, it has not been a well-defined direction of research, with researchers working on nonlocal structures often coming from very diverse and distinct fields, such as plasma physics, crystal optics, periodic structures, metasurfaces, and so on. One of the objectives of this article is to propose a coherent view of the inherently cross-disciplinary nonlocal materials research program, encompassing contributions coming from theoretical physics, applied physics, chemistry, engineering, with mathematical physics as the unifying framework of our inquiry.

### 2.2. Key Contributions and Motivations in the Present Work

Currently, there is an interest within applied physics and engineering in harnessing nonlocal media as a new generation of metamaterials for use in various settings, e.g., optical devices, energy control, antennas, circuit systems, etc., see Appendixes A.1 and A.3. The main goal of the present work is to explore, at a very general level, the conceptual and mathematical foundations of nonlocality in connection with applied electromagnetic metamaterials (MTMs). Our approach is conceptual and theoretical, with the main emphasis being laid on understanding the mathematical foundations of the subject and how they relate to the underlying physical bases of some illustrative examples. Indeed, while a massive amount of numerical and experimental data on all types of nonlocal materials abound in a literature that goes back to as early as the 1950s, the purpose of the present paper is attaining some clear understanding of the essentials of the subject, particularly in connection with the ability to build a very general superspace formalism for nonlocal continuum field theory without restricting the formalism first to particular classes of materials such as metals, plasma, or semiconductors.

The central theoretical idea in this work is the introduction of the superspace concept into the process of constructing a general formalism suitable for understanding, analyzing, and designing nonlocal material systems in classical field theory. The superspace formalism has a long history in physics, mathematical physics, and mathematics (see Appendix A.4). It will be shown below that nonlocal continuum field theory appears to lead very naturally to a reformulation of its essential configuration space by upgrading the conventional space–time or frequency space to a larger superspace in which the former spaces serve as base spaces for the new (larger) superspace. Such reconsideration of the fundamental structure of the problem may help foster future numerical methods and potential applications as will be discussed later, e.g., see Appendix A.11.2.

The key motivation behind the proposed superspace approach is explicating a subtle, but often overlooked, difference between two fundamental scales of interactions in nature:

1. *Infinitesimal interactions*: this characterizes local field theories, e.g., local electromagnetism, where all operators are differential operators.
2. Non-infinitesimal but local interactions: here, nonlocal operators, such as integral operators, may be present. In this type of theory, interactions are extended into small *topological neighborhoods* around the source/observation point.

We believe that this topological difference has not received the attention it deserves in the growing theoretical and methodological literature on nonlocal media. In particular, the author believes that a majority of present approaches to nonlocal metamaterials conflate the topologically local (but EM nonlocal) domain of small neighborhoods and global domains. However, general topology and much of modern mathematical physics is based on clearly distinguishing the last two topological levels. Explicating these subtle conceptual differences emanating from the existence of distinct types of spatial scales in field–matter interactions, while aided by a precise, rigorous, and powerful mathematical language, is one of the principal aims of this work. In fact, we believe that a complete understanding of material nonlocality in nature cannot be attained without relying on a fairy advanced mathematical apparatus such as the theory of smooth fiber bundles and infinite-dimensional manifolds developed below.

Let us give a brief summary of the main conceptual findings of this research. First, we highlight that the main idea of the superspace formalism is not restricted to electromagnetic theory, but applies to all types of nonlocal continuum theories, i.e., field theories in nonlocal continuous media. However, for concreteness, and in order to reduce the complexity of the mathematical formalism, we chose to work with a specific type of field theories, namely the classical field paradigm based on Maxwell's equations. As will be seen below, It turns out that the standard formalism of local field theory, which is based on spacetime points and their *differential* (but not topological) neighborhoods, viewed as the basic configuration space of the problem, is *not* the most natural or convenient framework for formulating field theory in nonlocal materials. This is mainly because the *physics*-based domain of nonlocality (to be defined precisely below), which captures the effective region of field–matter nonlocal interactions, is found to not always be naturally transportable into the mathematical formalism of boundary-value problems characteristic of classical field theory, as practiced in several domains, such as applied electromagnetism, heat transfer, hydrodynamics, etc. By investigating the subject from an alternative but enlarged and intrinsically broader perspective, it will be shown that a natural space for conducting nonlocal metamaterials research is the *vector bundle* structure, more specifically, a Banach bundle [26] where every element in the fiber superspace is a vector field on the entire domain of nonlocality.

The main result of this paper is that every generic nonlocal domain can be topologically described by a superspace comprised of a Banach (infinite-dimensional) vector bundle $\mathcal{M}$. If two materials described by their corresponding vector bundles $\mathcal{M}_1$ and $\mathcal{M}_2$ are juxtaposed, then one may use topological methods to combine them and to compare their topologies. The present paper's focus is mainly on the first part, i.e., how to construct the material bundle $\mathcal{M}$. That is, the derivation of the various vector bundle structures starting from a generic phenomenological model of electromagnetic nonlocality is the main contribution of the present work. It is hoped by the author that the superspace theory developed below will stimulate new approaches to computational field theories by adopting methods borrowed from or inspired by computational topology and differential topology to help supporting ongoing efforts to solve challenging problems in complex material domains as in nanoscale hydrodynamics, nonlocal optical materials, topological insulators, topological photonic devices, and other areas where nonlocality is currently important or expected to play an increasingly dominant role in the future.

### 2.3. An Outline of the Present Work

Because of the considerable complexity of the present article, which is unavoidable in treatment of the subject of nonlocality in the continuum field theory at this broad theoretical level, and in order to help make our contribution accessible to a wider audience involving, for instance, physicists, engineers, and mathematicians, we have divided the argument into different stages with different flavors, as follows. First, Section 3 provides a general mathematical description of nonlocality in the continuum field theory, emphasizing the settings of the electromagnetic case. The key ingredients of nonlocal metamaterials/materials are illustrated in Section 3.1 using an abstract excitation-response model. This is followed in Section 3.2 by a more detailed description of the special but important case of spatial dispersion, which tends to arise naturally in many investigations of nonlocal metamaterials. In Section 4, we begin the elucidation of the main topological ideas behind electromagnetic nonlocality, most importantly, the concept of EM *nonlocality microdomains*, which provides the key link between physics, material engineering, and topology in this paper. The various physical and mathematical structures are spelled out explicitly, followed in Section 5 by a more careful construction of a natural fiber bundle superspace structure that appears to satisfy simultaneously both the physical and mathematical requirements of EM nonlocality (Sections 5.1 and 5.2). We then provide a key computational application of the proposed theory in Section 5.3, where it is shown that the material response function is representable as a special fiber bundle homomorphism over the metamaterial base space. In this way, a more general map than linear operators in local field theory is derived, providing solid

mathematical foundations for possible future computational topological methods where, for example, the bundle homomorphism itself might be discretized instead of the original spacetime-based linear operator. The fiber bundle superspace algorithm is summarized in Section 6, where it is highlighted that the main data needed are the physics-based (e.g., electromagnetic) nonlocality microdomains, which do not arise solely from purely mathematical considerations, but require some empirical input, for example the microscopic theory of materials, which ultimately would involve both electromagnetism and quantum mechanics. In this manner, the entire construction of the nonlocal metamaterial superspace may proceed as per the procedure outlined there. In order to illustrate how the above mentioned microdomain structure can be actually estimated in practice, in Section 7 we present a fairly detailed computational example based on nonlocal semiconductors, where we also explore in depth the physical origin of nonlocality in this particular setting. Insights into the lack of general EM boundary conditions in nonlocal EM are provided in Section 8 based on the superspace formalism.

This paper provides a series of technical appendices designed to provide necessary information to expand the scope of the treatment found in the main text. In Appendix A.1, we back up our major formulation as developed by introducing a general review of electromagnetic nonlocality targeting a wide audience of mathematicians, physicists, engineers, and applied scientists. This review does not restrict itself to specific types of materials, such as plasma, metals, and semiconductors, but aims at integrating the author's own understanding of the vast literature on the subject in a tentative and necessarily provisional, but somehow more coherent view. Because of the extreme importance of the special case of spatial dispersion for understanding nonlocality, we provide some brief historical remarks on this subject in a separate Appendix A.2. Some technical and historical explications of the concept of superspace, as needed and used in the main text, is given in Appendix A.4, which is not meant as a complete rigorous introduction to the concept of superspace in mathematics and theoretical physics, a topic far from being well-defined and focused. Instead, the goal of this appendix is to fix the very specific meaning we have in mind in this paper whenever we speak about superspace structures in order to avoid confusing our concept with other usages found in physics, such as in supersymmetry.

The Appendixes A.6–A.9 supply important technical information needed in order to fully comprehend the specific main example developed in this paper, to illustrate the use of the superspace formalism in actual real-life scenarios (the inhomogeneous nonlocal excitonic semiconductor material system of Section 7). We opted to separate the content of these appendices from the main text in order to simplify the presentation. The subject of nonlocal semiconductor metamaterials is already well-known in the specialized literature, but is also highly technical. In order to help keep the flow of the various ideas treated in the main text tightly focused on the conceptual and mathematical aspects of our proposed superspace theory, we relegated some background material, especially detailed derivations and explanations more related to semiconductor physics than the superspace formalism, to the three appendices mentioned above.

Some basic familiarity with vector bundles and Banach spaces is assumed, but essential definitions and concepts will be reviewed briefly within the main formulation and references where more background on vector bundles can be found will be pointed out. The paper intentionally avoids the strict theorem-proof format to make it accessible to a wider audience. Most of the time we give only proof sketches and leave out straightforward but lengthy computations. In general, just the very basic definitions of smooth manifolds, vector bundles, Banach spaces, etc., are needed to comprehend this theory (also see Appendix A.5 for a guide to the mathematical background.) The only place where the treatment is mildly more technical is in Section 5.3 when the bundle homomorphism is constructed using partition of unity technique as a detailed computational application of the superspace theory.

In Appendixes A.3 and A.11, various additional current and future applications to fundamental methods, applied physics, and engineering are outlined in brief form. Some of

the applications mentioned there, for instance numerical methods and topological devices, appear to us to be directly relevant to the scope of a superspace extension of conventional nonlocal electromagnetic field continuum theory, such as the one attempted below within the main text. On the other hand, some of the other applications discussed there, e.g., digital communications and energy, are of a more general nature and belong to our broader tentative global review of the subject of nonlocality in nature and engineering attempted in the Appendix A sections of this paper. Finally we end with the conclusion.

### 3. The Nonlocal Continuum Response Model

*3.1. A Generic Nonlocal Response Model in Inhomogeneous Continua*

In order to introduce the concept of nonlocality in the simplest way possible, let us first start with a scalar field theory setting. As mentioned in the introduction, vacuum classical fields cannot exhibit nonlocality, so in order to attain this phenomenon, one must consider fields in specialized domains. We therefore kick-start the technical mathematical treatment by reviewing the broad theory of such media. The goal is to outline the main ingredients of the spacetime-based *configuration space* on which such theories are often founded in literature. To further simplify the presentation, we work in the regime of *linear response theory*: i.e., all material media considered throughout this paper are assumed to be linear with respect to field excitation.

In detail, if the medium response and excitation fields are captured by the spacetime functions $R(\mathbf{r}, t)$ and $F(\mathbf{r}, t)$, respectively, then the most general response is given by an operator equation of the form [9]

$$R(\mathbf{r}, t) = \mathcal{L}\{F(\mathbf{r}, t)\}, \tag{1}$$

where $\mathcal{L}$ is a linear operator describing the medium, and is ultimately determined by the laws of physics relevant to the structure under consideration [27–29].

Now, the entire physical process will occur in a spacetime domain. In a nonrelativistic formulation (like the one in the present work), we intentionally separate and distinguish space from time. Therefore, let us consider a process of field–matter interactions where $t \in \mathbb{R}$, while we spatially restrict to a "small" region spanned by the position

$$\mathbf{r} \in D \subset \mathbb{R}^3, \tag{2}$$

where $D$ is an open set containing $\mathbf{r}$. (Throughout this paper, we assume the normal Euclidean topology on $\mathbb{R}^3$ for all spatial domains.) Since the operator $\mathcal{L}$ is linear, one may argue (informally) that its associated *Green's function* or *kernel function*

$$K(\mathbf{r}, \mathbf{r}'; t, t') \tag{3}$$

must exist. Strictly speaking, this is not correct in general and one needs to prove the existence of the Green's function for every given linear operator on a case by case basis by actually constructing one [30,31].[2] However, we will follow (for now) the common trend in physics and engineering by assuming that linearity alone is enough to justify the construction of Green's function. If this is accepted, then we can immediately infer from the very definition of the Green's function itself that [12,32]

$$R(\mathbf{r}, t) = \int_D \int_{\mathbb{R}} \mathrm{d}^3 r' \, \mathrm{d}t' \, K(\mathbf{r}, \mathbf{r}'; t, t') F(\mathbf{r}', t'). \tag{4}$$

The relation (4) represents the most general response function of a (scalar) material medium valid for linear field–matter interaction regimes [37,38]. The kernel (Green) function $K(\mathbf{r}, \mathbf{r}'; t, t')$ is often called the *medium response function* [9,32,37].

If we further assume that all of the material constituents of the medium are time invariant (the medium is not changing with time), then the relation (4) maybe replaced by

$$R(\mathbf{r}, t) = \int_D \int_{\mathbb{R}} \mathrm{d}^3 r' \, \mathrm{d}t' \, K(\mathbf{r}, \mathbf{r}'; t - t') F(\mathbf{r}', t'), \tag{5}$$

where the only difference is that the kernel function's temporal dependence is replaced by $t - t'$ instead of two separated arguments. Such superficially small difference has nevertheless considerable consequences. Most importantly, by working with (5) instead of (4), it becomes possible to apply the Fourier transform method to simplify the time-dependent formulation of the problem [39]. Indeed, taking the temporal Fourier transform of both sides of (5) leads to

$$R(\mathbf{r}, \omega) = \int_D \mathrm{d}^3 r' K(\mathbf{r}, \mathbf{r}'; \omega) F(\mathbf{r}'; \omega), \tag{6}$$

where the Fourier spectra of the fields are defined by

$$F(\mathbf{r}; \omega) := \int_{\mathbb{R}} \mathrm{d}t F(\mathbf{r}; t) e^{-\mathrm{i}\omega t},$$

$$R(\mathbf{r}; \omega) := \int_{\mathbb{R}} \mathrm{d}t R(\mathbf{r}; t) e^{-\mathrm{i}\omega t}. \tag{7}$$

On the other hand, the medium response function's Fourier transform is given by the essentially equivalent formula

$$K(\mathbf{r}, \mathbf{r}'; \omega) := \int_{\mathbb{R}} \mathrm{d}(t - t') K(\mathbf{r}, \mathbf{r}'; t - t') e^{-\mathrm{i}\omega(t - t')}. \tag{8}$$

In this paper, we focus on time invariant material media and, hence, work exclusively with frequency domain expressions, such as (6), (7), and (8), though we often suppress the frequency dependence on $\omega$ in order to simplify the notation whenever no confusion would arise.

The generalization to the three-dimensional (full-wave) electromagnetic picture is straightforward when the dyadic formalism is employed [28,40]. The relation corresponding to (4) is

$$\mathbf{R}(\mathbf{r}, t) = \int_D \mathrm{d}^3 r' \int_{\mathbb{R}} \mathrm{d}t' \, \overline{\mathbf{K}}(\mathbf{r}, \mathbf{r}'; t - t') \cdot \mathbf{F}(\mathbf{r}', t'), \tag{9}$$

where we replaced the scalar fields $F(\mathbf{r})$ and $R(\mathbf{r})$ by vector fields $\mathbf{F}(\mathbf{r}), \mathbf{R}(\mathbf{r}) \in \mathbb{R}^3$. The kernel function $K$, however, must be transformed into a *dyadic* function (tensor of second rank) [14,28,41,42]:

$$\overline{\mathbf{K}}(\mathbf{r}, \mathbf{r}'; t - t'). \tag{10}$$

In the (temporal) Fourier domain, (9) becomes

$$\mathbf{R}(\mathbf{r}, \omega) = \int_D \mathrm{d}^3 r' \, \overline{\mathbf{K}}(\mathbf{r}, \mathbf{r}'; \omega) \cdot \mathbf{F}(\mathbf{r}'; \omega), \tag{11}$$

where

$$\overline{\mathbf{K}}(\mathbf{r}, \mathbf{r}'; \omega) := \int_{\mathbb{R}} \mathrm{d}(t - t') \, \overline{\mathbf{K}}(\mathbf{r}, \mathbf{r}'; t - t') e^{-\mathrm{i}\omega(t - t')} \tag{12}$$

is the frequency domain response kernel, while

$$\mathbf{F}(\mathbf{r}; \omega) := \int_{\mathbb{R}} \mathrm{d}t \, \mathbf{F}(\mathbf{r}; t) e^{-\mathrm{i}\omega t},$$

$$\mathbf{R}(\mathbf{r}; \omega) := \int_{\mathbb{R}} \mathrm{d}t \, \mathbf{R}(\mathbf{r}; t) e^{-\mathrm{i}\omega t}, \tag{13}$$

are the corresponding frequency domain excitation and response fields, respectively.

The essence of electromagnetic nonlocality can be neatly captured by the mathematical structure of the basic relation (9). It says that the field response $\mathbf{R}(\mathbf{r})$ is determined not only by the excitation field $\mathbf{F}(\mathbf{r}')$ applied at location $\mathbf{r}'$, but at all points $\mathbf{r}' \in D$. Consequently, here we find that the following is true:

> *In nonlocal continuum field theories, knowledge of the field response at a specific point $\boldsymbol{r}$ requires knowledge of the cause (excitation field) on an entire topological neighborhood set $D \ni \boldsymbol{r}$.*

On the other hand, if the medium is local, then the material response function can be written as

$$\overline{\mathbf{K}}(\mathbf{r}, \mathbf{r}'; \omega) = \overline{\mathbf{K}}_0(\omega)\delta(\mathbf{r} - \mathbf{r}'), \tag{14}$$

where $\overline{\mathbf{K}}_0$ is a spatially constant tensor and $\delta(\mathbf{r} - \mathbf{r}')$ is the three-dimensional Dirac delta function. In this case, (11) reduces to [13]

$$\mathbf{R}(\mathbf{r}; \omega) = \overline{\mathbf{K}}_0(\omega) \cdot \mathbf{F}(\mathbf{r}; \omega), \tag{15}$$

which is the standard constitutive relation of linear electromagnetic materials. Clearly, (15) says that only the exciting field $\mathbf{F}(\mathbf{r})$ data at $\mathbf{r}$ is needed in order to induce a response at the same location. In a nutshell, locality implies that the natural configuration space of the electromagnetic problem is just the point-like spacetime manifold $D \subset \mathbb{R}^3$ or the entire Euclidean space $\mathbb{R}^3$.

**Remark 1 (Infinitesimal domains).** One may use the "infinitesimally immediate vicinity" of a given point $\mathbf{r}$, where a response is sought, for computing that response itself, yet while still remaining within the *local* regime of continuum field theory. Indeed, for the case of electromagnetic theory, we note that, according to the constitutive relation (15), while only the exciting field at $\mathbf{r}$ is required for computing the response, Maxwell's equations themselves, on the other hand, *still* must be coupled with the local constitutive relation model of the problem. Now, the fact that Maxwell's equations are *differential* equations implies that the "largest" domain beside the point $\mathbf{r}$ needed for carrying out the mathematical description of the details of the relevant field–matter interaction physics is just the region *infinitesimally* close to $\mathbf{r}$. In other words, in continuum field theories, infinitesimal domains should be treated as neither topological domains nor neighborhoods. The infinitesimal belong to *any* type of continuum field theory built on the differential calculus and, hence, is not a criterion for distinguishing local and nonlocal theoretical structures.

Conventional boundary-value problems in applied electromagnetism are formulated in this manner, i.e., with a three-differential manifold as the main problem space on which spatial fields live [28,29,32,40,41,43–46]. Note that, strictly speaking, the full configuration space in local electromagnetism (also called *normal optics* [16]) is the four-dimensional manifolds $D \times \mathbb{R}$ or $\mathbb{R}^4$ since either time $t$ or the (temporal) circular frequency $\omega$ must be included to engender a full description of electromagnetic fields. However, nonlocal materials are most fundamentally a *spatial* type of materials/metamaterials where it is the spatial structure of the field what carries most of the physics involved [32,47]. For that reason, *throughout this paper, we investigate the required configuration spaces with focus mainly on the spatial degrees of freedom.* This will naturally lead to the discovery of the fiber bundle structure of nonlocality, the main topic of the present work.

*3.2. Spatial Dispersion in Homogeneous Nonlocal Material Domains*

Spatial dispersion is considered by some researchers as one of the most promising routes toward nonlocal metamaterials, e.g., see [16,47–49]. It is by large the most intensely investigated class of nonlocal media, receiving both theoretical and experimental treatments by various research groups since the early 1960s.[3] The basic idea is to restrict electromagnetism to the special, but important case of media possessing *translational symmetry*, an

important special scenario of material nonlocality that holds when the medium is *homogeneous*. In such situation, the material tensor function satisfies

$$\overline{\mathbf{K}}(\mathbf{r}, \mathbf{r}'; \omega) = \overline{\mathbf{K}}(\mathbf{r} - \mathbf{r}'; \omega). \tag{16}$$

The spatial Fourier transforms are defined by

$$\overline{\mathbf{K}}(\mathbf{k}, \omega) := \int_{\mathbb{R}^3} \mathrm{d}^3(r - r') \, \overline{\mathbf{K}}(\mathbf{r} - \mathbf{r}'; \omega) e^{i\mathbf{k} \cdot (\mathbf{r} - \mathbf{r}')}, \tag{17}$$

with

$$\mathbf{F}(\mathbf{k}, \omega) := \int_{\mathbb{R}^3} \mathrm{d}^3 r \, \mathbf{F}(\mathbf{r}; t) e^{i\mathbf{k} \cdot \mathbf{r}},$$
$$\mathbf{R}(\mathbf{k}, \omega) := \int_{\mathbb{R}^3} \mathrm{d}^3 r \, \mathbf{R}(\mathbf{r}; t) e^{i\mathbf{k} \cdot \mathbf{r}}. \tag{18}$$

After inserting (16) into (11), taking the spatial (three-dimensional) Fourier transform of both sides, the following equation is obtained:

$$\mathbf{R}(\mathbf{k}, \omega) = \overline{\mathbf{K}}(\mathbf{k}, \omega) \cdot \mathbf{F}(\mathbf{k}, \omega). \tag{19}$$

The dependence of $\overline{\mathbf{K}}(\mathbf{k}, \omega)$ on the wave vector ("spatial frequency") $\mathbf{k}$, here added to the already existing temporal frequency $\omega$ dependence, is *the* signature of spatial dispersion. As a spectral transfer function of *a homogeneous* medium, $\overline{\mathbf{K}}(\mathbf{k}, \omega)$ includes *all* the information needed to compute the nonlocal material domain's response to arbitrary spacetime field excitation functions $\mathbf{F}(\mathbf{r}, t)$ (through the application of inverse four-dimensional Fourier transform [16]).

**Remark 2.** In several treatments of the subject within electromagnetic theory, the excitation field is taken as the electric field $\mathbf{E}(\mathbf{r}, t)$, while the response function is $\mathbf{D}(\mathbf{r}, t)$. In such formulation, the material tensor function $\overline{\mathbf{K}}(\mathbf{k}, \omega)$ takes into account *both* electric and magnetic effects [14–16,37–39,50–54]. This is different from the permittivity tensor often invoked in local electromagnetism [28], which is ultimately based on the popular *multipole* model [43] of electromagnetic interactions in material media. A comparison between the two material response formalisms, the one based on $\overline{\mathbf{K}}(\mathbf{k}, \omega)$ and the multipole model, is given in [32,47,53].

Complex heterogeneous arrangements of various nonlocal materials can be realized by juxtaposing several subdomains where each subunit is homogeneous, hence can be described by a spatial dispersion profile of the form $\overline{\mathbf{K}}(\mathbf{k}, \omega)$ discussed above. The idea is that even materials that are inhomogeneous at a given spatial scale may become homogeneous at a different (less refined) spatial level, leading to a "grid-like" spatially dispersive cellular building blocks at the lower level. In Figure 1, we show a nonlocal metamaterial system with various multiscale structures. A large nonlocal domain, e.g., $\overline{\mathbf{K}}_3(\mathbf{r}, \mathbf{r}')$ in the figure, acts like a "substrate" holding together several other smaller material constituents, such as $\overline{\mathbf{K}}_n(\mathbf{r}, \mathbf{r}'), n = 1, 2, 4$. We envision that each nonlocal subdomain may possess its own specially tailored nonlocal response function profile serving one or several applications.[4] By concatenating multiple regions, interfaces between subdomains with different material constitutive relations are created. We here show subdomains $\mathcal{D}_n, n = 1, 2, 3, 4$, while some of the possible intermaterial interfaces include $\mathcal{D}_1/\mathcal{D}_2, \mathcal{D}_1/\mathcal{D}_3, \mathcal{D}_2/\mathcal{D}_3, \mathcal{D}_3/\mathcal{D}_4$. More complex geometrical and topological interfaces than those shown in Figure 1 are possible where the topological type of the interface manifold can be controlled by introducing handles, holes, gluing, cutting, and so on.

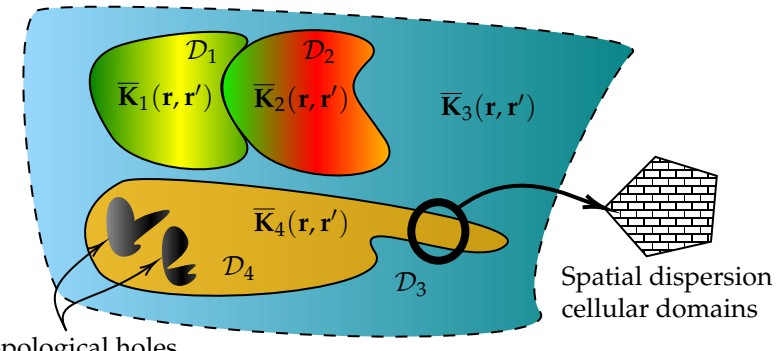

**Figure 1.** A generic depiction of an electromagnetic nonlocal metamaterial system. Each of the domains $\mathcal{D}_n$ is captured by a general linear nonlocal response function $\overline{\mathbf{K}}_n(\mathbf{r}, \mathbf{r}')$.

Recall that in *local* electromagnetism each intermaterial interface should be assigned a special electromagnetic boundary condition in order to ensure the existence of a unique solution to the problem [9,41]. This, however, is not possible in nonlocal electromagnetism. Indeed, and as already mentioned earlier, nonlocal electromagnetism introduces several subtle issues that are absent in the local case: *additional boundary conditions* are often invoked to handle the transition of fields along barriers separating different domains, such as between two nonlocal domains, or even one nonlocal and another local domain [16,55,56]. The topological fiber bundle theory to be developed in Section 5 will provide a clarification of why this is so since it turns out that the traditional spacetime approach often employed in local electromagnetism is not necessarily the most natural one (see also Section 8). There is a need, then, to examine in a more in-depth fashion the detailed structural phenomena associated with the presence of *multiple topological scales* in nonlocal metamaterials. This paper will provide some new insights into these issues.

*3.3. Preliminary Remarks on the Existence of Multiple Topological Scales in Nonlocal Continuum Field-Theoretic Structures*

For completeness and maximal clarity, we discuss here some of the directly observable topological scales in nonlocal continuum systems whose preliminary understanding at this stage of our presentation does not require the use of the quite elaborate mathematical apparatus to be carefully constructed in the remaining parts of this paper. We list the most important of these topological levels as follows:

1.  The first is the geometrical separation between different nonlocal domains, such as $\mathcal{D}_1$ and $\mathcal{D}_2$ discussed in Section 3.2 and illustrated by Figure 1.
2.  The second is the case captured by the inset in the right hand side in Figure 1. Fine "microscopic" cells, each homogeneous and, hence, describable by a response function of the form $\overline{\mathbf{K}}(\mathbf{k}, \omega)$, can be combined to build up a complex effective nonlocal response tensor $\overline{\mathbf{K}}_n(\mathbf{r}, \mathbf{r}')$ over its topologically global domain $\mathcal{D}_n$. Such juxtaposition at the microscopically local level that effectively leads to the emergence of a global behavior is a classic example of multiscale physics. However, note that it even acquires a higher importance in the present context due to the fact that *both* of the constituent cell level (rectangular "bricks" in the inset of Figure 1) and the global domain level $\mathcal{D}_n$ already belong to the *physically*, e.g., electromagnetically, nonlocal dimension of the relevant nonlocal continuum field theory.
3.  Finally, the third directly observable topological scale is that connected to what we termed "topological holes" in Figure 1. These are arbitrarily-shaped gaps, such as holes, vias, etchings, etc., which are intentionally introduced in order to influence the electromagnetic response by modifying the topology of the three-dimensional material manifolds $\mathcal{D}_n$.

The above topological levels are called "directly observable" because their determination does not require the use of abstract and advanced concepts from continuum field theory. This is in contrast to the more subtle distinction that will be discussed next.

In Remark 3, we discuss the very important conceptual distinction between topology-based and physics-based nonlocal domains, a demarcation between two concepts that has already been invoked several times above, and will also figure up repeatedly throughout the remaining parts of this article.

**Remark 3** (**Distinction between physics- and topology-based locality/nonlocality**). The terms *local* and *global* possess two different senses, one physics-based, e.g., electromagnetic theory; the other is spatio-geometric in essence, belonging to the purely formal and mathematical dimensions of the structure of the nonlocal continuum theory of the material system. Elucidating this subtle interconnection between the two senses will be one of the main objectives of the present work but we will first need to introduce the various relevant microscale topological concepts to be given in Section 4 (see also Remark 17) For the time being, let the following be known:

1. *Physics-based local/non-local distinction*: this is where basically physical considerations are at stakes. We distinguish between:

   (a) *Physics-based non-local level*: this includes how the response of the material continuum depends on locations $\mathbf{r}'$ *not* infinitesimally close to the point $\mathbf{r}$ where the excitation field is applied. That is, $\mathbf{r} - \mathbf{r}'$ is nonzero but it is also not a differential. (On infinitesimal domains, see Remark 1.)

   (b) *Physics-based local level*: this is the physical regime whose essence is captured by local constitutive relations of the form (15).

2. *Topology-based local/non-local distinction*: mathematical considerations dominate at this level. We have:

   (a) *Topology-based non-local level*: this is the topologically global level, e.g., the entire topological manifold in contrast to the local description applicable only to a coordinate patch [57], and so on. At this level, the non-local-as-global is an emerging structure based on gluing together "smaller pieces" of the total manifold. We will see examples of processes occurring basically at this level when we use partition of unity methods.

   (b) *Topology-based local level*: this is the topological layer associated with structures, such as open sets, topological neighborhoods, closed sets, and so on. A topological space is defined as a collection of all such local sets [58,59].

The two concepts outlined above interact with each. There is a subtle relation between physics and topology. This paper will address some of these delicate interrelations in subsequent sections.

**Remark 4** (**Electromagnetic Domains**). For simplicity, in what follows we will occasionally use 'electromagnetic (EM) domain' and 'physics-based nonlocal domain' as interchangeable terms. It should be kept–in min–that the concept of physics-based nonlocality is broader than EM nonlocality. The former refers to a characteristic structural trait enjoyed by all nonlocal continuum field theories, while the latter is restricted to the realm of just one such theory, that of the electromagnetism of continuous media.

## 4. The Microscopic Topological Structure of Physics-Based Nonlocal Domains
### 4.1. Introduction

In this section, we begin our careful examination of the mode of interrelation between the physics- and topology-based types of nonlocality introduced and discussed above.[5] Let the nonlocality domain of the electromagnetic medium, the region $D \subset \mathbb{R}^3$ in (11), be bounded. Corresponding to (1), a similar operator equation in the frequency domain

representing the most general form of a nonlocal electromagnetic medium can be posited, namely

$$\mathbf{R}(\mathbf{r};\omega) = \mathcal{L}_\omega\{\mathbf{F}(\mathbf{r};\omega)\}, \tag{20}$$

where the nonlocal medium linear operator is itself frequency dependent. For simplicity, and as stated before, whenever it is understood from the context that the material response operator is formulated in the frequency domain, all dependencies on $\omega$ appearing in its formal expression will be removed.

We are going to propose a change in the mathematical framework inside which electromagnetic nonlocality is usually defined. This will be done in two stages:

- Initially, in the present Section, we introduce the rudiments of the main physics-based microtopological structure associated with nonlocality in continuum field theories, but without delving into considerable mathematical details. The aim is to familiarize ourselves with the minimal necessary physical setting and how it naturally gives rise to a more refined picture of the nonlocal material domain compared with the traditional (and much simpler) topological structure of local electromagnetism based on spacetime points.
- In the second stage, treated in Section 5, a more careful mathematical picture is developed using the theory of topological fiber bundles. We eventually show (Section 5.3) that the physics-based (in this case the electromagnetic) nonlocal operator (20) can be reformulated as a Banach bundle map (homomorphism) over the three-dimensional space of the material domain under consideration. Some computational examples and applications are provided in the later Sections, e.g., see Section 7.

The key conceptual idea behind the entire theory presented here is that of the *topological microdomains* associated with the field theory of nonlocal continua, e.g., the electromagnetism of continuous media, which we first develop thematically in the next Section 4.2 before moving subsequently to the more rigorous and exact topological formulation of Section 5.

### 4.2. The Concept of Topological Microdomains in Nonlocal Continuum Field Theories

In conventional frequency domain local electromagnetism, the boundary-value problem of multiple domains is formulated as a set of coupled partial differential equations or integro-differential equations interwoven with each other via the appropriate intermaterial interface boundary conditions dictating how fields change while crossing the various spatial regions inside which the equations hold [28,40,41]. This has been traditionally achieved by taking up the electromagnetic response function $\overline{\mathbf{K}}(\mathbf{r}, \mathbf{r}'; \omega)$ as an essential key ingredient of the problem description, which traditionally has been exploited in two stages: First, the constitutive relations would enter into the governing equations in each separate solution domain. Second, the constitutive relations themselves are used in order to construct the proper electromagnetic boundary conditions prescribing the continuity/discontinuity behavior of the sought field solutions as they move across the various interfaces separating domains with different material properties.

Unfortunately, it has been well known for a long time that it is not possible to formulate a universal electromagnetic boundary condition for nonlocal media, especially for the case of spatial dispersion. This will be discussed later with more details in Section 8, but also see the discussion around additional boundary conditions (ABC) in Appendix A.1. For now, we concentrate on gaining a deeper understanding of the generic structure of spatial nonlocality in continuum field theories.

Consider the microdomain structure depicted in Figure 2. A key starting observation is how nonlocality forces us to associate with every spacetime point $(\mathbf{r}, t)$, or frequency space point $(\mathbf{r}; \omega)$, a *topological neighborhood* of $\mathbf{r}$, say $V_\mathbf{r}$, such that $\mathbf{r} \in V_\mathbf{r}$. For now, let us assume that the spatial material domain $D \ni \mathbf{r}$ is just an *open set* in the technical sense of the topology of the Euclidean space $\mathbb{R}^3$ inherited from the standard Euclidean metric [59]. By restricting $D$ to be open, we avoid the notorious problem of dealing with boundaries or interfaces between such (possibly overlapping) open sets. That is, the topological closure

of $D$, denoted by $\mathrm{cl}(D)$, is excluded from the domain of nonlocality. Let $D$ be the maximal such topological neighbored for the problem under consideration.[6] We now associate with each point $\mathbf{r}$ a "smaller" open set $V_{\mathbf{r}}$ where the following holds:

$$\forall\, \mathbf{r} \in D,\ \exists\, V_{\mathbf{r}} \subset D \text{ such that } \mathbf{r} \in V_{\mathbf{r}}, \text{ and } V_{\mathbf{r}} \text{ is open.} \tag{21}$$

Note that the assumed openness of $D$ makes the above construction technically possible. We will call the proposition (21) the *principle of nonlocal microdomain generation*. It formally captures the main content of the structure of nonlocality at the microscopic level. In Section 7, a practical example taken from nonlocal semiconductor metamaterials will be investigated in depth in order to illustrate the applicability of (21).

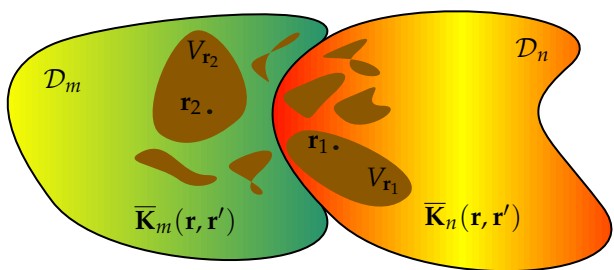

**Figure 2.** The microtopological structure of nonlocal metamaterial systems includes more than just the three-dimensional spatial domains $\mathcal{D}_n, n = 1, 2, \ldots$ It is best captured by classes $\mathcal{V}(\mathcal{D}_n)$ composed of various open sets $V_{\mathbf{r}} \subset \mathcal{D}_n$ based at each point $\mathbf{r} \in \mathcal{D}_n$. On every such subset a vector field is defined, representing the external field excitation field. The collection of all vector fields on a given set $V_{\mathbf{r}}$ gives rise to a linear topological function space $\mathcal{F}(V_{\mathbf{r}})$. The topologies consisting of the base spaces $\mathcal{D}_n$, the nonlocal microdomains $V_{\mathbf{r}}$, and the function spaces $\mathcal{F}(V_{\mathbf{r}})$, collectively give rise to a total "macroscopic" topological structure (superspace) that is considerably more complex than the base spaces $\mathcal{D}_n$.

Now, instead of considering fields like $\mathbf{R}(\mathbf{r})$ and $\mathbf{F}(\mathbf{r})$ defined on the entire maximal domain of nonlocality $D$ (which can grow "very large") we propose to reformulate the problem of nonlocal continua as a *topologically local*[7] structure by exploiting the fact that the physics of field–matter interactions gives the field response at location $\mathbf{r}$ due to independent excitation fields essentially confined within a "smaller domain" around $\mathbf{r}$, namely the open set $V_{\mathbf{r}}$.[8]

Furthermore, if the response at another *different* point $\mathbf{r} \neq \mathbf{r}'$ is needed, then a *new*, generally different, "small" open set $V_{\mathbf{r}'}$ will be required. That is, in general we allow that

$$V_{\mathbf{r}} \neq V_{\mathbf{r}'} \tag{22}$$

even though it is expected that typically there should be some overlap between these two small local domains of electromagnetic nonlocality in the sense that

$$V_{\mathbf{r}} \cap V_{\mathbf{r}'} \neq \varnothing, \tag{23}$$

especially if the nonlocality radius $|\mathbf{r} - \mathbf{r}'|$ is small.

The following fundamental collection of "smaller" sets, where a metric scale characterizing "smallness" is not implied, written as

$$\{V_{\mathbf{r}}, \mathbf{r} \in D\}, \tag{24}$$

will be dubbed *nonlocal microdomains*, or just *microdomains* in short. A possible precise definition is given next.

**Definition 1** (**Nonlocal microdomains: the physics-based scenario**). Consider a material domain $D$ with the associated nonlocal response function $\overline{\mathbf{K}}(\mathbf{r}', \mathbf{r})$. We define the (physics-based) nonlocal microdomain $V_{\mathbf{r}} \subset D$, labeled by $\mathbf{r} \in D$, as the interior of the compact[9] support of $\overline{\mathbf{K}}(\mathbf{r}', \mathbf{r})$. The support itself is defined by the standard formula

$$\operatorname{supp} \overline{\mathbf{K}}(\mathbf{r}', \mathbf{r}) := \operatorname{cl}_D \{\mathbf{r}' \in D,\ \|\overline{\mathbf{K}}(\mathbf{r}', \mathbf{r})\| \neq 0\}, \tag{25}$$

where $\|\cdot\|$ is a suitable tensor norm, for example the matrix norm.[10] The topological closure operator $\operatorname{cl}_D$ is here taken with respect to the total material space $D$ where the latter is viewed as a topological space on its own.

**Remark 5** (**Microdomain topology**). By Definition 1 above, the nonlocal microdomain $V_{\mathbf{r}}$ is always open. It can be shown that the collection of open sets $\{V_{\mathbf{r}}, \mathbf{r} \in D\}$ induces a topology on the total space occupied by the nonlocal material (the details are omitted since they are lengthy though straightforward.) In what follows, this topology will be referred to by the term *microdomain topology*. The set of physics-based nonlocality microdomains (microdomains for short), as constructed in Definition 1, explicate the fine microtopological structure of nonlocal electromagnetic domains at a spatial scale different from that of the (topologically "larger") material domain $D$ itself and are fundamental for the theory developed in this paper.

**Remark 6** (**Discrete topology in local continua**). In local media, the microdomains topology reduces to the trivial discrete topology

$$\{\{\mathbf{r}\}, \mathbf{r} \in D\} \tag{26}$$

since the external field interacts only with the point $\mathbf{r}$ at which it is applied and hence

$$V_{\mathbf{r}} = \{\mathbf{r}\} \tag{27}$$

holds as the "smallest" possible topological microdomain in that rather special case. Therefore, the microdomain topology is interesting only for the case of physics-based nonlocality, e.g., the scenario of EM microdomains discussed in more details in the examples and applications below. In particular, from the point of view of this article, *local* metamaterials are not topologically interesting.

*4.3. Construction of Excitation Field Function Spaces on the Topological Microdomains of Nonlocal Media*

After enriching the MTM domain $D$ with the finer topology of nonlocality microdomains $V_{\mathbf{r}}, \mathbf{r} \in D$, we wish to equip this total medium with additional mathematical structure based on the physics of field–matter interaction. Consider the set of all sufficiently differentiable vector fields $\mathbf{F}(\mathbf{r})$ defined on $V_{\mathbf{r}}, \mathbf{r} \in D$. This set possesses an obvious complex *vector space structure*: for any two complex numbers $a_1, a_2 \in \mathbb{C}$, the sum

$$a_1 \mathbf{F}_1(\mathbf{r}) + a_2 \mathbf{F}_2(\mathbf{r})$$

is defined on $V_{\mathbf{r}}$ whenever $\mathbf{F}_1(\mathbf{r})$ and $\mathbf{F}_2(\mathbf{r})$ are, while the null field plays the role of the origin. In what follows, we will denote such function spaces by $\mathcal{F}(V_{\mathbf{r}})$ or just $\mathcal{F}$ if it is understood from the context on which material spatial domains the fields are defined.

**Remark 7** (**The excitation field function space and Sobolev spaces**). It is possible to equip $\mathcal{F}(V_{\mathbf{r}})$ with a suitable topology in order to measure how "near" to each other are any two fields defined on $V_{\mathbf{r}}$, e.g., see [59,62,63]. Therefore, in this manner $\mathcal{F}(V_{\mathbf{r}})$ acquires the structure of a *topological vector space* [59]. In particular, it can be made a *Sobolev space*, where the latter is not only a Banach space (normed space), but also a Hilbert space (inner product space) [64–66].

The detailed construction of a Sobolev space on a given microdomain is not needed for what follows in this paper, but can be found in the literature, including the references quoted in this Remark 7.

*4.4. The Global Topological Structure of Nonlocal Electromagnetic Material Domains: First Look*

In light of the analysis above, each microdomain $V_{\mathbf{r}}$ induces an infinite-dimensional linear function space (Sobolev space) $\mathcal{F}(V_{\mathbf{r}})$ indexed by the position $\mathbf{r} \in D$, with the corresponding topology being essentially determined by the geometry of $V_{\mathbf{r}}$. On the other hand, this latter geometry is obtained from the *physics* of field–matter interaction in nonlocal media. Consequently, the physical content of nonlocal materials is encoded at the level of the topological microstructure encapsulated by the following formal scheme:

$$D \ni \mathbf{r} \xrightarrow[\text{Physics-Based Nonlocality}]{\text{Physical Data}} V_{\mathbf{r}} \xrightarrow[\text{Sobolev Space}]{\text{Mathematical Data}} \mathcal{F}(V_{\mathbf{r}}) \qquad (28)$$

Let us first identify the main relevant collections of subsets needed in order to understand the formal set-theoretic structure of the problem. We begin by

$$\mathcal{V}(D) := \{ V_{\mathbf{r}} \subset D \,\big|\, \mathbf{r} \in D, V_{\mathbf{r}} \text{ is open}\}, \qquad (29)$$

as the class of physics-based nonlocal microdomains (Definition 1). On the other hand, it is also possible to introduce the useful construction

$$\mathcal{G}[\mathcal{V}(D)] := \{ \mathcal{F}(V_{\mathbf{r}}) \,\big|\, \mathbf{r} \in D, \mathcal{F}(V_{\mathbf{r}}) \text{ is a Sobolev function space}\}, \qquad (30)$$

as a convenient class into which we collect all the function spaces of excitation fields on each nonlocality microdomain $V_{\mathbf{r}}$ as spanned by the position index $\mathbf{r} \in D$ (see Remark 7 for the construction of each such function space.) It follows then that (28) can be neatly captured by the ordered triplet

$$D \times \mathcal{V}(D) \times \mathcal{G}[\mathcal{V}(D)]. \qquad (31)$$

We wish now to unpack this compact structure in a careful, step-by-step manner, proceeding as follows:

1. Each open domain in $D \subseteq \mathbb{R}^3$ will by assigned a distribution $\mathcal{V}(D)$ of open sets $V_{\mathbf{r}}$, i.e., the physics-based nonlocality microdomains topology defined in Section 4.2, see in particular Definition 1 and Remark 5. Physically, it expresses the fine microtopological structure of nonlocal continua, e.g., electromagnetic material nonlocality.
2. The structure $\mathcal{V}(D)$ is solely determined by the physics of field–matter interaction. A concrete example explicitly illustrating how the detailed physical content of the underlying process contributes to the construction of $\mathcal{V}(D)$ will be given in Section 7.
3. We further emphasize that the various sets $V_{\mathbf{r}} \in \mathcal{V}(D)$ constitute an open cover of $D$, that is, we have

$$D = \bigcup_{\mathbf{r} \in D} V_{\mathbf{r}}. \qquad (32)$$

   In this way, the model can accommodate excitation fields $\mathbf{F}(\mathbf{r})$ applied at every point in $\mathbf{r} \in D$.
4. The decomposition of the material domain $D$ into smaller building blocks exemplified by (32) is fundamental for computational topological models of nonlocal MTMs. For example, in Section 7 we will exploit this expansion in order to construct a topological coarse-grained model for inhomogeneous nonlocal semiconductor metamaterials.
5. Finally, the topology $\mathcal{V}(D)$ induces the "function superspace" $\mathcal{G}[\mathcal{V}(D)]$ (30) defined as a class of function spaces $\mathcal{F}(V_{\mathbf{r}}), \mathbf{r} \in D,$, where each vector field acts on one microdomain element $V_{\mathbf{r}}$ chosen from the topology $\mathcal{V}(D)$.

**Remark 8 (Topology, physics, and multiple scales).** It is interesting to observe how, within the framework proposed above, some sort of delicate constructive "division of labor" is seen to emerge into the picture, where a fruitful interaction between physics and mathematics generates the various required multiscale topological microstructures characteristic of nonlocality in continuum field theories. This is also the source of some potential difficulties hidden in the formal set-theoretic structure (31). Indeed, we will next try to smooth out the differences between the two main substructures $\mathcal{V}(D)$, which is principally controlled by physics, on one side, and $\mathcal{G}[\mathcal{V}(D)]$, which is dominated by purely mathematical considerations. One way to achieve a resolution of this philosophical tension between the physical and mathematical is by developing the entire theory of the set-theoretic structure (31) in a form that can encode all of its main substructures within a single, rich enough "meta-structure": the Banach vector bundle *superspace* (see Section 5 for the detailed construction).

As can be seen from Remark 8, there is indeed some strong motivation to search for alternative formulations of physical theory in complex and rich systems such as nonlocal material continua, where there exists multiple spatial topological scales. It will be seen that the superspace theory appears to provide some form of rare direct and transparent unity between physics and topology in this regard. In order to reach there, gradual, step-by-step changes in the conventional formulation of continuum field theory will be introduced. We now begin to look into such a reformulation, starting with a straightforward one.

*4.5. A Reformulation of the Nonlocal Continuum Response Function*

It is now possible to provisionally construct the nonlocal continuum response function by working on the fundamental topological domain structure (31) instead of the global domain $D$, the later being the favored arena of conventional continuum field theory that we would like to ultimately move beyond. Again, for concrete expressions, the special case of electromagnetic theory will be presupposed but it should always be kept in mind that the mathematical structure of the theory is quite general and applies to all nonlocal continuum field theories governed by an abstract material response function model, such as the one discussed in Section 3.

We start by noting that the response field $\mathbf{R}(\mathbf{r})$ can be re-expressed by the map

$$\mathbf{R} : D \times \mathcal{V}(D) \to \mathbb{C}^3, \tag{33}$$

where the codomain is taken to be $\mathbb{C}^3$ because the electric or magnetic response functions $\mathbf{D}$ or $\mathbf{B}$, respectively, are complex vector fields in the frequency domain.[11] The value of the EM nonlocal response field due to excitation field $\mathbf{F}(\mathbf{r})$ applied at a microdomain $V_{\mathbf{r}}$ can be computed by means of

$$\mathbf{R}(\mathbf{r};\omega) = \int_{V_{\mathbf{r}}} \mathrm{d}^3 r' \, \overline{\mathbf{K}}(\mathbf{r},\mathbf{r}';\omega) \cdot \mathbf{F}(\mathbf{r}';\omega). \tag{34}$$

Although (34) may appear at first sight to be only slightly different from (11), the underlying difference between the two formulas is significant. In essence, the construction of the EM response field $\mathbf{R}(\mathbf{r})$ via the map (33) amounts to *topological localization* of electromagnetic nonlocality, since in the latter case, the EM response function $\overline{\mathbf{K}}(\mathbf{r},\mathbf{r}';\omega)$ is no longer allowed to extend globally onto "large and complicated material domains." Indeed, with the recipe (34) only the response to "small"–or more rigorously *topologically local*[12]—domains, namely the microdomains $V_{\mathbf{r}}$, is admitted. On the other hand, in order to find the response field $\mathbf{R}(\mathbf{r})$ everywhere in $D$, one needs to use sophisticated topological techniques to extend the response from one point to another until it covers the entirety of $D$. This local-to-global extension application of differential topology is discussed in detail in Section 5.3 and again briefly in Appendix A.11.

In such a manner, it becomes possible to provide an alternative, more detailed explication of the behavior of the medium at topological interfaces (boundary conditions in

nonlocal metamaterials are treated–provisionally–in Section 8) and also explore the effect of the topology of the bulk medium itself on the allowable response functions and the production of non-trivial edge state, with obvious applications to emerging areas such as nonlocal metamaterials.[13]

## 5. The Fiber Bundle Superspace Formalism in the Field Theory of Generic Nonlocal Continua

Here, an outline of the direct construction of a fiber (Banach) bundle over an entire (global) nonlocal generic material domain is given, where our purpose is to attach to every point $\mathbf{r} \in U_i$ a *fiber* $\mathcal{F}_i$, actually a vector space in our case. The contents of this section are the most technically advanced in this paper. Readers interested in applications may skim through Sections 5.1 and 5.2, skip Section 5.3, then move directly to Section 6 for a general summary of the fiber bundle algorithm. Concrete computational models are outlined in Section 7 using a practical nonlocal model, while additional remarks and discussions about current and future uses of the theory are provided in Appendixes A.3 and A.11. However, even readers not fully familiar with the differential manifold theory will benefit from reading the present technical section, because we strive to illustrate the physical intuition behind the various mathematical computations and steps therein.

### 5.1. Preparatory Step: Promoting the Material Domain D to a Manifold $\mathcal{D}$

In order to investigate in depth the fundamental physico-mathematical constraints imposed on nonlocal continua, the domain $D$, which we have working with so far as the main total spatial space of the material, should be upgraded in complexity to the higher level of a *differential manifold*, the latter which posses a quite rich and sophisticated structure that allows performing calculus and geometrical reasoning simultaneously [26,57,62,63,68]. There are several reasons why this is highly desirable:

1. It provides a natural and obvious generalization of the basic structure (31) from the mathematical perspective.
2. Engineers often need to insert metamaterials into specific device settings, hence the shape of the material becomes highly restricted. It is therefore important to develop efficient tools to deal with variations of geometric and topological degrees of freedom and how they could possibly impact the design process.
3. Applied scientists and engineers are often interested in deriving fundamental limitation on metamaterials, e.g., what are the ultimate allowable response–excitation relations or constitutive response functions possible given this material domain topology?
4. Sophisticated full-wave electromagnetic numerical solvers prefer working with local coordinates in order to handle complicated shapes, even if a global coordinate system is sometimes available, making the deployment of the three-manifold structures for describing the material domain $D$ useful.
5. In topological photonics and materials [11], most applications seem to focus on lower-dimensional states of matter like those associated with quantum Hall effects and edge states (surface waves).[14] There, new phenomena appear at material structures where the base space (material domain $D$) is a two-surface, which is best described mathematically as a differential two-manifold.

For all these reasons, it is desirable to strive to furnish the domain $D$ with the most general and flexible mathematical apparatuses available to us, which, in this case, amounts to equipping the material/metamaterial spatial domain with a *smooth manifold structure*.

We quickly illustrate how this can be accomplished. If we denote by $\mathcal{D}$ a three-manifold (three-dimensional smooth manifold), then, since $D \subset \mathbb{R}^3$, there is a natural differential structure defined on $D$, inherited from the ambient three-dimensional Euclidean space itself. (Throughout this paper, such differential three-manifold structure will be presupposed as the de facto space for the *total*, i.e., largest, material space.)

Following the standard theory of smooth manifolds, let

$$(U_i, \phi_i) \tag{35}$$

be a countable collection of charts (an atlas), labeled by

$$i \in I \subset \mathbb{N}, \tag{36}$$

where $I$ is an index set. Together, the devices (35) and (36) can equip $\mathcal{D} \subset \mathbb{R}^3$ with a differential three-manifold structure. For simplicity, we will refer to the points of the manifold $\mathcal{D}$ by $\mathbf{r}$, i.e., using the language of the global (ambient) Euclidean space $\mathbb{R}^3$. Symbolically, by adding a differential manifold structure, we effected the transformation

$$D \xrightarrow[\text{Introduce a differential atlas}]{\text{Insert a smooth manifold structure}} \mathcal{D} \tag{37}$$

This well-known construction [26,57,63] constitutes the differential atlas on $\mathcal{D}$, which will be used in what follows.

*5.2. Attaching Fibers to Generic Points in the Nonlocal Material Manifold D*

Our current goal is to attach a *vector fiber* (a linear function space in this case) at every point $\mathbf{r} \in \mathcal{D}$, namely the function space $\mathcal{F}(V_{\mathbf{r}})$ introduced in Section 4.3. It turns out that accomplishing this requires finding suitable "compatibility laws" dictating how coordinates change when two intersecting charts $U_i$ and $U_j$ interact with each other, which is typical in such types of constructions [26]. In particular, we will need to later find the law of mutual transformation of vectors in the fibers $\mathcal{F}(V_{\phi_i(\mathbf{r})})$ and $\mathcal{F}(V_{\phi_j(\mathbf{r})})$. Here, the expression

$$\mathcal{F}\left(V_{\phi_i(\mathbf{r})}\right) \tag{38}$$

means the fiber space attached to the point whose coordinates are $\phi_i(\mathbf{r})$, i.e., the *function space* where all functions are expressed in terms of the language of the $i$th chart $(U_i, \phi_i(\mathbf{r}))$.

In this connection, the major technical problem facing us is a mathematical one induced by the physics of the situation. We first isolate and describe the main problem by the following brief technical resume:

*Since the differential structure associated with charts*

$$\{(U_i, \phi_i(\boldsymbol{r})), i \in I, \}$$

*can be fixed by essentially mathematical considerations alone, while the collection of microdomains*

$$\mathcal{V}(\mathcal{D}) = \{V_{\boldsymbol{r}}, \boldsymbol{r} \in \mathcal{D}\}$$

*is solely determined by the physics of electromagnetic nonlocality (See Remark 3 and Section 4), there is no direct and simple way to determine and express the vector transformation*

$$\mathcal{F}(V_{\phi_i(\boldsymbol{r})}) \longrightarrow \mathcal{F}(V_{\phi_j(\boldsymbol{r})}),$$

*because several different coordinate patches other than $U_i$ and $U_j$, belonging to the differential three-manifold $\mathcal{D}$ atlas, might be involved in geometrically building up the microdomain $V_{\boldsymbol{r}}$.*

The above technical problem will be solved in Section 5.3 by using the technique of *partition of unity* borrowed from differential topology [26,57,62]. It will allow us to split up each full microdomain $V_{\mathbf{r}}$ into several suitable sub-microdomains (details below), which can be later joined up together in order to give back the original EM nonlocality microdomain $V_{\mathbf{r}}$.

For now, we start by recalling that the microdomain structure represented by the collection $\mathcal{V}(\mathcal{D}) = \{V_{\mathbf{r}}, \mathbf{r} \in \mathcal{D}\}$ is an *open cover* of the manifold $\mathcal{D}$. Therefore, and since the

material domain manifold $\mathcal{D}$ possesses a *countable* topological base [59], it contains a locally finite open cover subordinated to $\mathcal{V}(\mathcal{D})$ [26,57].[15] This implies that an atlas $(U_i, \phi_i), i \in I$, with diffeomorphisms

$$\phi_i : U_i \to \mathbb{R}^3, \tag{39}$$

describing the differential structure of the manifold $\mathcal{D}$ exists such that the elements $\{U_i, i \in I\}$ constitute the above mentioned locally finite subcover *subordinated* to the microdomains collection $\mathcal{V}(\mathcal{D})$. Moreover, the images $\phi_i(U_i)$ are *open balls* centered around 0 in $\mathbb{R}^3$ with finite radius $a > 0$ (henceforth, such balls will be denoted by $B_a$) [26].

In this way, the physics-based open cover set $\mathcal{V}(\mathcal{D})$ provides a first step toward the construction of a complete *topological* description of the *physics-based* nonlocal microdomain structure. The reason is that the coordinate patches $(U_i, \phi_i), i \in I$, are *subordinated* to the microdomains $\{V_{\mathbf{r}}, \mathbf{r} \in \mathcal{D}\}$ [26].

It is also known that there exists a *partition of unity* associated with the $\mathcal{D}$-atlas $(U_i, \phi_i), i \in I$, constructed above summarized by the following lemma [26,57,62,63,68]:

**Lemma 1** (**Partition of Unity**). There is a collection of functions

$$\psi_i : U_i \subset \mathcal{D} \to \mathbb{R} \tag{40}$$

satisfying the following requirements:

1. $\psi_i(\mathbf{r}) \geq 0$ and each function is $C^p, p \geq 1$.[16]
2. The *support* of $\psi_i(\mathbf{r})$, denoted by supp $\psi_i$, is contained within $U_i$, that is, the condition

$$\operatorname{supp} \psi_i \subset U_i, \tag{41}$$

holds. Recall that the support is defined as the (topological) closure of the set

$$\{\mathbf{r} \in \mathcal{D} | \psi_i(\mathbf{r}) \neq 0\}. \tag{42}$$

See for example [30,68,69].
3. Since the open cover $U_i, i \in I$, is *locally finite*, at each point $\mathbf{r} \in \mathcal{D}$, only a *finite* number of $U_i$ will intersect $\mathbf{r}$.
4. Let the set of indices of those intersecting $U_i s$ be $I_{\mathbf{r}}$. Then we require that

$$\sum_{i \in I_{\mathbf{r}}} \psi_i(\mathbf{r}) = 1, \tag{43}$$

where the sum is always convergent because the set $I_{\mathbf{r}}$ is finite.

**Remark 9.** It can be shown that the sets

$$\phi_i^{-1}(B_{a/3}), i \in I, \tag{44}$$

where $B_{a/3}$ is a standard Euclidean ball centered at the origin with radius $a/3$, already cover $\mathcal{D}$ [57]. Moreover, the closure

$$\operatorname{cl}\{\phi_i^{-1}(B_{a/3})\} \tag{45}$$

may be taken to constitute the support of $\psi_i(\mathbf{r})$, while [26,57,70]

$$\mathbf{r} \notin \operatorname{supp}\{\psi_i(\mathbf{r})\} \implies \psi_i(\mathbf{r}) = 0. \tag{46}$$

The partition of unity functions $\psi_i$ can be computationally constructed using standard methods, most prominently the bump functions, see [57,71] for details.

The motivation behind the deployment of the partition of unity technique and how it immediately arises in connection with our fundamental EM nonlocal structure should now be clear. We have found that the following three-step process is natural:

1. Initially, the *physics*-based collection of sets

$$\mathcal{V}(\mathcal{D}) = \{V_{\mathbf{r}}, \mathbf{r} \in \mathcal{D}\},$$

for example, the EM nonlocal microdomain structure based on each point **r** in the nonlocal metamaterial $\mathcal{D}$, is obtained using a suitable physical microscopic theory or some other procedure.[17]

2. Introduce a differential atlas

$$(U_i, \phi_i(\mathbf{r})), \ i \in I,$$

on the smooth manifold $\mathcal{D}$ subordinated to $\mathcal{V}(\mathcal{D})$ and representing the nonlocal material domain under consideration.

3. Finally, the same atlas is linked to a set of functions $\psi_i(\mathbf{r})$ (partition of unity) that can be recruited as "topological bases" in order to expand any differentiable field excitation function into sum of individual sub-fields defined on open subsets of the material domain $\mathcal{D}$ (see Section 5.3).

The three-step process outlined above is summarized in Figure 3, illustrating how to progressively construct micro-coordinate systems allowing one to see through increasingly smaller spatial scales in the fundamental characterization of electromagnetic material nonlocality.

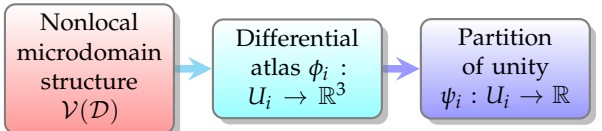

**Figure 3.** The three-step process of constructing micro-coordinate representations of material nonlocality starting from the nonlocal microdomain set and ending with the partition of unity on the material continuum's superspace.

The key idea to be developed next is that both the base manifold $\mathcal{D}$ and the nonlocal physics-based microdomains $V_{\mathbf{r}}$ are described *locally* (in the topological sense[18]) by the *same* collection of charts, namely $(U_i, \phi_i(\mathbf{r})), i \in I$. This will permit us to construct a direct unified description of *both* the base manifold $\mathcal{D}$ *and* its fibers, i.e., the linear topological function spaces $\mathcal{F}(V_{\mathbf{r}})$, the latter being the model of the physical electromagnetic fields exciting the nonlocal material $\mathcal{D}$.

The construction of a fiber bundle superspace for nonlocal electromagnetic materials will be completed in two steps:

- Step I: Construct a tailored fiber bundle based on the partition of unity charts $(U_i, \phi_i(\mathbf{r}))$ introduced above.
- Step II: the original physical structure (31) is recovered by gluing together various sub-microdomain $U_i \subseteq V_{\mathbf{r}}$ of each EM nonlocal microdomain $V_{\mathbf{r}}$.

We start with Step I, while we leave the more complicated Step II to Section 5.3.

Consider the $(U_i, \phi_i(\mathbf{r})), i \in I$, as our atlas on the three-manifold $\mathcal{D}$ introduced in Section 5.1. At each point $\mathbf{r} \in U_i$, we attach a linear topological space $\mathcal{F}(U_i)$ defined as the Sobolev space

$$W^{p,2}(U_i), \ p \geq 1, \tag{47}$$

of functions on the open set $U_i$, i.e., we write

$$\forall i \in I, \ \mathcal{F}(U_i) := \{\psi_i(\mathbf{r})\mathbf{F}(\mathbf{r}), \mathbf{r} \in U_i, \text{ is in the Sobolev space } W^{p,2}(U_i)\}, \tag{48}$$

where $\mathbf{F}(\mathbf{r})$ is a suitable $C^{p,2}$ vector field.

**Remark 10 (Sobolev Spaces).** For the precise technical definition of the infinite-dimensional Sobolev function space $W^{p,2}(U_i)$, see [64,65]. Appendix A.5 provides some additional information on the literature. Section 4.3 gives a simplified intuitive definition of the physics-based function space $\mathcal{F}_i$, in particular see Remark 8. The intricate details of the theory of such Sobolev function spaces will not be needed for our immediate purposes in what follows (compare with Remark 11).

Physically, the multiplication of the global excitation field $\mathbf{F}(\mathbf{r})$ by $\psi_i(\mathbf{r})$ in constructions like (48) above and (50) below effectively "localizes" (in the topological sense) the field into a smaller compact subdomain, namely the support of the "topological localization basis function" $\psi_i(\mathbf{r})$ itself. Moreover, because the $C^p$-functions $\psi_i(\mathbf{r})$ have *compact* supports satisfying the inclusion restrictions

$$\mathrm{supp}\{\psi_i\} \subset U_i, \ \ i \in I, \tag{49}$$

it follows that $\mathcal{F}(U_i)$ is effectively a *local* Sobolev space on $U_i$ [66]. Alternatively, it is also possible to seek different constructions, such as the one captured by the following remark.

**Remark 11.** We may define a less complicated function space on $U_i$ using the following construction:

$$\forall i \in I, \ \ \mathcal{F}'(U_i) := \{\psi_i(\mathbf{r})\mathbf{F}(\mathbf{r}), \mathbf{r} \in U_i, \text{ is an element of a } C^p \text{sup-norm function space}\}, \tag{50}$$

where the $C^p$-sup-norm is defined by

$$||\psi_i(\mathbf{r})\mathbf{F}(\mathbf{r})|| := \mathrm{sup}_{\mathbf{r}\in\mathrm{supp}\{\psi_i\}}[\psi_i(\mathbf{r})\mathbf{F}(\mathbf{r})]. \tag{51}$$

In the case of $\mathcal{F}'(U_i)$, one may further consider only $C^p$-vector excitation fields $\mathbf{F}(\mathbf{r})$. A choice of which linear function space to work with depends on the particular application under consideration. In what follows, we further simplify our notation by writing $\mathcal{F}_i$ instead of $\mathcal{F}(U_i)$ whenever the partition of unity's differential atlas' coordinate patches $U_i$ are used.

*5.3. Direct Construction of Bundle Homomorphism as Generalization of Linear Operators in Electromagnetic Theory*

We now demonstrate how the material constitutive relations in conventional (local) continuum theory may be absorbed into a new structure, the *bundle homomorphism*, which is the most natural generalization of linear operators in local electromagnetism taking us into the enlarged stage of the generic nonlocal medium's superspace formalism. In the future, these bundle homomorphisms may be discretized using topological numerical methods, e.g., see [72]. In what follows, we focus on the rigorous exact construction using the technique of partition of unity, which allows computations going from local to global domains.[19]

5.3.1. The Basic Definition of the Nonlocal Material, (or Continuum, Metamaterial (MTM), etc.), Banach (Fiber) Bundle Superspace

The initial step in formally defining the proposed nonlocal MTM bundle superspace is the following disjoint union construction:

**Definition 2 (Preliminary Definition of the Bundle Superspace).** Let the material continuum's superspace be denoted by $\mathcal{M}$, which is also called the *total bundle space*. We define this space as the disjoint union of all spaces $\mathcal{F}_i$ of the form:

$$\mathcal{M} := \{(\mathbf{r}, \mathcal{F}_i) | \forall i \in I, \mathbf{r} \in U_i \subset \mathcal{D}\}. \tag{52}$$

Associated with $\mathcal{M}$ is a surjective map

$$p : \mathcal{M} \to \mathcal{D}, \tag{53}$$

which "projects" the fiber onto its corresponding point in the base manifold $\mathcal{D}$, i.e., $p((\mathbf{r}, \mathcal{F})) := \mathbf{r}$.

**Remark 12 (Other constructions of bundle spaces).** In mainstream literature, the fiber bundle concept is often approached in a manner slightly differently from that of Definition 2. Indeed, the *fiber* of $\mathcal{M}$ at $\mathbf{r} \in \mathcal{D}$ is *defined* as the set $p^{-1}(\mathbf{r})$, but provided the map $p$ is already given as part of the bundle's initial data. However, in this paper, we *construct* the bundle data starting with the physics-based topological structure (31).

**Remark 13 (Fiber Projections and Local Isomorphisms).** The map $p$ is called the *projection* of the vector bundle $\mathcal{M}$ onto its *base space* $\mathcal{D}$. Moreover, from now on, we will also use the notation $\mathcal{F}_{\mathbf{r}}$ to denote the fiber $p^{-1}(\mathbf{r})$. By construction, it should be clear that

$$\forall i \in I : \quad p^{-1}(\mathbf{r}) = \mathcal{F}_i \iff \mathbf{r} \in U_i. \tag{54}$$

From the topological viewpoint, the material continuum superspace $\mathcal{M}$ manifests itself *locally* as a product space in the form

$$U_i \times \mathcal{F}_i. \tag{55}$$

In other words, the map $p$ should behave locally as a conventional projection operator; i.e., in a local domain $U_i$, the material's total bundle space $\mathcal{M}$ is isomorphic to $U_i \times \mathcal{F}_i$, and $p(U_i \times \mathcal{F}_i)$ should be isomorphic to $U_i$. Symbolically, we have:

$$\mathcal{M} \cong_{\text{locally}} U_i \times \mathcal{F}_i, \quad p(U_i \times \mathcal{F}_i) \cong_{\text{locally}} U_i, \tag{56}$$

for all $i \in I$, and where $\cong_{\text{locally}}$ means local topological (in this case also smooth) isomorphism.[20]

In order to complete the specification of the nonlocal material continuum superspace, we next construct the linear function space $X_i$ defined by

$$\forall i \in I, \ X_i := \left\{ \psi_i \left[ \phi_i^{-1}(\overline{x}) \right] \mathbf{F} \left[ \phi_i^{-1}(\overline{x}) \right] \text{ is an element of a Sobolev space for all } \overline{x} \in B_a \right\}, \tag{57}$$

which is the Sobolev space of $W^{p,2}(B_a)$ functions on the Euclidean 3-ball $B_a$. Here, each function is defined with respect to the *local* coordinates

$$\overline{x} := \phi^{-1}(\mathbf{r}), \ \mathbf{r} \in U_i. \tag{58}$$

In fact, it should be straightforward to deduce from the above that there exists maps

$$\tau_i : p^{-1}(U_i) \to U_i \times X_i, \tag{59}$$

for all $i \in I$, that are *isomorphisms* (*diffeomorphism* in our case), where such diffeomorphism may be expressed by

$$\forall i \in I : \ p^{-1}(U_i) \cong U_i \times \mathcal{F}_i. \tag{60}$$

We also add that the fact of (59) actually playing the role of such an isomorphism would naturally follow from the respective definitions of the spaces $\mathcal{F}_i$ and $X_i$, as specified by (48) and (57), and from the proposition that each $\phi_i$ is a diffeomorphism from $U_i$ into $\mathbb{R}^3$ (or,

equivalently, to the unit 3-ball $B_a$ with radius $a$ instead of $\mathbb{R}^3$.) Furthermore, note that by construction the diffeomorphism $\tau_i$ satisfies

$$\text{proj}_1 \circ \tau_i = p, \tag{61}$$

where $\text{proj}_1$ is the standard projection map defined by $\text{proj}_1(x, y) := x$. Finally, if we restrict $\tau_i$ to $p^{-1}(\mathbf{r})$, the resulting map

$$\tau_i|_{p^{-1}(\mathbf{r})} : p^{-1}(\mathbf{r}) \to \{\mathbf{r}\} \times X_i \tag{62}$$

is a (linear) topological vector space isomorphism from $\mathcal{F}_{\mathbf{r}}$ to $X_i$; namely, we have

$$\forall i \in I, \ \mathbf{r} \in U_i : \quad \mathcal{F}_{\mathbf{r}} \cong X_i. \tag{63}$$

**Remark 14.** The charts $(U_i, \tau_i)$ are called *trivialization covering* of the vector bundle $\mathcal{M}$. They provide a coordinate representation of local patches of the vector bundle. (The global topology of the bundle, however, is rarely trivial [62].) Since here all maps are $C^p$ smooth, $\tau_i$ are also called *smooth* trivialization maps. The complete derivations of the diffeomorphism (60) and the topological vector space isomorphism (63) are straightforward, but lengthy, and the full proofs are omitted.

Consider now two patches $U_i$ and $U_j$ with $U_i \cap U_j \neq \emptyset$. By restricting $\tau_i$ and $\tau_j$ to $U_i \cap U_j$, two diffeomorphisms

$$\begin{aligned}
\tau_i : p^{-1}(U_i \cap U_j) &\to (U_i \cap U_j) \times X_i, \\
\tau_j : p^{-1}(U_i \cap U_j) &\to (U_i \cap U_j) \times X_j,
\end{aligned} \tag{64}$$

are obtained, which together imply in turn that

$$(U_i \cap U_j) \times X_i \cong (U_i \cap U_j) \times X_j, \tag{65}$$

or, equivalently, the following expected Banach space isomorphism:

$$X_i \cong X_j, \tag{66}$$

In particular, it can be shown that the composition map

$$\tau_j \circ \tau_i^{-1} : (U_i \cap U_j) \times X_i \to (U_i \cap U_j) \times X_j \tag{67}$$

possesses the simple form

$$\tau_j \circ \tau_i^{-1}(\mathbf{r}, \mathbf{F}) = (\mathbf{r}, g(\mathbf{r})\mathbf{F}), \tag{68}$$

with the following formal structure:

$$\forall \mathbf{r} \in U_i \cap U_j, \ \mathbf{F} \in X_i, \ \ \exists g \in \mathrm{L}(X_i, X_j), \tag{69}$$

where the abstract vector linear space

$$\mathrm{L}(X_i, X_j) \tag{70}$$

is defined as the space of all linear operators [26]

$$g : X_i \to X_j \tag{71}$$

on Banach vector spaces. In particular, $g(\mathbf{r})$ is a $C^p$-Banach space isomorphism.

**Remark 15.** In the mathematical literature, the smooth maps $\tau_j \circ \tau_i^{-1}$ are called the *vector bundle transition maps*. They are essential technical tools for computing global data by starting from local data then gluing them together. For example, they will be used in Sections 5.3.2 and 5.3.3 as part of the toolbox needed in the process of generalizing local information into global domains.

We have now succeeded in directly constructing a specialized smooth Banach vector bundle $(\mathcal{M}, \mathcal{D}, \tau, p)$ consisting of the nonlocal material continuum's total fiber bundle space $\mathcal{M}$, the material domain's base three-manifold $\mathcal{D}$, a set of smooth trivialization charts $\tau_i, i \in I$, and a projection map $p$. The base manifolds $\mathcal{D}$ itself is described by a differential atlas $(U_i, \phi_i)$, also associated with the partition of unity $(U_i, \psi_i), i \in I$ as per our discussion in Section 5.2 above. This incredible increase in the complexity of the mathematical space of nonlocal continuum field theory; that is, the transition from spacetime (or space–frequency) as the configuration space to a a larger superspace, here the fiber bundle space (which might be time- or frequency-dependent), is a direct expression of the very significant complexity and richness of the *physics* of nonlocal field theory in general.

As will be seen in the next Section 5.3.2, it is possible to demonstrate yet another remarkable departure from conventional theory where the concept of *linear operator*, as such, a fundamental structural object in the mathematical and computational physics of local continuum field theories [65], is found to be generalizable to the concept of *homomorphism*, which is essentially topological in nature.

5.3.2. The Nonlocal Material Continuum Fiber Bundle Homomorphism

At this point, we need to describe how the *evaluation* process of the response field (33) may be formulated within the new enlarged framework of the fibered superspace $\mathcal{M}$. The most obvious method is to introduce a *new* vector bundle with the base space being the same base space $\mathcal{D}$, but with the fibers now taken as the complex Hilbert space $\mathbb{C}^3$. This is a well-known vector bundle, which we denote by $\mathcal{R}$, and dub the *range vector bundle*. Formally, the structure of this vector bundle is expressed by the ordered quadruple $(\mathcal{R}, \mathcal{D}, \tau', p')$, where $\tau'$ and $p'$ are the range bundle $\mathcal{R}$'s smooth trivialization and projection maps, respectively. On the other hand, the *source vector bundle* is taken as $\mathcal{M}$.

As a preparation for introducing the concept of the nonlocal continuum homomorphism, let us recap and comment on the overall physical process of exciting a material nonlocal continuous domain $\mathcal{D}$ as follows:

1. The continuum itself is mathematically modeled as a Banach bundle superspace $\mathcal{M}$ instead of its conventional differential manifold representation $\mathcal{D}$. The response of the medium is to be sought at some point $\mathbf{r} \in \mathcal{D}$.
2. The bundle superspace $\mathcal{M}$ encompasses an *additional* structure compared to $\mathcal{D}$, namely a distinct copy of a linear function space attached at each point $\mathbf{r} \in \mathcal{D}$. This is nothing but the fiber $p^{-1}(\mathbf{r})$, which is a Banach space of functions defined on the region $U_i$. This function space can be intuitively understood as a rigorous and exact model of the excitation field $\mathbf{F}(\mathbf{r})$ when the latter is restricted to (topologically localized at) the physics-based nonlocal domain $U_i$.
3. It should be noted that in local continuum field theory, e.g., conventional electromagnetism in normal temporally dispersive media, each one of the subdomains $U_i, i \in I$, is essentially *one* point $\mathbf{r} \in \mathcal{D}$. Therefore, in the case of local continua, the excitation field $\mathbf{F}$ is there found to be preferably defined as acting on the conventional space $\mathcal{D}$ instead of being a section of a Banach bundle superspace $\mathcal{M}$.
4. A vector bundle homomorphism (to be formalized in Definition 3) will map one element of this fiber function space, namely, the particular excitation field $\mathbf{F}(\mathbf{r}), \mathbf{r} \in U_i$, to its value in the range vector bundle $\mathcal{R}$. For the case of electromagnetic field theory, the latter may be taken as a vector space fiber isomorphic to $\mathbb{C}^3$ with a copy of this fiber attached to each $\mathbf{r} \in \mathcal{D}$.

We turn now to a precise definition for convenient maps between bundle superspaces. Formally, we may directly use the standard concept of homomorphism in fiber bundle theory, adapted to our purposes in the following manner [57,68]:

**Definition 3** (**Bundle homomorphism**). A smooth bundle homomorphism over a common base space $\mathcal{D}$ shared between the two vector bundles $\mathcal{M}$ and $\mathcal{R}$ is defined by the (smooth) map:

$$\mathcal{L} : \mathcal{M} \to \mathcal{R} \tag{72}$$

satisfying $p' \circ \mathcal{L} = p$. Moreover, the restriction of $\mathcal{L}$ to each fiber $p^{-1}(\mathbf{r})$ induces a linear operator on the corresponding vector space of that fiber. In effect, the following diagram

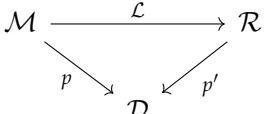

is commutative.

**Remark 16.** Because the nonlocal material continuum's superspace $\mathcal{M}$ and its range fiber bundle $\mathcal{R}$ both share an identical base manifold $\mathcal{D}$, the action of the homomorphism $\mathcal{L}$ as a bundle map is effectively reduced to how it interacts with each fiber $p^{-1}(\mathbf{r})$ by acting on the latter as a standard vector space linear operator. Therefore, a large portion of the conventional linear algebra and computational methods extensively deployed in the mathematical and numerical apparatus of local continuum field theory, such as nonlinear functional analysis [65], Hilbert space methods [64], and the Finite Element Method [75], may be reused as "sub-algorithms" within the larger, more general formalism of nonlocal continuum field theory proposed in this paper.

Now, since the Banach space $X_i$ is isomorphic to $p^{-1}(\mathbf{r})$, we may assemble the homomorphism $\mathcal{L}$ by specifying its *local* expression in each topological subdomain $U_i \subset \mathcal{D}$ of the open cover $\{U_i, i \in I\}$. In particular, we define the local action using the source and range bundles' trivialization maps $\tau_i$ and $\tau_i'$ by the intuitively obvious formula:

$$\tau_i' \circ \mathcal{L}_\omega \circ \tau_i^{-1} : U_i \times X_i \to U_i \times \mathbb{C}^3, \tag{73}$$

with

$$\tau_i' \circ \mathcal{L}_\omega \circ \tau_i^{-1} := (\mathbf{r}, \mathcal{L}_{i,\omega}\mathbf{F}), \quad \mathbf{F} \in X_i, \tag{74}$$

where

$$\mathcal{L}_{i,\omega} : X_i \to \mathbb{C}^3 \tag{75}$$

is the linear operator defined by

$$\mathcal{L}_{i,\omega}(*) = \int_{U_i} \mathrm{d}^3 r' \, \overline{\mathbf{K}}(\mathbf{r}, \mathbf{r}'; \omega) \cdot (*), \tag{76}$$

in which '*' stands for an element of the smooth Banach function space $X_i$.

Therefore, within the frequency domain formulation of this paper, the operator $\mathcal{L}$ will leave every point in the base space $\mathcal{D}$ unchanged while mapping each smooth function on $U_i$ (component of the total electromagnetic excitation field, see below) into its complex vector value in $\mathbb{C}^3$ at $\mathbf{r} \in U_i$. Physically, $\mathcal{L}_i$ models a (topologically) localized "piece" of the global electromagnetic material operator mapping excitation fields $\mathbf{F}(\mathbf{r})$ to response fields $\mathbf{R}(\mathbf{r})$, where the entire physics here is restricted to the physics-based nonlocal subdomain $U_i$. The global operator itself is assembled by gluing together these small pieces using the partition of unity technique, as we endeavor to show next.

### 5.3.3. Computing Global Data Starting from Local Data

The final step is tying up together the fundamental source Banach bundle superspace $\mathcal{M}$, range bundle $\mathcal{R}$ and the nonlocal microdomain physics space (31). The essential ingredients of the *physics* of nonlocal field–matter interaction are encoded in the geometrical construction of the collection of microdomains $\mathcal{V}(\mathcal{D}) = \{V_{\mathbf{r}}, \mathbf{r} \in D\}$, and the excitation fields $\mathbf{F}(\mathbf{r})$ defined on them, i.e., the sets $\mathcal{V}(\mathcal{D})$ *and* the (excitation) function spaces $\mathcal{G}(\mathcal{D})$ combined together in one space, the superspace $\mathcal{M}$.

So far, the vector bundle homomorphism $\mathcal{L}$ introduced above (Definition 3) can handle excitation fields supported on the open sets $U_i, i \in I$. However, the latter sets are *mathematical* fundamental building blocks, or "set-theoretic atoms", deployed in order to formally construct the source vector bundle superspace $\mathcal{M}$. The question that will be addressed presently is the following one:

*How can we extend the description of the nonlocal continuum's response operators starting from excitation fields defined locally to excitation fields applied on the entire physical cluster of nonlocal microdomains $\{V_{\boldsymbol{r}}, \, \boldsymbol{r} \in \mathcal{D}\}$?*

As mentioned before, it is the partition of unity $(U_i, \psi_i), i \in I$, what will make this expansion of the topological formulation technically feasible.

To see this, let us consider an electromagnetic field $\mathbf{F}(\mathbf{r})$ interacting with a nonlocal medium extended over the manifold $\mathcal{D}$. Our goal is to compute the response field $\mathbf{R}(\mathbf{r})$; that is, at point $\mathbf{r}$. Let us recall what the fundamental idea of EM nonlocality is: in order to compute the nonlocal material continuum's response at one point $\mathbf{r}$, one must know the excitation field in the *entirety* of an open set $V_{\mathbf{r}}$. This set $V_{\mathbf{r}}$ is one of these nonlocal microdomains composing $\mathcal{D}$ as per (29). Moreover, such $V_{\mathbf{r}}$ is also a topological neighborhood of its continuous index point $\mathbf{r} \in \mathcal{D}$ (cf. Section 4.2). However, in general this microdomain will change depending on the position $\mathbf{r}$. The goal now is to find $\mathbf{R}(\mathbf{r})$ using the vector bundle map $\mathcal{L}$ defined by (72) starting from the data:

1. Region $V_{\mathbf{r}}$;
2. Vector field $\mathbf{F}(\mathbf{r})$ acting on $V_{\mathbf{r}}$.

To accomplish this, we exploit the properties of the partition of unity functions $\psi_i$ (Lemma 1) for expanding the excitation field $\mathbf{F}(\mathbf{r})$ over *all* patches $U_i$ covering $V_{\mathbf{r}}$, resulting in

$$\mathbf{F}(\mathbf{r}'; \omega) = \sum_{i \in I_{\mathbf{r}}} \psi_i(\mathbf{r}') \mathbf{F}_i(\mathbf{r}'; \omega), \tag{77}$$

where (43) was used. The truncated function $\mathbf{F}_i$ is equal to $\mathbf{F}(\mathbf{r})$ only if $\mathbf{r} \in U_i$ and zero elsewhere, i.e., we have

$$\mathbf{F}_i(\mathbf{r}'; \omega) := \begin{cases} \mathbf{F}(\mathbf{r}'; \omega), & \mathbf{r} \in U_i, \\ 0, & \mathbf{r} \notin U_i. \end{cases} \tag{78}$$

Recall that according to Lemma 1, the set $I_{\mathbf{r}}$ is defined as the collection of indices $i \in I$ of all $U_i$ having the point $\mathbf{r}$ in their common set intersection; by construction, this index set $I_{\mathbf{r}}$ is always finite.

The main idea behind our construction should now become clear: while each truncated sub-field $\mathbf{F}_i$ fails to be differentiable (it is not even continuous), the multiplication by $\psi_i(\mathbf{r})$ fixes this problem. In fact, each function

$$\psi_i(\mathbf{r}') \mathbf{F}_i(\mathbf{r}'; \omega) \tag{79}$$

is a *smooth* component of the total excitation field $\mathbf{F}$ with support *fully* contained *inside* the coordinate patch $U_i$; that is, we have

$$\mathrm{supp}\{\mathbf{F}_i\} \subset U_i. \tag{80}$$

Consequently, the vector bundle map constructed in (72) can be applied to each such component field. From (73)–(75) and (77), the following can be deduced:

$$\mathbf{R}(\mathbf{r};\omega) = \sum_{i\in I_{\mathbf{r}}} \mathcal{L}_{i,\omega}[\psi_i(\mathbf{r}')\mathbf{F}_i(\mathbf{r}';\omega)]. \tag{81}$$

Finally, using (76), we arrive at our main superspace map theorem:

**Theorem 1** (**global superspace bundle map**). For the fiber bundle superspace $\mathcal{M}$ of the nonlocal continuum whose differential manifold representation is $\mathcal{D}$ and the nonlocal continuum response function (tensor) $\overline{\mathbf{K}}$, the response and excitation fields $\mathbf{R}$ and $\mathbf{F}$ can be related to other other via the global bundle (superspace) map:

$$\boxed{\mathbf{R}(\mathbf{r};\omega) = \sum_{i\in I_{\mathbf{r}}} \int_{U_i} \mathrm{d}^3 r'\, \overline{\mathbf{K}}(\mathbf{r},\mathbf{r}';\omega) \cdot \psi_i(\mathbf{r}')\mathbf{F}_i(\mathbf{r}';\omega),} \tag{82}$$

where $\psi_i, i \in I$, are the partition of unity basis functions subordinated to the $\mathcal{D}$-atlas $(U_i,\phi_i), i \in I$.

Physically, Theorem 1 states that the nonlocal continuum's source bundle (superspace) $\mathcal{M}$, the range bundle $\mathcal{R}$, and the nonlocal response superspace map $\mathcal{L}$, together, supply the fundamental formal scaffold upon which the material domain's response to generic excitation field, when the latter field operates on arbitrary configurations of nonlocality microdomain, can be constructed. By aggregating all of those physics-based microdomains constituting the topological microstructure of nonlocal processes in material continua, the main field-theoretic structures of the medium may be couched, computed and reformulated in the richer language of this more general superspace framework belonging to the Banach fiber bundle $\mathcal{M}$ instead of the position space $\mathcal{D}$ of conventional spacetime extensively used in local field theories. At this stage of our formulation, the vector bundle formalism of nonlocality becomes essentially complete, where the connection between the purely mathematical fiber superspace and the physical microdomain structures is secured by Theorem 1, especially the Formula (82).

### 6. Interlude: The Nonlocal Continuum Fiber Bundle Superspace Algorithm—Summary and Transition to Applications

We review and summarize here the salient features of the fiber bundle superspace construction, carefully developed above, by explicitly outlining the algorithm implicit in the various detailed derivations of the previous sections. Our main objective in this short transitional section is to highlight again the fact, already discussed above, which is that our superspace formalism is based on estimating the *physics*-based nonlocality microdomain set $\mathcal{V}(\mathcal{D}) = \{V_{\mathbf{r}}, \mathbf{r} \in \mathcal{D}\}$ associated with the nonlocal continuum $\mathcal{D}$. These data can be obtained only through physical theory and/or measurement. However, once available, the construction of the fibered space proceeds in a computationally well-determined manner. We first summarize the algorithm then provide few additional preparatory remarks before moving to the more detailed and concrete computational examples of Section 7.

In Figure 4, we show two distinct points $\mathbf{r}_1, \mathbf{r}_2 \in \mathcal{D}$ and their associated microdomains $V_{\mathbf{r}_1}$ and $V_{\mathbf{r}_2}$, respectively. From the locally finite subcover $\{U_i\}_{i\in I}$ subordinated to $\mathcal{V}(\mathcal{D}) = \{V_{\mathbf{r}}, \mathbf{r} \in \mathcal{D}\}$ we highlight two sets

$$U_i \subseteq V_{\mathbf{r}_1}, \quad U_j \subseteq V_{\mathbf{r}_2}, \tag{83}$$

where in general it is allowed that

$$V_{\mathbf{r}_1} \cap V_{\mathbf{r}_2} \neq \varnothing, \quad U_i \cap U_j \neq \varnothing, \tag{84}$$

as could be inferred from a glance at the Figure itself. For the partition of unity $(U_i, \psi_i)_{i \in I}$, which is subordinated to the open cover $\{U_i\}_{i \in I}$, we also highlight the two compact sets

$$S_i := \text{supp}\{\psi_i(\mathbf{r})\}, \quad S_i := \text{supp}\{\psi_i(\mathbf{r})\}, \tag{85}$$

forming the support of the corresponding partition of unity functions.

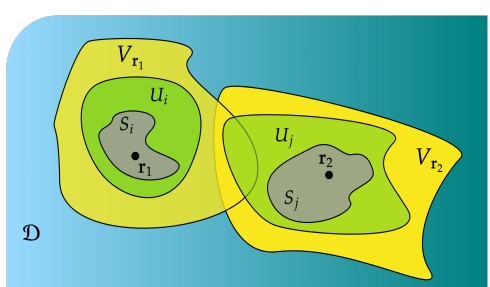

**Figure 4.** An example illustrating the various topological microstructures involved in modeling a generic nonlocal material. The microdomains $V_{\mathbf{r}_1}, V_{\mathbf{r}_2} \in \mathcal{V}(\mathcal{D})$ are open sets and belong to the nonlocal microstructure of the MTM $\mathcal{D}$. The open sets $U_i$ and $U_j$ are the corresponding coordinate sets and partition of unity functions $\{\psi_i\}_{i \in I}$'s domains subordinated to $V_{\mathbf{r}_1}$ and $V_{\mathbf{r}_2}$, respectively. The compact sets $S_i$ and $S_j$ are defined by $S_i := \text{supp}\{\psi_i(\mathbf{r})\}$ and $S_i := \text{supp}\{\psi_i(\mathbf{r})\}$.

The nonlocal material continuum's superspace algorithm itself is summarized in Algorithm 1. Once the microdomain dataset $\mathcal{V}(\mathcal{D})$ is given, the construction proceeds automatically using the partition of unity basis functions $(U_i, \psi_i)_{i \in I}$. The latter may be computed directly in terms of the standard bump functions, see [57,68,71], and also Remark 9.

Because of the fundamental importance of the physics-based nonlocality microdomain structure $\mathcal{V}(\mathcal{D})$, Section 7 will be entirely devoted to the explication of a quantitative practical example illustrating the origin of these microdomains in the concrete setting of a real-life advanced material system, including how the microdomain topology itself may be estimated in practice. In the subsequent sections Section 8 and Appendixes A.3 and A.11, we also explore the usefulness of the superspace homomorphism construction developed in Section 5 for reformulating boundary-value problems in the nonlocal continuum field theories of mathematical physics, besides also providing some hints and additional remarks on other current and future applications.

---

**Algorithm 1** The nonlocal continuum fiber bundle algorithm.

---

1. Start with a physics-based microdomain structure $\mathcal{V}(\mathcal{D}) = \{V_{\mathbf{r}}, \mathbf{r} \in \mathcal{D}\}$.
2. The open cover $\mathcal{V}(\mathcal{D})$ of $\mathcal{D}$ induces a locally finite subcover $\{U_i\}_{i \in I}$ subordinated to $\mathcal{V}(\mathcal{D})$. It is then automatically equipped with the differential structure of the manifold $\mathcal{D}$, generating the differential atlas $(U_i, \phi_i)_{i \in I}$.
3. The subcover $\{U_i\}_{i \in I}$ is equipped with a partition of unity function set $\{\psi_i\}_{i \in I}$, producing the partition of unity $(U_i, \psi_i)_{i \in I}$.
4. Generate an appropriate Banach/Sobolev/Hilbert space $X_i$ attached to each point $\mathbf{r} \in \mathcal{D}$ using constructions, such as (57).
5. Declare $\mathcal{D}$ the base manifold of the fiber bundle. Construct the bundle space $\mathcal{M}$ using

$$\mathcal{M} := \{(\mathbf{r}, X_i) | \forall i \in I, \mathbf{r} \in U_i \subset \mathcal{D}\}. \tag{86}$$

6. Construct the projection map $p : \mathcal{M} \to \mathcal{D}$ through the operation $(\mathbf{r}, X_i) \to \mathbf{r}$.
7. Use (68) to transform vector from one fiber (function) space to another.

---

## 7. Applications to Advanced Materials: Nonlocal Inhomogeneous Semiconductors

### 7.1. Introduction

A concrete example involving spatially-dispersive isotropic media is considered in this Section, where the intention is to provide an outline of how the intricate fiber bundle type

topological fine structure (the topology of microdomains attached to each point explored above as developed in detail in Section 5 and summarized in Section 6) may be estimated in actual practice. The contents of the example given below are rather detailed, and that is for two main reasons. First, in spite of the fact that nonlocal metamaterials are not proposed here for the first time, the author's experience indicates that there is still a general lack of appreciation of the subject in the large community, where most research on "metamaterials" concentrate on *temporally*-dispersive media. Because of that, we provide a very detailed example, including reintroducing some of the well-known physics of semiconductors (in some of the appendices) in order to make the presentation complete and self sufficient. Second, the detailed example to be found below is itself novel. The estimation of the microdomain nonlocal structure in inhomogeneous semiconductors seems to be achieved here for the first time. Therefore, it is a topic that could be treated not merely as an example illustrating the more general and abstract superspace theory developed in the earlier sections, but possibly as a stand-alone contribution to semiconductor materials and their physics. However, the main intention behind the inclusion of this highly-technical physical example continues to be the illustration of the fundamental superspace formalism. More detailed examinations of nonlocality in semiconductor metamaterials belong to a more specialized literature than the current article, whose main topic is the mathematical physics of nonlocal continuum field theories.

### 7.2. A Topological Coarse-Grained Model for Inhomogeneous Nonlocal Material Domains

A review of the homogeneous medium model of spatial dispersion is provided in Appendix A.6. Below, we describe a method that can help transitioning from the generic form (A3), valid for homogeneous nonlocal domains, to the *inhomogeneous* medium situation developed throughout this paper where nonlocality cannot be captured by a simple global dependence of the dielectric function on $\mathbf{k}$. However, instead of working with the full nonlocal function $\overline{\mathbf{K}}(\mathbf{r}, \mathbf{r}')$, an alternative simplified model is proposed which we entitle *the topological coarse-grained model.* The idea is as follows. Consider a global material domain $\mathcal{D}$, which is an open three-manifold, say an open subset of $\mathbb{R}^3$ that may be either simply connected or disconnected.[21] The material is nonlocal and inhomogeneous. At each point $\mathbf{r} \in \mathcal{D}$, a microdomain, i.e., and open set $V_\mathbf{r} \subset \mathcal{D}$, is assigned. The medium is *locally isotropic and homogeneous* in the sense that within each microdomain we can describe the response to an external field excitation $\mathbf{E}$ by means of a relation similar to (34), namely:

$$\mathbf{D}(\mathbf{r};\omega) = \varepsilon_0 \int_{V_\mathbf{r}} \mathrm{d}^3 r' \, \overline{\mathbf{K}}(\mathbf{r} - \mathbf{r}';\omega) \cdot \mathbf{E}(\mathbf{r}';\omega). \tag{87}$$

That is, the only difference between (87) and (34) is that, in the former, we use the correct form of *homogeneous* nonlocality $\overline{\mathbf{K}}(\mathbf{r} - \mathbf{r}';\omega)$ instead of $\overline{\mathbf{K}}(\mathbf{r}, \mathbf{r}';\omega)$. Moreover, we have put the proper response and excitation fields $\mathbf{D}(\mathbf{r})$ and $\mathbf{E}(\mathbf{r})$ and inserted the free space permittivity $\varepsilon_0$.

Fundamentally speaking, each material microdomain is now described by a *spatially-dispersive* model of the form (87). The "topological atoms" of nonlocality, namely the sets $V_\mathbf{r}$, spanned by the continuous index $\mathbf{r} \in \mathcal{D}$, are each a spatially-dispersive "medium" on its own. As will be seen later in this section, the idea of the locally–spatially-dispersive nonlocal semiconductor system is to build an *inhomogeneous* metamaterial that goes *beyond* spatial dispersion by assembling a more general form of nonlocality using the spatially-dispersive material "atoms" $V_\mathbf{r}$. In such systems, the engineered metamaterial is only *locally* spatial dispersive. On the other hand, at a larger spatial scale it does not follow the standard spatial dispersion law, but rather appears to belong to a more complicated class of nonlocal continua which, we believe, are best mathematically described using the fiber bundle superspace formalism of Section 5.

It may be seen then that as a topological coarse-grained process, the original inhomogeneous nonlocal medium, ultimately described by the material tensor $\overline{\mathbf{K}}(\mathbf{r}, \mathbf{r}';\omega)$, is sub-decomposable into "small topological cells", the microdomains $V_\mathbf{r}$, $\mathbf{r} \in \mathcal{D}$, such that

each "topological cell" or "atom" would in itself behave like a homogeneous nonlocal isotropic subdomain, hence may be described by (87), where the material tensor in that case takes the (topologically) locally correct form (A3). This can be considered a *quasi-local* model (also sometimes called *locally spatially dispersive*), where the global domain, electromagnetically speaking, is nonlocal, while, on the other hand, seen at the scale of a small region (cells) it would more or less behave like a typical electromagnetic local medium, see, for example, the discussion of some special cases of complex nonlocal crystals in [76].

**Remark 17.** We remind the reader again about the subtle difference between *mathematical* nonlocality and *physics*-based nonlocality, a distinction at the conceptual level that will become quite visible throughout this section. The term *local* is used in this paper in two senses. The first sense is the physical one in which *local* is set against *physical nonlocality*, which includes spatial dispersion (EM local/nonlocal.) On the other hand, in topology, a *local* property is that which holds in a small open neighborhood of a given point, in our case the topological microdomain $V_{\mathbf{r}}$. The distinction between the two technical senses of the same term should always be clear from the context. In the few cases when there is a risk of confusion, we say *topologically local* to emphasize the second meaning above from *EM local.* (see also Remark 3 and Section 3.3).

Our key objective now is to first develop a simple estimation of the "size" of the nonlocal microdomains $V_{\mathbf{r}}$. To do so, some metric methods must be introduced. An attractive approach would be to approximate the topology of the nonlocal metamaterial system using arrays of various *spheres*, and then use this array in order to obtain the topological content of the microdomain structure described in Section 4.

Let us illustrate the main ideas with a simple example first. Consider a point $\mathbf{r}_1$, which provides a label for one of the micro cells, we may deploy for creating a coarse-grained model for the inhomogeneous medium. To be more specific, let us construct the topological open ball defined by

$$B(\mathbf{r}, a_{\mathbf{r}}) := \left\{ \mathbf{r} \in \mathbb{R}^3 \,\middle|\, d(\mathbf{r}, \mathbf{r}') < a_{\mathbf{r}} \right\}, \tag{88}$$

where $a_{\mathbf{r}} \in \mathbb{R}^+$ is a number quantifying the smallness of this "nonlocality ball" centered at $\mathbf{r}'$, while $d$ is the distance metric. The number $a_{\mathbf{r}}$ will be determined later based on the actual physics of the problem.

Next, the fine-grained topological microdomain structure can be constructed by aggregating all these balls in order to produce a coarse-grained of the overall inhomogeneous nonlocal material domain $\mathcal{D}$. The choice of the *shape* of the microdomain $V_{\mathbf{r}}$ as a sphere $B(\mathbf{r}, a_{\mathbf{r}})$ defined by (88) is justified by our earlier assumption that the material is (topologically) *locally* isotropic. However, note that *globally* electromagnetic processes need *not* behave as they do in isotropic domains.

In Figure 5, a diagrammatic depiction of the two local and global processes is provided where we illustrate:

1.  The proposed topological coarse-grained model utilizing the set of balls $V_{\mathbf{r}}$, $\mathbf{r} \in \mathcal{D}$ (left).
2.  The conventional paradigm where the unit cells are non-overlapping (right).

As can be seen from the diagram, in the topological approach, there exists an open set (microdomain) $V_{\mathbf{r}}$ attached to each point $\mathbf{r} \in \mathcal{D}$ such that nearby microdomains may *overlap* with each other; i.e., in such case the set

$$\cup_{\mathbf{r}_1, \mathbf{r}_2 \in \mathcal{D}} \left[ V_{\mathbf{r}_1} \cap V_{\mathbf{r}_2} \right] \tag{89}$$

is not necessarily empty. On the other hand, the conventional approach to coarse-grained, depicted in Figure 5 (right), involves subdomains like $V'_{\mathbf{r}_1}$ and $V'_{\mathbf{r}_2}$ that are *non-overlapping*, leading to a grid-like structures or "tile covering up" of the material domain $\mathcal{D}$ where in general no holes are left.

In both approaches, it should be noted, we find that each type of the two subdomains, whether $V_{\mathbf{r}}$ or $V'_{\mathbf{r}}$, was already assumed to be *homogeneous*. The disadvantage of the conventional approach is that any *abrupt* change in the electromagnetic properties of the material, experienced when transitioning between two neighboring subdomains through their interface region, often requires imposing a suitable "boundary condition" at this geometric interface in order to obtain an accurate computational assessment of the physics. On the other hand, this problem does *not* exist in the topological approach, illustrated in Figure 5 (left), because the microdomains are allowed to overlap, where common regions between overlapping microdomains are treated correctly using the partition of unity basis functions as described in Section 5.3.

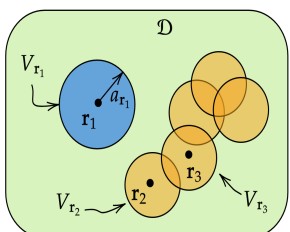 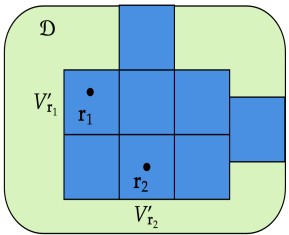

Topological Coarse-Graining Model          Conventional Coarse-Graining Model

**Figure 5.** Topological coarse-grained model for an inhomogeneous nonlocal material domain $\mathcal{D}$ (**left**) in comparison with a conventional coarse-grained process (**right**). The topological microdomains constitute an open cover of the domain in the sense that $\mathcal{D} = \bigcup_{\mathbf{r} \in \mathcal{D}} V_{\mathbf{r}}$, which is the obvious generalization of (32). Note how the topological approach allows overlapping microdomains, e.g., between microdomains $V_{\mathbf{r}_2}$ and $V_{\mathbf{r}_3}$. The technique of the partition of unity will take care of electromagnetic data "repeated" in such regions of overlap by assigning proper weights that always sum to unity at each point in $\mathbf{r} \in \mathcal{D}$.

### 7.3. Resonant Nonlocal Semiconductor Domains and the Nonlocal Exciton-Polariton Model

A concrete application of the topological coarse-grained algorithm proposed in Section 7.2 is now in order. The specific nonlocal metamaterial is a semiconductor with dielectric function exhibiting a single strong resonant exciton transition at the frequency $\omega = \omega_{\mathrm{e}}$. We first examine in detail the nonlocal exciton–polariton model to be used below. For a review on the physics of exciton–polariton interactions in solids, see Appendix A.7.

A *polariton* is simply a "photon living inside a dielectric medium". The quantum of an electromagnetic wave inside a dielectric domain is often called polariton instead of photons (sometimes polaritons are called "dressed photons"). An *exciton–polariton* is a polariton coupled with a mechanical exciton, e.g., an electron-hole pair. The latter should be distinguished from other types of polaritons such as *phonon-polaritons* defined as polaritons coupled with *phonons*, the quantum of lattice vibrations [37].

It is well known from quantum theory that near resonance, the dielectric function of such semiconductor materials may be approximated by the formula [16,77–80]:

$$\varepsilon(\mathbf{k}, \omega) = \varepsilon_0 + \frac{\chi}{k^2 - \gamma^2(\omega)}, \tag{90}$$

where

$$\chi = 4\pi \frac{\alpha m_{\mathrm{e}}^{\star} \omega_{\mathrm{e}}}{\hbar}, \quad \gamma^2(\omega) = \frac{m_{\mathrm{e}}^{\star}}{\hbar \omega_{\mathrm{e}}} \left( \omega^2 - \omega_{\mathrm{e}}^2 + \mathrm{i}\omega\Gamma \right). \tag{91}$$

Here, $\hbar$ is the reduced Planck constant, while $\alpha$ serves as the oscillator strength.[22] The effective mass of the exciton is denoted by $m_{\mathrm{e}}^{\star}$.[23] On the other hand, the *exciton lifetime* $\tau_{\mathrm{e}}$ is defined by

$$\tau_{\mathrm{e}} := \frac{2\pi}{\Gamma}, \tag{92}$$

hence, $\Gamma$ can be thought of as the *exciton decay* or *relaxation rate*. We emphasize that any dependence of $\Gamma$ and the oscillator strength $\alpha$ on $k$ is ignored in the excitonic model (90).

In Appendix A.7, the physical origin of nonlocality in the semiconductor is revisited, where it is traced to the quantum mechanical energy–momentum relations of exciton–polaritons. In order to actually see significant nonlocal physics taking place in the excitonic material system described by (90), the following sufficient condition may be imposed:

$$\Gamma \ll \frac{\hbar k^2}{2m_e^*}. \tag{93}$$

It can be shown that under such $\Gamma$-bound, the kinetic energy term in (A12) can induce significant nonlocal effects in (90). One way to realize nonlocal (spatially dispersive) semiconducting metamaterials is to operate with intrinsic semiconductors satisfying (93) by keeping the temperature low and the material pure (undoped) [83].

The model described by (90) and (91) can be viewed as a natural generalization of the local Lorentz model widely utilized to model temporal dispersion in solids and plasma [77,79]. It represents the simplest nonlocal resonant model with a single strong resonance at a characteristic frequency, here $\omega = \omega_e$. All other off-resonance excitonic transitions are gathered into the background dielectric constant $\varepsilon_0$ for simplicity. For frequencies well below $\omega \ll \omega_e$, the exciton–polariton behaves essentially like a photon propagating in a medium with background permittivity $\varepsilon_0$. For $\omega \gg \omega_e$, we again recover photons but usually with a background described by $\varepsilon_\infty$, the high-frequency limit of permittivity. In general, the difference between the static and high-frequency permittivities is quite small in the sense that

$$|\varepsilon_0 - \varepsilon_\infty| \ll \varepsilon_0. \tag{94}$$

Hence, for simplicity, in this example the two permittivities are treated as identical ($\varepsilon_0 \simeq \varepsilon_\infty$) since we are interested in the EM response around a single excitonic resonance while in fact the oscillator strength $\alpha$ in (90) is small. One consequence of this assumption is that the splitting between longitudinal and transverse modes can be neglected. Indeed, since the longitudinal and transverse frequencies $\omega_L$ and $\omega_T$ are related to each other via the relation [16]

$$\frac{\omega_L^2}{\omega_T^2} = \frac{\varepsilon_0}{\varepsilon_\infty}, \tag{95}$$

then the assumption (94) is equivalent to neglecting the longitudinal–transverse splitting

$$\omega_{L,T} := |\omega_L - \omega_T| \tag{96}$$

in the sense that

$$\omega_{L,T} \ll \omega_T. \tag{97}$$

A consequence of this is the near equality of the longitudinal and transverse frequencies, which allows us to considerably simplify the mathematical treatment.[24] In addition, assuming that the oscillator strength $\alpha$ in (90) is nearly the same for both the longitudinal and transverse part of the response function, then it follows that we need only work with a single *scalar* response function, namely the form (90) itself instead of the more general tensorial Formula (A3).[25]

Nonlocal effects associated with the model (90) emerge from the quantum mechanical nature of exciton–polariton interactions and the need to enforce conservation of energy/momentum as discussed in Appendix A.7, leading to the strong dependence on $k$ observed in (90). There is yet another physical explanation of nonlocality. Within the regime of the large exciton mass limit

$$m_e^* \to \infty, \tag{98}$$

the kinetic energy term in (A12) drops out and the excitonic dielectric function (90) becomes local. This is why spatial dispersion is sometimes referred to as the "finite-mass model",

with some suggestions that the origin of nonlocality in this case is the inertial effects of the exciton [79].[26] In what follows, we assume that the effective mass of the exciton is always finite and positive:

$$0 < m_{\mathrm{e}}^{\star} < \infty. \tag{99}$$

However, it should be noted that since excitons are *collective excitations* of solids [84,85], they may have negative mass [86]. While this will not be pursued here, the negativity of the excitonic mass may be exploited in order to further design and control the EM behavior of nonlocal MTMs constructed using excitonic semiconductors.

In order to gain a deeper insight into the various resonance structures of the exciton–polariton response function (90), we rewrite it in the equivalent form

$$\varepsilon(\mathbf{k}, \omega) = \varepsilon_0 + \frac{\chi / k_{\mathrm{e}}^2}{k^2 / k_{\mathrm{e}}^2 + 1 - \omega^2 / \omega_{\mathrm{e}}^2 - i\omega\Gamma / \omega_{\mathrm{e}}^2}, \tag{100}$$

where

$$k_{\mathrm{e}} := \frac{2\pi}{\lambda_{\mathrm{e}}} = \sqrt{\frac{m_{\mathrm{e}}^{\star} \omega_{\mathrm{e}}}{\hbar}} \tag{101}$$

is called the *exciton wave number*. The wavelength $\lambda_{\mathrm{e}}$ is a fundamental resonance spatial scale, which we will refer to as the *exciton wavelength* and is given by

$$\lambda_{\mathrm{e}} = \frac{1}{2\pi} \sqrt{\frac{\hbar}{m_{\mathrm{e}}^{\star} \omega_{\mathrm{e}}}}. \tag{102}$$

For example, with $\hbar\omega_{\mathrm{e}} = 2.5\,\mathrm{eV}$ and $m_{\mathrm{e}}^{\star} = 0.9 m_{\mathrm{el}}$, where $m_{\mathrm{el}}$ is the electron mass, the exciton wavelength $\lambda_{\mathrm{e}}$ is around 0.0293 nm, which is the same order of magnitude of interatomic spacing. The excitation field wavelength $\lambda$ is at least one order of magnitude larger. Later we will show typical values for the topological microdomain radius $a_{\mathbf{r}}$.

There are several fundamental spatial and temporal scales involved in the process of describing the generic nonlocal metamaterial domain $\mathcal{D}$. The excitation field $\mathbf{E}(\mathbf{r})$ itself introduces its own temporal excitation period

$$T := \frac{2\pi}{\omega}, \tag{103}$$

in addition to a purely spatial scale (wavelength) measured by the formula

$$\lambda := \frac{2\pi}{k}. \tag{104}$$

On the other hand, the excitonic transition as such is associated with the fundamental (temporal) transition period

$$T_{\mathrm{e}} := \frac{2\pi}{\omega_{\mathrm{e}}}, \tag{105}$$

while a fundamental spatial scale

$$\lambda_{\mathrm{e}} := \frac{2\pi}{k_{\mathrm{e}}} \tag{106}$$

can be unambiguously linked to the exciton at the same time. Table 1 gives a summary of all these parameters with their meaning explicitly stated. Moreover, it will be demonstrated later that the radius of a topological microdomain $V_{\mathbf{r}}$, which is based at a generic position $\mathbf{r} \in \mathcal{D}$, can be given by a special Formula (120). Nonlocality arises from the delicate interplay between all these different spatial and temporal scales. In what follows, we will emphasize their relative roles in determining the rich nonlocal microstructure of the material domain, while introducing quantitative calculations.

**Table 1.** A summary of the various spatial and temporal scales involved in understanding and designing generic nonlocal metamaterials with exciton–polariton resonance-type of nonlocality.

| Scale | Type | Meaning | Formula |
|:-----:|:----:|:-------:|:-------:|
| $\lambda$ | spatial | excitation field wavelength | $\lambda = 2\pi/k$ |
| $\lambda_{\mathrm{e}}$ | spatial | exciton wavelength | $2\pi/k_{\mathrm{e}}$ |
| $a_{\mathbf{r}}$ | spatial | microdomain radius | $1/\lvert\gamma''\rvert$ |
| $T$ | temporal | excitation field period | $2\pi/\omega$ |
| $\tau_{\mathrm{e}}$ | temporal | exciton lifetime | $2\pi/\Gamma$ |
| $T_{\mathrm{e}}$ | temporal | exciton period | $2\pi/\omega_{\mathrm{e}}$ |

Armed with this typology of spatial and temporal scales, we are now better positioned to understand the resonance structure associated with the exciton–polariton nonlocal dielectric function (100). Figure 6 illustrates two cases of resonance where the value of the dielectric function is examined with respect to variations in the excitation field wave number $k$ (or equivalently the wavelength $\lambda$). In order to focus on nonlocality, we only plot the nonlocal part of the total response, which is found here to be proportional to the dielectric residue

$$\varepsilon(\mathbf{k}, \omega) - \varepsilon_0. \tag{107}$$

As we may infer from Figure 6, a strong resonance takes place when the ratio

$$\frac{k}{k_{\mathrm{e}}} = \frac{\lambda_{\mathrm{e}}}{\lambda} \tag{108}$$

becomes comparable in magnitude to the quantities remaining in the denominator of (100). That is, the spatial resonance condition is

$$\frac{k^2}{k_{\mathrm{e}}^2} + 1 - \frac{\omega^2}{\omega_{\mathrm{e}}^2} \sim \frac{\omega\Gamma}{\omega_{\mathrm{e}}^2}. \tag{109}$$

However, the condition (109) holds only if the imaginary part of the denominator of (100), i.e., the quantity $\omega\Gamma/\omega_{\mathrm{e}}^2$, is relatively small. Otherwise, since $k$ and $k_{\mathrm{e}}$ are real, the ratio $k/k_{\mathrm{e}}$ can never lead to strong resonance when the relaxation rate $\Gamma$ is sufficiently large. Another way to say the same thing is the following: strong *spatial* resonances, whose main origin is nonlocality, can take place either when dissipation is small, or when the exciton lifetime is long enough. The latter scenario of long exciton lifetime is characterized by the condition

$$\frac{\omega\Gamma}{\omega_{\mathrm{e}}^2} \ll \left\lvert 1 - \frac{\omega^2}{\omega_{\mathrm{e}}^2} \right\rvert. \tag{110}$$

In such case, it is evident that the appropriate spatial and temporal sufficient conditions needed to secure nonlocal resonance are mutually related by the simple relation

$$\frac{k^2}{k_{\mathrm{e}}^2} \approx \frac{\omega^2}{\omega_{\mathrm{e}}^2} - 1. \tag{111}$$

From this, it can be inferred that nonlocal resonances generally occur only for $\omega/\omega_{\mathrm{e}} > 1$. In Figure 6 (left), we can see that for the above-resonance condition of $\omega/\omega_{\mathrm{e}} = 1.5$, the nonlocal domain possesses a spatial resonance at roughly $\lambda \approx \lambda_{\mathrm{e}}$. On the other hand, if we operate the material at larger frequency $\omega/\omega_{\mathrm{e}} = 2.5$, i.e., well above the exciton transition frequency, then spatial resonances may occur only at values of the excitation field wavelength $\lambda$ that are considerably smaller than the exciton wavelength $\lambda_{\mathrm{e}}$.

Finally, we add that when the nonlocal response is plotted as function of $\omega$ instead of $k$, resonance structures similar to Figure 6 are obtained under the condition (110) since in that case (111) approximately holds. In general, we would expect that for the best operation of the designed nonlocal MTM (maximal nonlocal response), the operating frequency should

be selected to be as close as possible to the exciton transition frequency, i.e., we would like to maintain the material design condition

$$\frac{\omega}{\omega_e} \approx 1,\tag{112}$$

which is needed since, in general, the excitonic relaxation rate $\Gamma$ is never exactly zero and, hence, the condition (110) seldom holds otherwise for all frequencies.

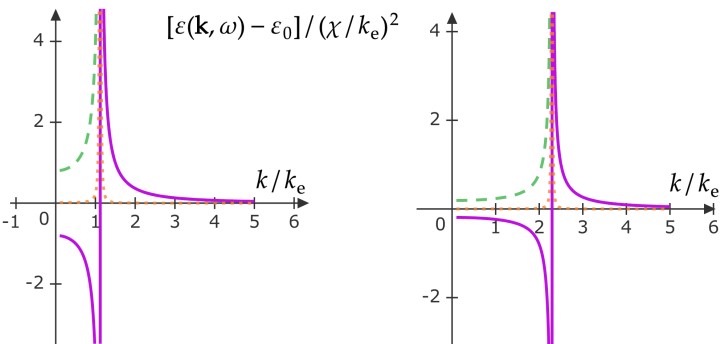

**Figure 6.** The nonlocal spatial resonance structure of the exciton–polariton dielectric response as a function of the excitation field wave number $k$. The normalized response function $(\varepsilon(\mathbf{k},\omega) - \varepsilon_0)/(\chi/k_e)^2$ is plotted, where the dashed line is the absolute value, the solid line represents the real part, while the dotted line is the imaginary part. For both figures, $\Gamma/\omega_e = 0.01$. (**Left**) $\omega/\omega_e = 1.5$. (**Right**) $\omega/\omega_e = 2.5$.

*7.4. Quantitative Estimation of the Electromagnetic Nonlocality Microdomain Structure in the Exciton-Polariton Dielectric Model*

In the spatial domain, the dielectric function can be obtained by computing the inverse Fourier transform

$$\varepsilon(\mathbf{r} - \mathbf{r}';\omega) = \mathcal{F}_{\mathbf{k}}^{-1}\{\varepsilon(\mathbf{k},\omega),\tag{113}$$

where $\mathcal{F}_{\mathbf{k}}^{-1}$ is the converse of the forward Fourier transformation defined by (17). We will need the following inverse Fourier transform relation (proved in Appendix A.9):

$$\mathcal{F}_{\mathbf{k}}^{-1}\left\{\frac{\chi}{k^2 - \gamma^2(\omega)}\right\} = \frac{\chi}{4\pi}\frac{e^{\gamma''(\omega)|\mathbf{r}-\mathbf{r}'|}e^{-i\gamma'(\omega)|\mathbf{r}-\mathbf{r}'|}}{|\mathbf{r} - \mathbf{r}'|},\tag{114}$$

where

$$\gamma' = -\sqrt{\frac{m_e^\star}{2\hbar\omega_e}}\sqrt{(\omega^2 - \omega_e^2) + \sqrt{(\omega^2 - \omega_e^2)^2 + (\omega\Gamma)^2}},\tag{115}$$

$$\gamma'' = -\sqrt{\frac{m_e^\star}{2\hbar\omega_e}}\sqrt{-(\omega^2 - \omega_e^2) + \sqrt{(\omega^2 - \omega_e^2)^2 + (\omega\Gamma)^2}}.\tag{116}$$

Hence, by substituting (100) into (113) and using (114), we arrive at

$$\varepsilon(\mathbf{r} - \mathbf{r}';\omega) = \underbrace{\varepsilon_0\delta(\mathbf{r} - \mathbf{r}')}_{\text{local response}} + \underbrace{\varepsilon_{\text{NL}}(\mathbf{r} - \mathbf{r}';\omega)}_{\text{nonlocal response}}.\tag{117}$$

The first terms in the RHS of (117) provides the background local response of the medium. On the other hand, all nonlocal effects are relegated to the second term in the RHS of (117):

$$\varepsilon_{\text{NL}}(\mathbf{r} - \mathbf{r}';\omega) := \frac{\alpha m_e^\star \omega_e}{\hbar}\frac{e^{-i\gamma'(\omega)|\mathbf{r}-\mathbf{r}'|}}{|\mathbf{r} - \mathbf{r}'|}e^{\gamma''(\omega)|\mathbf{r}-\mathbf{r}'|},\tag{118}$$

which is nothing but the Green's function of the electromagnetic semiconductor material system under investigation.

The Green's function (118) is the most fundamental physical quantity needed for the construction of the microdomain structure $\mathcal{D}$ of the nonlocal medium. It has some similarity with the scalar free-space Green function for radiation fields, i.e., spherical waves of the form:

$$\frac{\exp(ik|\mathbf{r} - \mathbf{r}'|)}{|\mathbf{r} - \mathbf{r}'|}. \tag{119}$$

However, there are notable differences:

1.  First, we note that (118) exhibits strong dispersive behavior due to the dependence of $\gamma'$ and $\gamma''$ on frequency per their Formulas (115) and (116).
2.  Second, the presence of a spatially-decaying exponential factor of the form $\exp(\gamma'|\mathbf{r} - \mathbf{r}'|)$ makes the Green function $\varepsilon_{\mathrm{NL}}(\mathbf{r} - \mathbf{r}'; \omega)$ highly attenuating in spite of the fact that this attenuation is *not* mainly due to thermodynamic losses.

Indeed, as can be seen from (90), dissipation is controlled by the exciton lifetime $\tau_{\mathrm{e}}$, or, equivalently, the decay rate $\Gamma$. Dissipation decreases as the lifetime increases, i.e., when $\Gamma$ is small. Figure 7 illustrates some examples where we plot both $\gamma'$ and $\gamma''$ as functions of frequency. The frequency-dependent behavior observable there strongly depends on $\Gamma/\omega_{\mathrm{e}}$, i.e., the ratio between the relaxation frequency and the excitonic transition frequency. For ratios as small as $\Gamma/\omega_{\mathrm{e}} = 0.1$, the intensity of attenuation per unit length $\gamma''$ is nearly constant for $\omega > \omega_{\mathrm{e}}$, while it assumes higher values for frequencies below the $\omega_{\mathrm{e}}$ as can be seen from Figure 7a. This is consistent with a "high-pass filtering behavior" typical for this type of resonance phenomena, where waves are often excited with frequencies slightly larger than the cutoff threshold at $\omega_{\mathrm{e}}$. For the propagation constant $\gamma'$ at the same relaxation-to-exciton transition ratio $\Gamma/\omega_{\mathrm{e}}$, Figure 7b shows that it becomes nearly straight line. Such behavior, when combined with nearly constant per-unit-length attenuation, represents negligible dispersion effects. On the other hand, when $\Gamma/\omega_{\mathrm{e}}$ increases, we begin to see strong dispersion effects, manifested by non-constant per-unit-length attenuation and nonlinear phase-delay relations.

In fact, the attenuation process described by the per-unit-length rate $\gamma''$ is not merely an expression of dissipation, but is also *the* signature of nonlocality in exciton–polariton semiconductor materials. The medium response weakens as the distance from the source increases, while the characteristic length scale of this nonlocality radius is found to be solely controlled by $\gamma''$. Figure 8 illustrates the real part of the dielectric function Green's function (118). The ability of the excitonic semiconducting medium to respond to spatially distant sources is graphically illustrated by its dielectric profile's functional spread around the origin $|\mathbf{r} - \mathbf{r}'| = 0$. The size of the nonlocal domain is then directly reflected by the rapidity of the decay of the Green's function (118) as one moves away from $\mathbf{r}'$, which is the origin here.

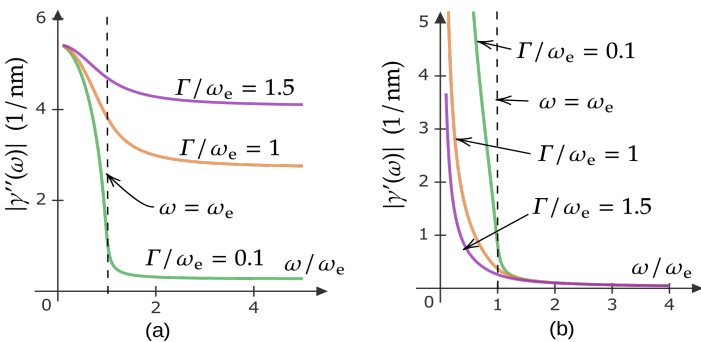

**Figure 7.** Frequency dependence of $\gamma''$ (**a**) and $\gamma'$ (**b**) for several values of the exciton decay rate $\Gamma$. Here, $m_{\mathrm{e}}^{\star} = 0.9 m_{\mathrm{e}}$, where $m_{\mathrm{e}}$ is the electron mass. The exciton transition frequency is $\omega_{\mathrm{e}} = 3.7977 \times 10^{15}$ rad/s ($\hbar\omega_{\mathrm{e}} = 2.5$ eV).

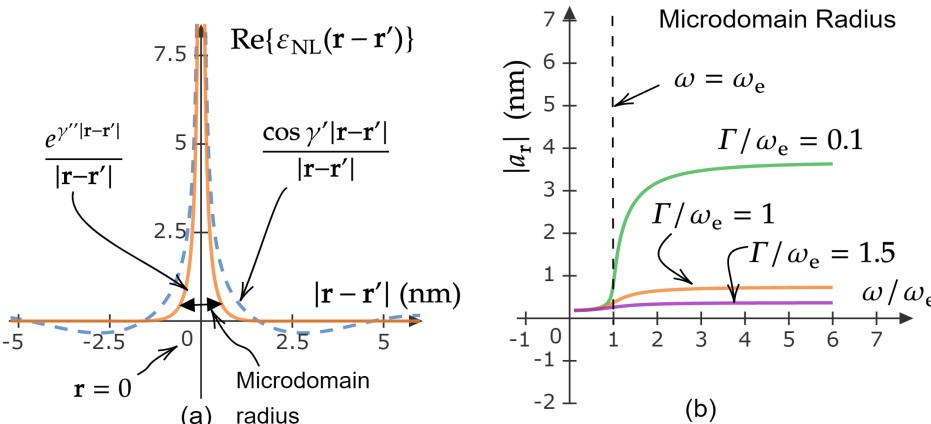

**Figure 8.** (**a**) Comparison between the real parts of the long-range decay of the excitonic nonlocal domain Green function $\varepsilon_{\mathrm{NL}}(\mathbf{r} - \mathbf{r}')$ with and without the full spatial dependence, including the exponential short-range decay factor $\exp(\gamma''|\mathbf{r} - \mathbf{r}'|)$ for $\gamma' = 1\ \mathrm{nm}^{-1}$ and $\gamma'' = 2\ \mathrm{nm}^{-1}$. (**b**) Frequency dependence of $a_{\mathbf{r}}$, the radius of the topological microdomain $B(\mathbf{r}, a_{\mathbf{r}})$ centered at some generic point $\mathbf{r}$ in the nonlocal excitonic material domain $\mathcal{D}$ for several values of the exciton lifetime $\Gamma^{-1}$. Here, $m_{\mathrm{e}}^{\star} = 0.9 m_e$, where $m_{\mathrm{e}}$ is the electron mass. The exciton transition frequency is $\omega_{\mathrm{e}} = 3.7977 \times 10^{15}\ \mathrm{rad/s}$ ($\hbar\omega_{\mathrm{e}} = 2.5\ \mathrm{eV}$).

### 7.5. The Locally-Homogeneous Model of Nonlocal Semiconducting Domains

Quasi-inhomogeneous, also known as smoothly-inhomogeneous or locally spatially dispersive nonlocal media, are some of the simplest possible prototypes of general (inhomogeneous) nonlocal materials where the spatial dispersion model $\varepsilon(\mathbf{k})$, with a dependence on only one spatial spectral variable $\mathbf{k}$, is found to be not adequate for the mathematical description of the physics of the nonlocal system [76,87]. In contrast, one would need the considerably more complex spectral functions of the form $\varepsilon(\mathbf{k}, \mathbf{k}')$, which are three-dimensional spatial Fourier transformations of generic nonlocal response functions like (3) or (10). In general, there has been quite few investigations aimed at going beyond spatial dispersion in homogeneous media. Examples include inhomogeneous plasma, such as those in controlled-fusion reactors [88], cold collisionless magnetoplasma [88], the electro-dynamics of nanostructures [89–92], and incommensurately-modulated superstructures in insulators [76,93].

Here, we will analyze a simple inhomogeneous model of semiconductors experiencing exciton–polariton transitions as outlined above. The EM nonlocal model is *locally-homogeneous* in the sense that around each point $\mathbf{r} \in \mathcal{D}$ there exists a *topologically*-local neighborhood, namely the microdomain $V_{\mathbf{r}}$, inside which the medium can be modeled as a homogeneous and spatially dispersive domain for all $\mathbf{r} \in V_{\mathbf{r}}$ (i.e., the second mention of "locally" here means *topological* nonlocality, see Remarks 3 and 17). It should be noted though that for maximum generality, we allow for variations in the spatial dispersion model to take place from one microdomain $V_{\mathbf{r}}$ to another.

We now wish to estimate the size of each nonlocality microdomain with the help of the exponential law in (118). Let us first expand the homogeneous model treated in Section 7.3 to the inhomogeneous setting of the present discussion, where currently we need to allow that at each point $\mathbf{r} \in \mathcal{D}$, the parameters of the original exciton–polariton model (100) would all become generally functions of the position. That is, in this more general case, one should write $\gamma'(\mathbf{r})$, $\gamma''(\mathbf{r})$, $\omega_{\mathrm{e}}(\mathbf{r})$, $\alpha(\mathbf{r})$, $m_{\mathrm{e}}^{\star}(\mathbf{r})$, etc, where it is understood that the medium's microscopic composition may change from one position to another.

The main formula for computing the size (radius) of the topological microdomain balls $V_{\mathbf{r}} = B(\mathbf{r}, a_{\mathbf{r}})$ can be easily given by the following expression:

$$a_{\mathbf{r}} \simeq \frac{1}{|\gamma''(\mathbf{r})|}. \tag{120}$$

Roughly speaking, the radius given by (120) quantifies the spatial extension of that characteristic phenomenon of field localization entailed by the presence in the medium Green function (118) of exponential factors like $\exp(-|\gamma''|r')$. Using the formula (116), the relation (120) becomes:

$$a_{\mathbf{r}} = \sqrt{\frac{2\hbar/m_{\mathrm{e}}^{\star}(\mathbf{r})\omega_{\mathrm{e}}(\mathbf{r})}{1 - \frac{\omega^2}{\omega_{\mathrm{e}}^2(\mathbf{r})} + \sqrt{\left(\frac{\omega^2}{\omega_{\mathrm{e}}^2(\mathbf{r})} - 1\right)^2 + \frac{\Gamma^2(\mathbf{r})}{\omega_{\mathrm{e}}^2(\mathbf{r})}\frac{\omega^2}{\omega_{\mathrm{e}}^2(\mathbf{r})}}}}. \tag{121}$$

This expression (121) is illustrated with some basic examples as given in Figure 8b for various values of the crucial parameter $\Gamma/\omega_{\mathrm{e}}$. When this ratio between the relaxation rate and the exciton transition frequency is small, the size of the EM nonlocality domain will increase due to the weakening of the corresponding nonlocality-based attenuation (field localization or confinement) processes. Conversely, one may control the size of each EM nonlocality microdomain $V_{\mathbf{r}}$ by modifying the ratio $\Gamma(\mathbf{r})/\omega_{\mathrm{e}}(\mathbf{r})$ evaluated at that position. This may provide a path toward an experimental realization of generalized nonlocal MTMs with controlled microtopological structures. In order to give a view on the numerical values of this structure, Table 2 provides some relevant microdomain data computed by means of the expression (121).

**Table 2.** Topological microdomain data at a generic position $\mathbf{r} \in \mathcal{D}$. The exciton transition frequency is $f_e = 23{,}862$ THz ($\hbar\omega_{\mathrm{e}} = 2.5$ eV), while $m_{\mathrm{e}}^{\star} = 0.9m_{\mathrm{e}}$. For the left table, $\Gamma/\omega_{\mathrm{e}} = 2 \times 10^{-5}$.

| $f$ (THz) | $\omega/\omega_e$ | $a_{\mathbf{r}}$ (μm) | $\Gamma/\omega_e$ | $\omega/\omega_e$ | $a_{\mathbf{r}}$ (μm) |
|---|---|---|---|---|---|
| 19,090 | 0.8 | 0.0003 | 0.00002 | 1.01 | 2.5834 |
| 21,476 | 0.9 | 0.0004 | 0.00020 | 1.01 | 0.2583 |
| 23,862 | 1.0 | 0.0582 | 0.00200 | 1.01 | 0.0259 |
| 26,248 | 1.1 | 7.6670 | 0.02000 | 1.01 | 0.0028 |
| 35,793 | 1.5 | 13.7174 | 0.02000 | 1.01 | 0.0028 |
| 47,724 | 2.0 | 15.9382 | 2.0000 | 1.01 | 0.0002 |
| 59,655 | 2.5 | 16.8674 | 20.000 | 1.01 | 0.0001 |

**Remark 18.** The approximation (120), strictly speaking, is not compatible with Definition 1 since the latter is based on assuming that the material response kernel possesses a compact support. However, for all practical purposes, a decaying exponential can be taken to approximate the behavior of a function with compact support. Nevertheless, in a more careful future treatment it is always possible to modify the exact Definition 1 in order to incorporate the decaying-exponential response kernel as another valid example of effective physics-based nonlocality mathematically realized by a topologically-localized function. An elementary discussion of some possible such modifications is given in Appendix A.10.

## 8. Application to Fundamental Theory: Electromagnetic Boundary Conditions in the Fiber Bundle Superspace Formalism

Armed with the general superspace formalism of nonlocal continua (Section 5) and the detailed practical example illustrating the theory (Section 7), we now turn to a brief reexamination of a topic in fundamental theory: the role of boundary conditions in nonlocal continuum field theories. The well-known tension between nonlocal electromagnetism and intermaterial interfaces has been already mentioned several times above. Here, we provide some application of the fiber bundle theory of Section 5, aiming at elucidating the nature of this tension, and we suggest some possible new formulation of the problem.

The natural starting point is Figure 2, where a zoomed-in topological picture based on the general structures explicated in Section 4 is given. The focus now is on the *interface* between two generic nonlocal domains $\mathcal{D}_n$ and $\mathcal{D}_m$. In traditional local electromagnetism, the constitutive relation material tensor $\overline{\mathbf{K}}_n$ is usually exploited to deduce conditions dictating how various electromagnetic field components behave as they cross the $\mathcal{D}_n/\mathcal{D}_m$

intermaterial interface. However, even if each response function $\overline{\mathbf{K}}_{n/m}(\mathbf{r}, \mathbf{r}')$ was to be treated as one belonging to a spatially dispersive domain, i.e., replacing it by $\overline{\mathbf{K}}_{n/m}(\mathbf{r} - \mathbf{r}')$, the presence of a boundary between two *distinct* material profiles completely destroys the *translational symmetry* of the structure on which the very rigorous derivation of the specific spatially-dispersive nonlocal response tensor $\overline{\mathbf{K}}_{n/m}(\mathbf{r} - \mathbf{r}')$ was originally based.

The breakdown of translational symmetry in inhomogeneous crystal configurations was very clearly identified and explained by Agranovich and Ginzburg [16], together with several proposals for a solution of such unusual electromagnetic problem. For example, because it is evident that close to the intermaterial interface the response tensor of each medium, when seen from its own side while approaching the boundary, must be reverted back to the most general nonlocal form, namely $\overline{\mathbf{K}}_{n/m}(\mathbf{r}, \mathbf{r}')$ instead of $\overline{\mathbf{K}}_{n/m}(\mathbf{r} - \mathbf{r}')$, it was then proposed that one may use the former, more general, functional form, but only within a "thin transitional layer" that includes the intermaterial interface, yet while additionally extending, along some necessarily "ambiguous distance", into the depths of the two material domains $\mathcal{D}_n$ and $\mathcal{D}_m$ on both sides of the boundary. Outside this fuzzy region, a gradual transition, or a continuously changing profile (a tapered channel), is introduced to proceed from the most general forms $\overline{\mathbf{K}}_{n/m}(\mathbf{r}, \mathbf{r}')$, valid in the vicinity of the intermaterial interface, to the special spatially dispersive forms $\overline{\mathbf{K}}_{n/m}(\mathbf{r} - \mathbf{r}')$, which are more accurate the further one goes away from the material boundary, where the latter response tensor functions are considered characteristic of "bulk" homogeneous material domains [16].

Another proposal is to keep using everywhere spatial dispersion profiles of the form $\overline{\mathbf{K}}_n(\mathbf{r} - \mathbf{r}')$, but introduce specialized *additional boundary conditions* (ABCs) at the intermaterial interface based on each particular problem under consideration. Although this latter approach is both mathematically and physically inconsistent (due to the breakdown of symmetry caused by the presence of intermaterial interfaces), it nevertheless remains popular because—at least in outline—nonlocal electromagnetism is thereby held up in a form as close as possible to familiar local electromagnetic theory methods, especially numerical techniques, such as finite element method (FEM) [44], method of moment (MoM) [46], and finite difference time-domain method (FDTD) [45], i.e., established full-wave algorithms where it is quite straightforward to replace one boundary condition by another without essentially changing much of the code.[27]

Nevertheless, both approaches discussed above require considerable input from the *microscopic* theory, mainly to determine the tapering transition region in the case of the first, and the ABCs themselves in the second. That motivated the third approach, called, the ABC-free formalism, where the relevant microscopic theory was utilized right from the beginning in order to formulate and solve Maxwell's equations. For example, in [50,53], a global Hamiltonian of the matter-field system is constructed and Maxwell's equations are derived accordingly. In [38], the rim zone (field attached to matter) is investigated using different physical assumptions to understand the transition from nonlocal material domains to vacuum going through the entire complex near-field zone. In [89], the symmetry group of carbon nanotubes was exploited to construct a set of Maxwell's equations in nonlocal nanoscale problems without using a homogenized electromagnetic field-based boundary condition.

We believe that the main common conclusion from all these different formulations is that in nonlocal electromagnetism *it is not possible in general to formulate the electromagnetic problem at a fully phenomenological level.* In other words, microscopic theory appears to be in demand more often than in the case of systems involving only local materials. However, since all existing solutions use the traditional spatial manifold $\mathcal{D}$ as the main configuration space, the question now is whether the alternative formulation proposed in this paper, the extended fiber bundle superspace formalism, may provide some additional insights into the problem of why nonlocal continuum field theory cannot be formulated in general for inhomogeneous domains as in the local version of that theory.

We provide a provisional elucidation of the topological nature of field theory across intermaterial interfaces by noting that, in Figure 2, it is not only the behavior of the fields

$\mathbf{F}(\mathbf{r})$ in the two domains that is mostly relevant, but also the *entire* set of local topological microdomains $V_{\mathbf{r}}$ clustered on both sides of the interface inside the material domains. More specifically, we attach a great importance to *how these microtopological domains, together with the corresponding set of excitation fields that are applied on them, would behave as they move across the boundary*. In *general (set-theoretic)* topology, *boundaries* are defined fully in terms of the behavior of *open sets* [58,59].

We now build on this key set-theoretic topological concept in order to illustrate how the problem of nonlocal inhomogeneous continuum field theory may be reformulated through the superspace formalism developed in Section 5. First, Figure 9 provides a finer or more structured picture of the topological content of Figure 2 based on replacing the spaces $\mathcal{D}_m$ and $\mathcal{D}_n$ by the corresponding Banach bundle superspaces $\mathcal{M}_m$ and $\mathcal{M}_n$, respectively. The thick horizontal curved lines represent the base spaces $\mathcal{D}_m$ and $\mathcal{D}_n$, while the wavy vertical lines stands for the fiber spaces $X_m$ and $X_n$ attached at each point $\mathbf{r} \in \mathcal{D}_{n/m}$ in the corresponding base manifolds. The double discontinuous lines at the "junction" of the two base spaces $\mathcal{D}_m$ and $\mathcal{D}_n$ indicate the joining together of the two vector bundles $\mathcal{M}_m$ and $\mathcal{M}_n$.

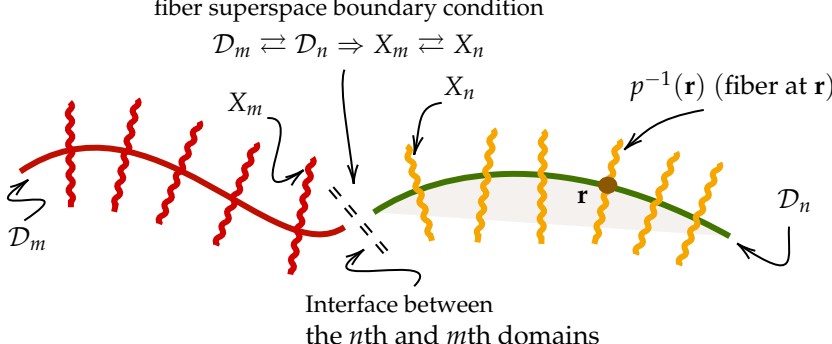

**Figure 9.** An abstract representation of the topological fiber bundle superspace structure behind Figure 2.

It should become clear now that since the two nonlocal material domains possess an *extra structure*, namely that of the individual copies of the fibers, each a linear vector Banach space attached to every point in the base space, we must also indicate how the various elements belonging to the Banach function spaces, i.e., the fields defined on the microdomains $V_{\mathbf{r}}$ in Figure 2,[28] would behave as they cross the boundary separating the two material domains $\mathcal{D}_m$ and $\mathcal{D}_n$. One obvious way to do this is to introduce a *bundle homomorphism* between the two vector bundles $\mathcal{M}_m$ and $\mathcal{M}_n$ over the interface submanifold $\partial \mathcal{D}_{mn}$ separating $\mathcal{D}_m$ and $\mathcal{D}_n$. This mathematical object is similar to the nonlocal response map $\mathcal{L}$ introduced by (72).

The motivation behind introducing this bundle homomorphism is to serve as a "boundary condition operator" acting on the fiber bundle *superspaces* $\mathcal{M}_m$ and $\mathcal{M}_n$ instead of the conventional spaces $\mathcal{D}_m$ and $\mathcal{D}_n$ always used in local continuum field theories. We will not go here into a detailed construction of such a new fiber bundle super-operator. Instead, we provide some additional remarks to illustrate the broad outline of the key idea behind our proposal. A more detailed investigation of the intermaterial interface homomorphism will be given somewhere else.

In continuum field theories, the formal expression of the traditional boundary condition applied to the two materials' $\mathcal{D}_n$ and $\mathcal{D}_m$ base spaces (spacetime, space–frequency, or space differential manifolds) will be summarized by the symbolic formula

$$\mathcal{D}_m \rightleftarrows \mathcal{D}_n \tag{122}$$

in order to highlight that in such traditional formulation, it is the direct geometric relations between the individual material manifolds that usually holds the center stage. For example,

the electromagnetism of continuous media, is usually spelled out in the more specific space-limit form:

$$\lim_{\mathbf{r} \to \partial \mathcal{D}_{mn}} \{\mathbf{F}_m(\mathbf{r}) - \mathbf{F}_n(\mathbf{r})\} = \Gamma_{b_1}[\mathbf{F}_m(\mathbf{r}), \mathbf{F}_n(\mathbf{r})],$$

$$\lim_{\mathbf{r} \to \partial \mathcal{D}_{mn}} \{\mathbf{R}_m(\mathbf{r}) - \mathbf{R}_n(\mathbf{r})\} = \Gamma_{b_2}[\mathbf{R}_m(\mathbf{r}), \mathbf{R}_n(\mathbf{r})],$$

(123)

where $\partial \mathcal{D}_{mn}$ is the boundary between $\mathcal{D}_m$ and $\mathcal{D}_n$. Here, $\Gamma_{b_1}$ and $\Gamma_{b_2}$ are "base space boundary functions", which are not universal, but whose detailed expressions depends on the concrete content of the field theory and the material system under consideration.

On the other hand, in the superspace formalism of nonlocal metamaterials and continua, it can be seen that the various elements belonging to each fiber space $X_{n/m}$ attached at the point $\mathbf{r} \in \mathcal{D}_{n/m}$ of the base manifolds, i.e., the excitation field functions operating on the microdomains $V_{\mathbf{r}}$, $\mathbf{r} \in \mathcal{D}_{n/m}$, are to be mapped onto each other via an expression of the form:

$$X_m \rightleftarrows X_n : \quad \lim_{\mathbf{r} \to \partial \mathcal{D}_{mn}} (X_m - X_n) = \Gamma_f[X_m, X_n].$$

(124)

Here, $\Gamma_f$ is a new "fiber superspace boundary function". The full formulation of (124) is considerably more complex than the local field-theoretic case of (122) and (123) due to the fact that, additionally, the boundary condition quantity $\Gamma_f$ must be also proven compatible with the detailed corresponding fiber bundle structures of the materials involved. Consequently, for the field theoretic treatment of complex nonlocal continuum systems, the *global* topology of the metamaterial superspaces $\mathcal{M}_m$ and $\mathcal{M}_n$ will have to be assessed and utilized in the process of formulating a generalized "superspace boundary condition" of the form (124).

We summarize our main provisional view on the status of boundary conditions in the nonlocal field theory of inhomogeneous continua as follows:

- The existence of *extra* or *additional* structures in the fiber bundle superspace approach to nonlocality in complex continua forces on us the need for introducing additional boundary conditions or information coming from the microscopic *topological* structure of the corresponding material superspaces.
- The fiber bundle superspace formalism of nonlocal metamaterials appears to be able to capture the intricate processes taking place inside and across various nonlocal material domains joined together through interfaces.
- This is achieved by providing an efficient apparatus to topologically encode some of rich and complex physics of field–matter interactions via the construction of appropriate infinite-dimensional function spaces (Banach space fibers) attached at each point of the materials' base manifold.
- It is suggested that the relations between those additional *fiber spaces* are in fact what should be mainly taken into account while formulating boundary conditions for nonlocal continuum field theories, hence not merely the conventional relations involving only spatial interfaces between the material base manifolds as has been usually the practice in local field theories.

However, despite the fact that the full mathematical formulation of the proposed fiber bundle boundary condition homomorphism (124) is beyond the scope of this paper, it is hopped that the initial insight provided in this section can at least clarify the subject and stimulate further researches into the fundamental theory of nonlocal continua and metamaterials. Additional possible applications are given in the Appendixes A.3 and A.11.

## 9. Conclusions

We provided a general theoretical and conceptual investigation of nonlocal continuum field theories that aimed to achieve several goals. First, the subject was revisited from a new perspective, with the intention of introducing it not only to mathematical and theoretical physicists, but also invite a wider audience, including engineers, material scientists, chemists, applied physicists, and applied mathematicians. The various essential

ideas behind nonlocality in material continua were put under new light with the help of an abstract field-response model developed in three dimensions. Next, the fine-grained topological microstructure of nonlocal metamaterials was explicated in detail. We introduced the concept of physics-based nonlocality microdomains, demonstrating how the latter regions present an important structural topological feature of the physics of nonlocal media. Afterwards, it was proved using differential topology that a natural fiber bundle structure, serving as a "source (excitation) superspace", can be constructed. The material source fiber bundle superspace, or the material superspace in short, was shown to possess all of the required properties of a standard fiber bundle yet while faithfully reflecting the physics of nonlocal microdomains. Eventually, and using the technique of partition of unity, it was proved that the fiber bundle superspace can be deployed for the purpose of constructing and computing the complete nonlocal material response function over arbitrary microdomain configurations. This was accomplished by building a bundle homomorphism to replace the well-known, but now inadequate, material tensor linear operators commonly utilized in local continuum field theories, for instance, conventional electromagnetism. This new homomorphism can be viewed as a generalization of the linear operators of the various classical boundary-value problems of mathematical physics. It is hoped that in the future this homomorphism may be "topologically discretized" using suitable methods borrowed from other advanced fields such as algebraic topology, computational topology, and global analysis. The new fiber bundle superspace formulation suggested that nonlocal continuum field theories could be reformulated in an alternative way compared with the prevailing existing methods. Most importantly, nonlocality in material continua forces us to introduce an entire array of infinite-dimensional Banach spaces attached to every point in the conventional three-dimensional base space inside which the material is conventionally defined. This extra or additional fiber structure provides a natural explanation of why traditional boundary conditions often fail to account for the physics of nonlocal metamaterials. Moreover, the fiber bundle theory opens the door for several new applications, including the ability to understand the deep connection between topology and field theories, e.g., electromagnetism, in engineered artificial media. Overall, the author proposes that future research in metamaterials will gradually require more extensive collaboration between engineers and mathematicians in order to explore in full the deep consequences of this organic topology/electromagnetism relation.

**Funding:** This research received no external funding.

**Conflicts of Interest:** The author declares no conflict of interest.

## Appendix A

*Appendix A.1. Survey of the Literature on Nonlocal Metamaterials*

Appendix A.1.1. Introduction

We first provide a non-exhaustive and selective review of the development of nonlocal electromagnetic materials research. More information and proposals regarding engineering applications are given in Appendix A.3, where additional references can be found. The main propose behind this literature overview is to suggest that the area of nonlocal metamaterials research might be approached as an approximately coherent field of investigation, i.e., more than just being merely a technical sub-discipline selected from within the sciences of metals, semiconductors, plasma, periodic structures, metasurfaces, etc. In fact, one of the main objectives of this paper is to demonstrate that a unified theoretical treatment of the entire subject is mathematically possible (the superspace formalism.) However, one needs to be convinced first of the presence of substantial past researches into this area. Hence, convincing readers not familiar with the topic about the long and very rich history of investigations into various nonlocal phenomena in material systems is one of the objectives of this Appendix.

### Appendix A.1.2. Historically Important Examples

Some of the physical phenomena that cannot be understood using local electromagnetic theory include spatial dispersion effects [83], extreme negative group velocity and negative refraction [52,94], new diffraction behavior in optical beams [95], superconductivity [96], natural optical activity [16,97,98], non-Planck equilibrium radiation formulas in nonlocal plasma [99]. Outside electromagnetism but within wave phenomena, there also exists processes that cannot be fully accounted for through simple local material models, for instance, we mention phase transitions, Casimir force effects [100], and streaming birefringence [9]. By large, *spatial dispersion* has attracted most of the attention of the various research communities working on nonlocal electromagnetic materials. Indeed, few book-length researches on spatial dispersion already exist in literature, most notably [14–16,83]. We provide additional remarks on the history of spatial dispersion in Appendix A.2.

### Appendix A.1.3. General Theories of Nonlocal Continua

The majority of the published research on nonlocal media and nonlocal electromagnetism tend to focus on applications and specialized materials (see the majority of the references quoted below). Few exceptions include investigations attempting to approach the subject at a more general level. For example, from the perspective of general thermodynamics, see [8,9]. A unified perspective inspired by condensed-matter physics, especially plasma physics, can be found in [47]. Within nanoscale electrodynamics, nonlocality was treated broadly as an essential feature of microscopic interactions at the nano- and mesoscopic scales [50,53]. Some of the topics reexamined within the framework of a general nonlocal field–matter interaction theory include the applicability of optical reciprocity theorems [101–104], energy/power balance [105], quantization [106–108], operator methods [87], extension of spatial dispersion to include inhomogeneous media [76], and alternative formulations of spatial dispersion in terms of the Jones calculus [109].

### Appendix A.1.4. Semiconductors, Metals, Plasma, Periodic Structures

The bulk of the available literature on nonlocality is concentrated in the very large area of general field–matter interactions. There already exists a well-attested body of research on nonlocality in metals based on various phenomenological approaches, e.g., see [110] for a general review. Nonlocality has also been extensively investigated in dielectric media, for example semiconductors [83,111]. A comprehensive recent review of nonlocality in crystal structures is provided in [112], which updates the classic books [16,55]. Moreover, numerous researches conducted within condensed-matter physics and material science implicitly or explicitly assume that nonlocality is essentially based on microscopic (hence quantum) processes, and develop an extensive body of work where the spatially dispersive dielectric tensor is deployed as the representative constitutive material relation [37,38,50,53,54]. On the other hand, one can also treat nonlocality without resort to spatial dispersion by modeling certain classes of material media as periodic structures [113], e.g., photonic nanocrystals [114], where the susceptibility tensor is derived from the symmetry of the overall structure [37,89,115] or from the lattice dynamics approach [92,116].

### Appendix A.1.5. Boundary Conditions in Nonlocal Metamaterials

For solving nonlocal problems, several methods have been proposed in order to deal with the notorious problem of the lack of exact universal nonlocal response models at the intermaterial interface between a nonlocal domain and other media. The so-called additional boundary condition (ABC) approach adjoins new boundary conditions to the standard Maxwell's equations in order to account for "additional waves" excited at the interface, which otherwise would not be explicable by the standard local theory alone [16]. However, it must be noted that without exception all ABC formulations are inherently model-specific since each boundary condition model presupposes a particular type of nonlocal media, or simply just postulates specific ABCs based on their ease of use in applications, e.g.,

see [55,56,80,117–119]. We note that such ABC formalisms are not inevitable since there exists several boundary-condition free formulations, e.g., see [50,53,89].[29]

Appendix A.1.6. Computational Techniques

For performing full-wave field analysis in the presence of nonlocal materials, a number of discretization strategies have been proposed. For example, an FDTD-based method was suggested to deal with metallic spatially dispersive objects [120]. The formulation, discretization, and solution of surface integral equations for nonlocal plasmonic materials were also attempted in [121,122], where the reduction of the electromagnetic problem to a finite-matrix form was achieved using the RWG basis functions. Moreover, specialized methods were proposed for various possible scenarios involving nonlocal field–matter interactions, such as nonlocal dielectric profile retrieval from measurable data [123], iterative solutions of nonlocal wave equations [124,125], applications of the derivative expansion method to nonlocal plasma analysis [126], application of Kramers–Kronig relation method [127], application of the Pade approximation to homogenization [128].

Appendix A.1.7. Novel Systems and Devices with New Electromagnetic Behavior

The idea of exploiting nonlocality to design and develop a new generation of meta-materials (MTMs) exhibiting novel EM behavior has also received a revival in recent years [32,47,129], though the basic concept in itself is not completely new, going back to at least the 1980s and possibly earlier [16]. Recent examples of research focused in explicating nonlocal behavior to harness the associated new physics include spatial dispersion in photonic crystals [130], wire media [131–134], semiconductor nanoparticles [135–138], optically nonlinear liquids [139], hyperbolic metamaterials [140], layered dielectric-metal structures [141,142] and thin films [143], plasma-based metamaterials [144–146], quantum wells [147], soliton interactions with matter [148–153], superconducting films [154] and circuits [155], plasmonic devices and structures [156–158], nanocubes [159], cloaking [160], Chern metamaterials [161] and superconductors [162], dispersion management profiles [52,163], biomedical applications in materials [164], nonlocal antennas [165,166], and nonlocal uniaxial metamaterials [167]. Due to the practical importance of this area of research, we provide additional information in Appendixes A.3 and A.11.

Appendix A.1.8. Homogenization

Numerous homogenization theories for nonlocal MTMs, where averaging operations are considered over multiple spatial scales, have been reported in the literature, e.g., see [133,168–171]. We note that the subject of estimating the effective electric and magnetic properties of electromagnetic metamaterials, with or without nonlocality, is enormous and it is beyond the scope of this paper to even summarize the main papers in the field. Nevertheless, it is curious to note that until fairly recently, most publications have tended to focus on *non*-spatially-dispersive media; hence, local scenarios are still dominant in the area of advanced artificial material systems. This situation has began to change in the last few years, and nowadays an increasing number of reports appear to move from the old opinion that "spatial dispersion is a bug" to the more positive and fruitful perspective that nonlocality may provide pathways to novel physical behavior that can be exploited for various applications in metamaterial system design. However, we also note that progress in this second direction, where nonlocality is embraced rather than being treated with suspicion, has been generally slow.

Appendix A.1.9. Topological Materials and Photonics

A particularly interesting direction of research in nonlocal media is the recent subject of *topological photonics*. The main idea was inspired by previous researches in Chern insulators and topological insulators [10], where the focus has been on electronic systems. There, it has already been observed that the nonlocal behavior of the fermion wave function may exhibit a rather interesting and nontrivial dependence on the entire configuration space of

the system, in that case the momentum space (the wave vector **k** space). In addition to the already established role played by nonlocality in superconductors, quantum Hall effects are among the most intriguing physically observable phenomena that turned out to depend fundamentally on purely topological aspects of the electron wave function [96]. The major themes exhibited by electrons undergoing topological transition states include topological robustness of the excited edge (surface) states moving along a two-dimensional interface under the influence of an external magnetic fields. More recently, it was proposed that the same phenomenon may apply to photons (electromagnetism) [172], where the key idea is to use photonic crystals to emulate the periodic potential function experienced by electrons in fermion systems. However, since photons are bosons, transplanting the main theme of topological insulators into photonics is not trivial and is currently generating a great attention, see for example the extensive review article [11], which provides a literature survey of the field. One of the most important applications of topological photonics is the presence of "edge states", which are topologically robust unidirectional surface waves excited on the interface between two metamaterials with topologically distinct invariants. Since edge states are immune to perturbations on the surface, they have been advocated for major new applications where topology and physics become deeply intertwined [173]. Topology can also be exploited to devise non-resonant metamaterials [174] and to investigate bifurcation transitions in media [175]. Another different but related exciting subject illustrating the synergy between topology, physics, and engineering is non-Hermitian dynamics, especially in light of recent work related to the origin of surface waves [176,177], which is now being considered as essentially non-trivial topological effect. In Appendix A.11, the subject of topological photonics is taken up again but from the viewpoint of applications.

### *Appendix A.2. On the History of Spatial Dispersion in Crystal and Plasma Physics*

Historically, spatial dispersion had been under the radar since the 1950s, especially in connection with researches on the optical spectra of material domains [77,78,178]. However, the first systematic and thorough treatment of the subject appeared in 1960s, prominently in the first edition of Ginzburg's book on plasma physics, which was dedicated to electromagnetic wave propagation in plasma media. The second edition of the book, published in 1970, contained a considerably extended treatment of the various mathematical and physical aspects of the electromagnetism of spatially dispersive media [14]. Spatial dispersion in crystals had been also investigated by Ginzburg and his coworkers during roughly the same time [179–181]. The book [83] contains good summaries on spatial dispersion research up to the end of the 1980s. More recently, media obtained by homogenizing arrays of wires, already very popular because of their connection with traditional (temporal) metamaterials, are known to exhibit spatial dispersion effects, though many researchers ignore that effect to focus on temporal dispersion [182–184]. Other types of periodic or large finite arrays composed of unit cells like spheres and desks also exhibit spatial dispersion effects [185]. Nonlinear materials with observable nonlocality have also been investigated in the optical regime [186]. More recently, much of the resurgence of interest in spatial dispersion can be traced back to the observation that nonlocal phenomena cannot be ignored at the nanoscale level [187], especially in problems of low-dimensional structures, such as carbon nanotubes [89,91,92,188] and graphene [189,190]. The subject was also introduced at a pedagogical level for applications involving current flow in spatially dispersive conductive materials, such as plasma and nanowires [191].

### *Appendix A.3. Some Further Engineering Applications of Nonlocal Metamaterials*

The purpose of this Appendix is to provide a sample of some other current and future possible applications of nonlocal metamaterials based on the author's own experience, which may serve as a supplementary text to be read in conjunction with the general survey of Appendix A.1.

Appendix A.3.1. Communications Systems and Information Transmission

Nonlocal metamaterials offer a very wide range of potential applications in wireless communications and optical fibers. The basic idea is to introduce specially engineered nonlocal domains either as part of the communication channel (e.g., optical fibers, plasmonic circuits, microwave transmission lines) [189], or as a control structure integrated with existing antennas [129,192]. Spatial dispersion was also used as a method to engineer wave propagation characteristics in material domains, e.g., see [193] for applications to high-efficiency modulation of free-space EM waves. A general linear partial equation explicating how spatial and temporal dispersion can be jointly exploited to produce zero distortion (e.g., constant negative group velocity) was derived and solved in [52]. The main idea originated from the fact that one of the main sources of distortion in communication systems is that due to *non*-constant group velocity $\mathbf{v}_g := \nabla_{\mathbf{k}}\omega$ [194,195]. Since $\mathbf{v}_g$ is a strong function of the dependence of the material response tensor $\overline{\mathbf{K}}(\mathbf{k}, \omega)$ on both $\mathbf{k}$ and $\omega$, *dispersion management equations* can be derived for several applications. For example, it was proved in [52] that in simple isotropic spatially dispersive media with high-symmetry, one may obtain exact solutions where the group velocity is constant at an entire frequency band. This happens because while strong temporal dispersion is present (which alone causes strong distortion), incorporating optimized spatially dispersive profiles leads to *complete compensation* (cancellation) of distortion, resulting in essentially a distortion-free communication channel. There are enormous potentials of research into this new exciting area. The reason is that most practical realizations of nonlocal metamaterials involve complex material response tensors, where the relevant mathematics of dispersion engineering is still underdeveloped (and in fact underappreciated by researchers), which implies that, to the best of our knowledge, relatively very little has been done in this emerging field so far.

Appendix A.3.2. Electromagnetic Metamaterials

While this paper attempts to analyze and understand the general structure of nonlocality in generic field theories of continuous media, we have already mentioned above that artificial media, better known nowadays as metamaterials systems, could provide one of the most direct paths toward building new functional advanced materials and also providing models to further explore nonlocality both experimentally and numerically. As early as the 1960s, it was proposed that EM nonlocality can be exploited to produce materials with very unusual properties. For example, in [16], negative refraction materials were noted as one possible application of spatial dispersion where the path toward attaining this goes through controlling the direction of the group velocity vector. Since in nonlocal media, power does not flow along the Poynting vector [14], new (higher-order) effects were shown to be capable of generating arbitrary group velocity profiles by carefully controlling the spatial and temporal dispersion profiles. Overall, the ability of spatial dispersion to induce higher-order corrections to power flow is a unique advantage enjoyed by nonlocal metamaterials exhibiting weak or strong spatial dispersion in addition to normal dispersion. This extra spatial degrees of freedom provided by nonlocality was researched, reviewed and highlighted in many publications, including, for example, works such as [32,47,115,129,134,141,163,175,184,196].

Appendix A.3.3. Near-Field Engineering, Nonlocal Antennas, and Energy Applications

Another interesting application of nonlocality in electromagnetic media is near-field engineering, a subject that has not yet received the attention it deserves. It was observed in [129] that a source radiating in homogeneous, unbounded isotropic spatial dispersive medium may exhibit several unusual and interesting phenomena due to the emergence of extra poles in the radiation Green's function of such domains. Both longitudinal and transverse waves are possible (dispersion relations), and the dispersion engineering equations relevant to finding suitable modes capable of engineering desired radiation field patterns are relatively easy to set and solve. For example, by carefully controlling the modes of the radiated waves, it is possible to shape the near field profile, including total confinement of

the field around the antenna even when losses is very small, opening the door for applications like energy harvesting, storage, and retrieval in such media [197]. The direct use of especially-engineered nonlocal metamaterials, however, has been explored only for simple materials so far and mainly at the theoretical level [32]. However, the increasing importance of energy localization [198] at both the level of numerical methods [199] and the device level applications [200], suggest the need to reconsider the role played by nonlocality in complex media.

On the other hand, away from the source region, the subject of *far*-field radiation by sources embedded into nonlocal media was investigated previously by some authors within the context of plasma domains [104]. Recently, it has been systematized into a general theory for nonlocal antennas with media possessing an arbitrary spatial dispersion profile [25,165,166,192]. However, no general theory exists for nonlocal media, which are inhomogeneous. The superspace formalism proposed in this paper may help stimulate research into this direction in order to overcome the limitations of the existing theory of nonlocal antenna systems.

*Appendix A.4. On the Concept of Superspace*

The concept of superspace is not new, and has been proposed several times in both physics and mathematics. For a brief but general view on the definition of superspaces, see [201]. For example applications, various superspaces have been proposed as fundamental structures in quantum gravity [202,203], which are frequently infinite-dimensional. Superspace concepts are also now extensively researched in quantum field theory and the standard model of particle physics, e.g., see [204–206]. In general, dealing with topics such as supergravity, supersymmetry, superfields, superstrings, and noncommutative geometry often requires the use of one superspace formalism or another [204]. In mathematics and mathematical physics, where the concept itself originated, a notable recent example of the superspace concept includes *sheaves*, which are used in differential and algebraic topology and algebraic geometry and have numerous applications in physics [30,207,208].

In this paper, the superspace concept has very little to do with applications to supersymmetry or supergravity, such as the examples mentioned above (and many others we do not mention.) Instead, our use of the concept is more aligned with the mathematical practice of *extending* one space by *embedding* it into a larger superspace as in the schema:

$$\text{Space} \xrightarrow[\text{injection}]{\text{embedding}} \text{Superspace}. \tag{A1}$$

In other words, the embedded space is injected as a substructure into the (larger) embedding superspace. The key interest behind the formula superspace-as-embedding (A1), of course, goes beyond mere set-theoretic inclusion. We are not here trying simply to say that Space $\subset$ Superspace, which would be devoid of mathematical substance. Instead, the main motivation behind the superspace construction (A1) is that the embedded Space becomes a *substructure* attached to or placed within the larger, embedding "container", which is here superspace.

The most important thing to note here is that the latter superspace acquires a more coherent and fundamental status than the former. Eventually, Space becomes nothing but a mere "substructure" or "index space" of the more originary mother space that we originally called superspace. Strangely, with time, superspaces tend to become so familiar and basic to the degree one begins to call them regular spaces, while the original Space fades into oblivion. This last observation regarding the ontological primacy of the superspace concept over space can be best seen from the *converse* generative schema:

$$\text{Superspace} \xrightarrow[\text{projection}]{\text{de-embedding}} \text{Space}. \tag{A2}$$

Here, we recover the original space through a *projection* operation by which a de-embedding of the superspace substructure, the interior placeholder occupied by space, is achieved by projecting the mother space, the superstructure, superspace, onto the substructure, space. It is really the purely formal *structural* relation dictating how sub- and super-structures are organized within a common unifying global schema what is at stake in such type of superspace theories, i.e., not just the simple set-theoretic inclusion of one space into another.

Both operations, the injection (A1) and projection (A2) are necessary to fully understand the idea of superspace in general. However, in practice, usually only one of them is emphasized on the expense of the other. It is rare to find in superspace theories that both projection and injection operations are allotted the same ontological status. For instance, in the fiber bundle approach adopted in the present paper, space is recovered (or generated) from the fundamental superspace through the projection map of the mother fiber bundle, which will send each fiber into its "representative point" in the base manifold. In this manner, regular space may be seen as if it was actually *generated* or "produced" by the more primordial superspace mother structure [18,209].

A specific example more related to the subject of nonlocal MTMs is the original superspace concept introduced earlier for the analysis of deformed crystal [210] and subsequently utilized for fundamental investigations of EM nonlocality in incommensurate (IC) superstructures in insulators [76]. Such modulated-structure materials possess spaces with dimensions greater than spacetime [211]. Nevertheless, for fairly concrete models one may exploit group theory to construct finite-dimensional (dimension $> 4$) approximations of them. The general theory of superspace formalisms in quasi-periodic crystals is presented in [212]. Other examples from condensed-matter physics where superspace methods where applied include mesoscopic superconductivity [213].

*Appendix A.5. Guide to the Mathematical Background*

We provide a brief overview on how to read the mathematical portions of this paper and where to find detailed references that might be needed in order to expand some of the technical proof sketches provided in the main text. We emphasize that in this paper only the *elementary* definitions of

1.   Differential manifolds,
2.   Banach and Sobolev spaces,
3.   Vector bundles, and
4.   Partition of unity

that are needed in order to understand the mathematical development. Here, we briefly go over the principal ideas behind each one of these four key mathematical topics listed above, providing also additional references for readers interested in learning more about the required background. The current Appendix is not intended as a complete review; some familiarity with all four elementary mathematical topics listed above is required for a complete understanding of the technical proofs and constructions found in Section 5.

Appendix A.5.1. Topology on Smooth Manifolds

A *differential manifold* is a collection of fundamental "topological atoms" each composed of an open set $U_i$ and a chart $\phi_i(x)$, which serves as a *coordinate system*, basically an invertible differentiable map to the Euclidean space $\mathbb{R}^n$. That is, locally, every manifold looks like a Euclidean space with dimension $n$. When the differentiable map is smooth, the differential manifold is called *smooth manifold*. The collection of open sets $U_i, i \in I$, where $I$ is an index set, covers this $n$-dimensional manifold. Since some of these open sets are allowed to overlap, the crucial idea underlying the concept of the differential manifold is that over the common intersection region $U_i \cap U_j$, there exists a smooth reversible coordinate transformation function mutually relating the two coordinates of the same abstract point when expressed in the two (generally different) languages belonging to the topological atoms $U_i$ and $U_j$. *Note that the key concept of topology is how to propagate*

*information from the local to the global levels.* In this sense, differential manifolds present elementary structure allowing us to rigorously conduct this process using the efficient apparatus of the differential calculus. Note that only the elementary definition of smooth manifolds is required in this paper, which can be found in virtually any book on differential or Riemannian geometry, e.g., see [26,30,57,62,65,70,204,214].

Appendix A.5.2. Banach and Hilbert Spaces

A *Banach space* is a vector space equipped with a norm satisfying the standard properties that a generic norm should have (namely, being positive, being zero only for the null vector, scale linearity, and the triangle inequality [215].) Most importantly, Banach spaces are also required to be *topologically complete* in the sense that every Cauchy sequence converges to an element in the space itself. In this way, no "holes" are left in the space thus defined, hence one may deploy a Banach space in order to do analysis on operators as in solving differential equations or the analysis of numerical methods. A *Hilbert space* is a Banach space equipped with an inner product. An important fact to remember about Banach and Hilbert spaces is that when they are employed to model *function* spaces (as in this paper), they most often lead to *intrinsically-infinite* dimensional vector spaces [30].

Appendix A.5.3. Banach and Hilbert Manifolds

A straightforward process of combining Banach or Hilbert spaces with differential manifolds leads to the concept of *Banach* or *Hilbert manifold*, which are prominent examples of *infinite-dimensional manifolds.* A Banach/Hilbert manifold is simply a differentiable/smooth manifold that is locally isomorphic to a Banach/Hilbert space instead of the regular $n$-dimensional Euclidean space $\mathbb{R}^3$ invoked in the basic definition of an $n$-dimensional manifold. The isomorphism itself can be either differentiable or smooth, where a suitable derivative operator, such as the Fréchet derivative, may be defined on Banach/Hilbert spaces, leading to the resulting Banach/Hilbert manifold itself being either a differentiable or smooth infinite-dimensional manifold. A Banach/Hilbert manifold is then an intrinsically infinite-dimensional manifold. An elegant formulation of the theory of Banach manifolds can be found in Lang's text [26]. Applications of this theory in the general fields of analysis and geometry can be found in textbooks on global analysis, e.g., see [70,216]. In general, much of the theory of $n$-dimensional manifolds carry over unchanged into the case of infinite-dimensional manifolds. However, there exists some subtle technical differences, which are carefully highlighted in [214].

Appendix A.5.4. Sobolev Spaces

The most economic approach to constructing *Sobolev spaces* is to define them as Hilbert spaces consisting of (Lebesgue) square integrable functions that posses "generalized derivative", a concept in itself technical but straightforward. For the basic definition of Sobolev spaces and their applications to partial differential equations in mathematical physics and finite-element method in engineering, we recommend [65]. The subject of Banach manifolds is less commonly treated in the literature on Sobolev spaces than finite-dimensional manifolds. For a very readable account on the functional analytic background to the use of Sobolev spaces, see [65], while [64] provides information on the applications of Sobolev spaces in the analysis of linear partial differential equations. The generalization of the theory of Sobolev spaces into the wider setting of functions defined on differential manifolds is tackled in [66] (with applications to nonlinear functional analysis).

Appendix A.5.5. Vector Bundles

The quite general structure known as *fiber bundles*, of which *vector bundles* are famous special cases, are now standard topics in both mathematics (topology, geometry, differential equations), theoretical physics (quantum field theory, cosmology, quantum gravity), and applied physics (condensed-matter physics, many-body problems). On the major importance of vector and fiber bundles within the overall area of modern fundamental

physics, see [30,204,208]. In quantum field theory, gauge field theories use vector bundles as essential ingredients in the standard model of particle physics [18,30]. The increasing importance of methods based on quantum field theory in applications to condensed-matter physics has contributed into making knowledge of fiber bundle techniques useful and more widespread in physical and engineering research than originally anticipated; e.g., see the area of the Berry phase and the associated gauge connection [11,96]. The key idea behind the vector bundle is to attach an entire vector space to every point on a base manifold. To be more specific, consider a differential manifold $\mathcal{D}$ serving as the base manifold. Each copy of the vector space that is attached to a point in this base space will be called the *fiber* at that point. The standard *tangent space* of a smooth manifold is the most obvious example of such vector bundles. However, more complicated structures than finite-dimensional tangent spaces can also be captured by a suitable vector bundle concept. In this paper, we have shown that physics-based nonlocality in material continua can be modeled, very naturally in the mathematical sense, by considering the Banach space of all excitation fields acting on the microdomains indexed by a point in the material configuration space (base space). The fiber bundle superspace formalism may then be seen as a highly efficient and economic apparatus available for encoding, storing, and processing a large amount of topological and geometrical data pertinent to the problems of nonlocality in physics and engineering since fiber bundles lend themselves easily to complex calculations. Readable technical descriptions of vector bundles can be found in [30,57,63,70].

Appendix A.5.6. Additional Remarks on the Use of Sobolev Spaces in the Fiber Bundle Superspace Formalism

In Section 4, we introduced Sobolev space over the open domain $D$ instead of simply operating with the more generic Banach space. The reason behind our decision to invoke the more specialized (and technical) structure of a Sobolev space was mainly to actually simplify the technical development and in anticipation of future work on the superspace formalism. Indeed, in this paper, the fiber bundle $\mathcal{M}$ is referred to just as Banach bundle, not Sobolev bundle for the reason that all our essential results and insights apply to the more general concept of Banach space, which contains Sobolev spaces as a special case. In fact, Sobolev spaces are easier to work with in problems involving integro-differential equations such as nonlocal continuum field theory. Nevertheless, we only used the elementary definition of Sobolev space itself in Section 5, not its advanced properties. In particular, none of the other technical properties of Sobolev spaces are needed in the paper. Nevertheless, since in the future the material bundle space $\mathcal{M}$ is expected to be employed in order to construct solutions of Maxwell's equations in new form (i.e., in superspace instead of conventional spacetime), Sobolev spaces are projected to play the most important role since they have proved very efficient in analysis and the theory of partial differential equations [64].

Appendix A.5.7. Partition of Unity Techniques

In analysis and differential topology, the title *the partition of unity lemma* refers to a somehow rather technical tool used by topologists and analysts in order to help propagate information from the local to the global setting. They were found to be quite handy and easy to apply. The main theorem (Lemma 1) permits us to move from one topological "atom" to another by "gluing" them together using smooth standard domain-division functions. The technique was stated and used only toward the end of Section 5 in order to justify expansions, such as (77) and can be skipped in first reading of the paper. Partition of unity is usually taught in all topology and some geometry textbooks, e.g., see [26,57,63,70].

*Appendix A.6. The General Electromagnetic Model of Nonlocal (Spatially-Dispersive) Isotropic Domains*

One of the simplest—yet still demanding and interesting—nonlocal media is the special case of isotropic, homogeneous, spatially-dispersive, but optically inactive domains [14]. In this case, very general principles force the generic expression of the material response tensor to acquire the concrete form [13,15,16]:

$$\overline{\mathbf{K}}(\mathbf{k}, \omega) = K^{\mathrm{T}}(k, \omega)(\overline{\mathbf{I}} - \hat{k}\hat{k}) + K^{\mathrm{L}}(k, \omega)\hat{k}\hat{k}, \tag{A3}$$

where

$$k := |\mathbf{k}|, \ \hat{k} := \mathbf{k}/k, \tag{A4}$$

and **k** is the wave vector (spatial-frequency) of the field. The first term in the RHS of (A3) represents the *transverse* part of the response function, while the second term is clearly the *longitudinal* component, with behavior captured by the generic functions $K^{\mathrm{T}}(k, \omega)$ and $K^{\mathrm{L}}(k, \omega)$, respectively.[30] The tensorial forms involving the dyads $\hat{k}\hat{k}$, however, are imposed by the formal requirement of the need to satisfy the Onsager symmetry relations in the absence of external magnetic fields [16]

Using a proper microscopic theory, ultimately quantum theory, it is possible in general to derive fundamental expressions for the transverse and longitudinal components of the response functions in (A3) [14–16,37,38,50]. These forms are often obtained in the following way:

1.  First, fundamental theory is deployed to derive analytical expressions for $K^{\mathrm{T}}(k, \omega; \mathbf{r}')$ and $K^{\mathrm{L}}(k, \omega; \mathbf{r}')$.
2.  Afterwards, depending on the concrete values of the various physical parameters that enter into these expressions, e.g., frequency, temperature, molecular charge/mass/spin, density, etc., the obtained analytical expressions are expanded in power series with the proper number of terms.
3.  The expression of the dielectric tensor function is then put in the form of either a polynomial or rational polynomial in **k**.

A concrete example is given in Section 7.3 to illustrate the use of such physics-based dielectric functions for the case of exciton–polariton-based semiconductor materials.

*Appendix A.7. Origin of Electromagnetic Nonlocality in Excitonic Semiconductors*

Appendix A.7.1. Review of the Semiconductor Physics of Excitons

Very early in the history of condensed-matter physics, excitons were introduced by Frenkel [84,85], and further developed by other researchers, such as Wannier [217]. In the late 1950s, excitonic phenomena were transplanted into a central stage in the framework of light-matter interaction through the concept of *exciton–polariton* [178], which will be defined below. Pekar [178], Ginzburg [77], and others [79,119,218,219] affirmed the nonlocal approach to exciton–polariton materials by explicitly highlighting the strong impact of spatial dispersion near excitonic resonances. The subject of excitons is vast and multidisciplinary. For extensive treatments covering various applications in physics, chemistry, and technology, see [16,55,86,220–222].

In order to understand the particular nonlocal model to be presented in Section 7.3, let us first briefly explain the relevant physics of exciton–polariton interactions and why they can lead to strong nonlocal response. In a direct-band gap semiconductor the minimum of the conduction band is aligned along the maximum of the valance band, allowing electronic transitions from lower (unexcited) to excited bands upon interaction with external EM fields. For insulating semiconductors of the II-VI and III-V groups, exciton transitions occur in the visible or near-ultraviolet range of the electromagnetic spectrum. By engineering the material/metamaterial parameters, these transition frequencies can be shifted.

It should be noted that in contrast to metals and plasma, no free charged carriers are assumed to exist in the material. An electron exiting the valance band after the absorption

of an external photon will leave behind a *hole*, which acts as an independent quasiparticle that can travel throughout the material in the form of a collective excitation [84,85,223,224]. The *exciton* is defined as a coupled pair composed of the two bound states of the electron and hole. Here, both electrons and holes must be understood as "dressed" particles (quasiparticles) with effective mass and charge different from those of the bare (noninteracting) particle [225]. We may apply the Bohr model to the exciton (electron-hole pair) with simple modifications that can be summarized by the following procedure:

1. The electron mass must be replaced by the reduced exciton mass

$$m_{\mathrm{r}} := \frac{m_{\mathrm{el}} m_{\mathrm{h}}}{m_{\mathrm{el}} + m_{\mathrm{h}}}, \tag{A5}$$

   where $m_{\mathrm{el}}$ and $m_{\mathrm{h}}$ are the electron and hole masses, respectively.
2. The numerical values of $m_{\mathrm{el}}$ and $m_{\mathrm{h}}$ are determined by the curvature of the conduction and valance bands, respectively, and, hence, they follow from accurate quantum mechanical calculations of the band structure, see for example [55,226,227].
3. Due to the screening of Coulomb attraction by the dielectric medium, the effective electron charge $e^- = -e$ should be replaced by $e^- / \sqrt{\varepsilon_0}$, where $\varepsilon_0$ is the static dielectric constant.

From this, it follows that the exciton binding energy $E_{\mathrm{b}}$ is given by

$$E_b = \frac{m_{\mathrm{r}} e^4}{2\hbar^2 \varepsilon_0^2}. \tag{A6}$$

Therefore, the total energy needed to create an exciton state is given by

$$\hbar \omega_{\mathrm{e}} = E_{\mathrm{g}} - E_{\mathrm{b}}, \tag{A7}$$

where $E_{\mathrm{g}}$ is the semiconductor band gap energy. In most applications, the binding energy $E_{\mathrm{b}}$ is in the order of meV, while $E_{\mathrm{g}}$ is usually few eV. That is, the energy needed to create an exciton is slightly less than the band gap energy and typically we have

$$E_{\mathrm{b}} \ll E_{\mathrm{g}}. \tag{A8}$$

However, it is recommended to include binding energy in some applications for accurate calculations to help explaining the fine structure of measured excitonic transitions.

Appendix A.7.2. A Simple Explanation of How Nonlocality Emerges in the Excitonic Semiconductor

They key to the origin of nonlocality is the scenario when the excitation photon has an energy $\hbar\omega$ that is *greater* than the minimum exciton energy (A7). In the case where

$$\hbar\omega > \hbar\omega_{\mathrm{e}}, \tag{A9}$$

the excess energy will be transformed into *kinetic energy*. Due to the conservation of momentum, the wave vector of the exciton is equal to the photon wave vector **k** and, hence, the exciton kinetic energy $E_{\mathrm{e}}^{\mathrm{kinetic}}$ is given by

$$E_{\mathrm{e}}^{\mathrm{kinetic}} = \frac{\hbar\mathbf{k} \cdot \hbar\mathbf{k}}{2m_{\mathrm{e}}}, \tag{A10}$$

where

$$m_{\mathrm{e}} := m_{\mathrm{el}} + m_{\mathrm{h}} \tag{A11}$$

is the translational mass of the exciton in the effective-mass approximation [217]. Consequently, the total exciton energy $E_e$ is given by [228]

$$E_e(\mathbf{k}) = \hbar\omega_e(\mathbf{k}) = \hbar\omega_e + \frac{\hbar^2 k^2}{2m_e}. \tag{A12}$$

Consequently, the exciton frequency $\omega_e(\mathbf{k})$ acquires a novel dependence on $\mathbf{k}$, which is mainly due to the kinetic energy term in expression (A12). It is *precisely* such dependence that eventually leads to the emergence of electromagnetic nonlocality in semiconductors around excitonic resonances when photons couple with excitons. In other words, away from the excitonic transition regime, the effective dielectric function of the semiconductor exhibits only the typical dependence on $\omega$ (normal or temporal dispersion.)

*Appendix A.8. An Alternative Intuitive Derivation of the Dielectric Model* (90) *and the Quantum Origin of Nonlocality in Excitonic Semiconductors*

The model (90) itself may be intuitively derived as follows. A generic oscillator model is the one having the following well-attested Lorentzian expression:

$$\frac{1}{\omega_e^2 - \omega^2 - i\Gamma\omega}. \tag{A13}$$

This Lorentzian form models a large number of physical processes in nature, from lattice vibrations to electronic transitions and numerous many others [13,15,37,229]. Substituting the wave vector-dependent $\omega_e$ expression (A12) into the above Lorentzian form (A13), the dielectric function formula (90) can be immediately obtained when we keep only quadratic terms of $\mathbf{k}$. For a more careful quantum mechanical derivation, see [16,54,86].

*Appendix A.9. Computation of the Inverse Fourier Transform* (114)

We start from the standard Fourier transform pair

$$\mathcal{F}_{\mathbf{k}}^{-1}\left\{\frac{\chi}{k^2 - \gamma^2(\omega)}\right\} = \frac{\chi}{4\pi}\frac{e^{-i\gamma(\omega)|\mathbf{r}-\mathbf{r}'|}}{|\mathbf{r}-\mathbf{r}'|}, \quad \text{Im}\{\gamma(\omega)\} < 0, \tag{A14}$$

where the spatial Fourier transform is defined by (17). The condition

$$\text{Im}\{\gamma(\omega)\} < 0 \tag{A15}$$

is due to the physical requirement that fields do not grow exponentially in passive domains [13]. We also have written $|\mathbf{r} - \mathbf{r}'|$ instead of $|\mathbf{r}|$ in anticipation of the fact that the inverse Fourier transform will produce a Green function.

Our main task now is to make a proper choice of the correct sign when performing the square root operation $\sqrt{\gamma^2}$. Let us write

$$\gamma^2 = \text{Re}\left\{\gamma^2\right\} + i\,\text{Im}\left\{\gamma^2\right\}. \tag{A16}$$

From (91), we have

$$\text{Re}\left\{\gamma^2\right\} = \frac{m_e^\star}{\hbar\omega_e}\left(\omega^2 - \omega_e^2\right), \quad \text{Im}\left\{\gamma^2\right\} = \frac{m_e^\star}{\hbar\omega_e}\omega\Gamma. \tag{A17}$$

On the other hand, $\gamma$ also can take the form

$$\gamma(\omega) = \gamma'(\omega) + i\gamma''(\omega), \tag{A18}$$

where both $\gamma'$ and $\gamma''$ are real. The goal now is to derive expressions for $\gamma'$ and $\gamma''$ in terms of $\text{Re}\{\gamma^2\}$ and $\text{Im}\{\gamma^2\}$ with the correct sign since the square root is a many-one function.

To accomplish this, we use the following elementary theorem: let $x, y, a, b \in \mathbb{R}$. Then the square root of $x + iy$ is given by

$$\sqrt{x + iy} = \pm(a + ib), \tag{A19}$$

where the following expressions hold

$$a = \sqrt{\frac{x + \sqrt{x^2 + y^2}}{2}}, \quad b = \frac{y}{|y|} \sqrt{\frac{-x + \sqrt{x^2 + y^2}}{2}}. \tag{A20}$$

Substituting (A17) into (A20), the following is obtained

$$a = \sqrt{\frac{m_e^\star}{2\hbar\omega_e}} \sqrt{(\omega^2 - \omega_e^2) + \sqrt{(\omega^2 - \omega_e^2)^2 + (\omega\Gamma)^2}}, \tag{A21}$$

$$b = \sqrt{\frac{m_e^\star}{2\hbar\omega_e}} \sqrt{-(\omega^2 - \omega_e^2) + \sqrt{(\omega^2 - \omega_e^2)^2 + (\omega\Gamma)^2}}. \tag{A22}$$

Here, we used the calculation

$$y/|y| = \text{Im}\{\gamma^2\}/|\text{Im}\{\gamma^2\}| = \omega\Gamma/|\omega\Gamma| = 1, \tag{A23}$$

which follows from the fact that $\omega, \Gamma > 0$.

It remains now to find the correct signs. From (A14), the condition

$$\gamma'' = \text{Im}\{\gamma(\omega)\} < 0 \tag{A24}$$

must be satisfied. Therefore, we choose the *negative* sign in (A19). The final expressions become $\gamma = -a, \gamma'' = -b$, and after inserting $\gamma'$ and $\gamma''$ into (A14), the required relation (114) is obtained.

*Appendix A.10. On Extending Definition 1 to Noncompact Regions*

The localization of the physics-based nonlocality microdomains estimated using the formula (121) is based on approximating the exact mathematical definition of the topological microdomain (Definition 1) by response kernel functions possessing spatial decaying exponential behavior as in (114). It might be advisable then to provide a modification of Definition 1 taking into account the noncompact setting, which is the scenario more often encountered in physical applications.

**Definition A1** (**Nonlocal Microdomains: The Noncompact Case**). Consider a material domain $D$ with the associated nonlocal response function $\overline{\mathbf{K}}(\mathbf{r}', \mathbf{r})$. Assume further that the following bound holds:

$$\|\overline{\mathbf{K}}(\mathbf{r}', \mathbf{r})\| \leq A \exp(-a|\mathbf{r} - \mathbf{r}'|), \tag{A25}$$

for all $\mathbf{r}' \in D$, where $a$ and $A$ are some positive numbers. (Here, $\|\cdot\|$ is as in Definition 1). The form (A25) is called the *exponentially-decaying nonlocal kernel response function*. We define the effective (physics-based) nonlocal microdomain $V_\mathbf{r} \subset D$, labeled by $\mathbf{r} \in D$, as the open ball

$$\{\mathbf{r}' \in D, \|\overline{\mathbf{K}}(\mathbf{r}', \mathbf{r})\| < A \exp(-1)\}, \tag{A26}$$

where $A$ is as defined in (A25). In other words, an effective onlocality microdomain of this type, such as the one in Definition 1, is still locally compact.

**Remark A1.** The physical meaning of the ball (A26) is that it effectively approximate the spatial region where most of the "energy" of the response is concentrated, hence providing a physical means to estimate practical microdomain physical problems, since the Coulomb

interaction and other types of molecular forces are long range forces and, hence, cannot be described by an exact compact support such as the one originally introduced in the Definition 1.

**Remark A2.** It is straightforward to modify Definition A1 to include other forms of decay functions weaker than the exponential form. For example, we may replace the exponential in (A25) by a decaying function $1/r^n$, where $n$ is a suitable positive integer. Obvious modifications of the ball (A26) can then be subsequently made to construct the nonlocality microdomain.

*Appendix A.11. Possible Applications of the Superspace Formalism to Fundamental Methods in Metamaterials Research*

Appendix A.11.1. Estimating Fundamental Limitations on Nonlocal Metamaterials

Fundamental continuum response maps, such as $\mathcal{L}$ (72), can be completely reformulated in a different setting, that of the space of *vector bundle sections* [57,62,70]. The latter topic, the theory of sections, is an extremely well-developed subject in mainstream differential topology. In fact, in some cases the electromagnetic response field function $\mathbf{R}(\mathbf{r})$ itself may be obtained by working directly with the source bundle superspace $\mathcal{M}$. For example, under some conditions, this can be achieved by replacing each fiber $X_i$ by $X_i \times \mathbb{C}^3$. In this way, the entire nonlocal response problem reduces to understanding how vector bundle sections interact with the topology of the underlying base manifold $\mathcal{D}$. There is a large literature in differential topology and geometry focused on this latter technical mathematical problem, especially how local information can be transported from one place to another in order to extend local structures into global ones [26,57,63].

The author believes that by starting from local data in a given nonlocal metamaterial domain, e.g., the global shape of the device, the distribution of topological holes, etc, one may then use existing techniques borrowed from differential topology, e.g., the theory of characteristic classes, to determine allowable EM response functions that are in principle permissible at the global level. Engineers are typically interested in acquiring in advance the knowledge of what the best (or worst) performance measures obtainable from specific topologies are. Hence, reformulating the electromagnetism of nonlocal metamaterials in terms of vector bundles could be of help in this respect since it opens a pathway, within metamaterials research, toward a synergy between general topology, physics, and engineering.

Appendix A.11.2. Numerical Methods

Traditional full-wave numerical methods are sometimes deployed in order to deal with nonlocal EM materials, often using the additional boundary conditions framework, in spite of the latter's lack of complete generality.[31] At the heart of the traditional approach to numerical methods in local electromagnetism is the concept of operators between linear spaces. However, by reformulating the source space of field–matter interaction in terms of a Banach bundle, it should be possible to reformulate Maxwell's equations to act on this extended geometric superspace instead of the conventional spacetime framework. As an alternative to the concept of the linear operator of classical mathematical physics and numerical methods, we now have the much more general and richer concept of bundle homomorphism developed in Section 5. Some of the advantages anticipated from such reformulation include

1.  The ability to resolve the issue of generalized boundary condition (already discussed in Section 8).
2.  Since every point belonging to a fiber superspace is in itself a smooth function defined on an entire material sub-microdomain, by building a new system of discretized recursive equations approximating the behavior of electromagnetic solutions living in the enlarged superspaces $\mathcal{M}$ and $\mathcal{R}$ one may anticipate arriving at a deeper understanding of the physics of nonlocality. The reason is that the topology of the nonlocal

interaction regime is explicitly encoded into the *geometry* of the new expanded solution superspace $\mathcal{M}$ itself. Characterizing this geometry is then possible through a suitable discrete approximation of the *interior* microtopological content of the superspace (fiber bundle) structure itself; i.e., not just at the "exterior" parts often found in the boundary conditions of classical local field continuum theories, but also "going inside" the problem space as such.

3.  It is also possible that such numerical methods may emerge as more computationally efficient and broader in applicability than the conventional methods rooted in local electromagnetism. One reason for this is that the Banach vector bundle formulation introduced in this paper is quite natural and appears to reflect the underlying physics of nonlocal metamaterials in a direct manner. From our general experience in numerical methods, "natural operations" tend to translate into numerical methods with better convergence, sensitivity, and robustness.

As directly related to the three possible advantages of the superspace formalism discussed above, we also add that in recent years the subject of *computational topology* has gained momentum, where some researchers are now building new numerical methods by exploiting the topological structure of the problems under considerations, e.g., see [72,230]. The fiber bundle superspace formalism of this paper might provide a way to link research done in electromagnetic and non-electromagnetic nonlocal materials with such advances in the computational and applied mathematical sciences.

Appendix A.11.3. Topological Photonics

One of the main applications of the proposed vector bundle formalism is that it opens the door for a new way to investigate the topological structure of materials. It has already been noted that the nonlocal EM response is essential in topological photonics, e.g., see [11,161] and also Appendix A.1. Indeed, since in topological photonics the wave function of bosons, usually the Bloch state, is examined over the entirety of momentum space (usually the Brillouin zone), then it is the dependence of the EM response on **k** what is at stake, which naturally brings in nonlocal issues. But now since by using our theory we can associate with every nonlocal material a concrete fiber bundle superspace reflecting the rich information about the multiscale topological microdomain structure and the global shape of the material plus the impact of the boundaries separating various material domains, it is natural to examine whether a topological classification of the corresponding fiber bundles may lead to a new way to characterize the topology of materials other than the Chern invariants used extensively in literature. The advantage of the superspace approach in this case is that the complicated topological and geometrical aspects of the boundaries and inhomogeneity in nonlocal media can be encoded very efficiently in the local structure of the material fiber bundle. Using standard techniques in differential topology [62], it should be possible to propagate this local information to the global domain (the entirety of the system), for example by computing suitable fiber bundle topological invariants like its homology groups [70]. Our approach is then a "dual" to the standard approach since we work on an enlarged configuration space (spacetime or space–frequency), while the mainstream approach operates in the momentum space of the wave function.

## Notes

1. The author would like to thank one of the anonymous reviewers for suggesting this connection.
2. This is argued in detail in [32]. In particular, the recently-introduced current Green's function of electromagnetic devices was inspired by finding a Green's function structure similar to that corresponding to nonlocal media [33–36].
3. See Appendixes A.1 and A.2 for the literature review.
4. For a brief discussion of some possible engineering applications of metamaterials, see Section A.3.
5. Cf. Section 3.3.

[6] If *D* asymptotically grows into an unbounded region, then the problem reduces to that of homogeneous unbounded domain (bulk media), well treated in the basic literature on spatial dispersion. Clearly, in this paper, we are not interested in such a topologically trivial problem.

[7] Cf. Remark 3.

[8] This is well known from the quantum theory of nonlocality [16,38,50], but concrete examples illustrating this behavior will be given in Section 7.

[9] This compactness of the response kernel's support cannot be proved in general, but is very plausible on physical grounds (causality considerations). Therefore, we posit such compactness as an axiomatic feature of all physically-realizable causal nonlocal continuum field theories. However, see Appendix A.10.

[10] For some possible definitions of matrix norms, see for example [60,61].

[11] In this section and the one to follow, we do not worry much about the details of the electromagnetic model and for simplicity assume that only one vector field **F** acts as excitation and one response field **R** is induced. More complex media like bi-anisotropic domains and others [67] may also be treated within this formulation. For example, if two response fields are needed, the codomain in (33) can be simply changed to $\mathbb{C}^6$.

[12] Cf. Remark 3.

[13] See the discussion of nonlocal and topological metamaterials applications in Appendix A.3.

[14] Cf. Appendix A.1.

[15] See [59,68] for the full technical definition of *subordinated cover*. A collection of subsets of a topological space is said to be *locally finite*, if each point in the space has a neighborhood that intersects only finitely many of the sets in the collection. What we need here is that there exists some $i$ and **r** such that $U_i$ is inside $V_\mathbf{r}$, i.e., $U_i \subseteq V_\mathbf{r}$ where $\mathbf{r} \in U_i$.

[16] The function space $C^p$ is comprised of the set of real functions that are continuously differentiable $p$-times [62,69].

[17] See Section 7 for one possible method and examples.

[18] Cf. Remark 3.

[19] The discretization of the nonlocal MTM bundle homomorphism itself is outside the scope of the present work and will be addressed elsewhere.

[20] On the technical difference between local and global topological isomorphisms, especially in differential topology, see [62,73,74].

[21] For instance, by introducing holes into a simply-connected domain in order to make the latter disconnected.

[22] The numerical value of $\alpha$ may be different for transverse and longitudinal excitation fields.

[23] In the effective-mass approximation, a simple way to estimate the exciton mass $m_e^\star$ is via the relation $m_e^\star = m_{el} + m_h$, i.e., the sum of the effective electron and hole masses introduced in Appendix A.7. However, it must be noted that this relation is far from being universal, e.g., it should be modified when there are strong interactions [81,82]

[24] In typical crystal materials, the ratio $\omega_{L,T}/\omega_T$ is already about $10^{-3}$ [16,55].

[25] It should be noted that there is no loss of generality here. The computational model to be presented shortly allows the estimation of the nonlocal microdomain topology based on a generic model of the form (90). If $\varepsilon^L$ and $\varepsilon^T$ are not identical, then the same procedure can be applied to each one of them separately.

[26] There is a nice parallelism here with *temporal* dispersion where the latter is known to arise from the inertial effects of electrons in interaction with radiation fields [37].

[27] This is more obvious in FEM or FDTD than MoM.

[28] Cf. Section 5.2.

[29] Cf. Section 8.

[30] Even for isotropic materials, the response tensor $\overline{\mathbf{K}}$ remains a tensor. This is due to the manner in which the equivalent dielectric function is defined using the Fourier transform instead of the conventional multipole approach, e.g., see [16,32,37].

[31] Cf. Section 8 and Appendix A.1.

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
