# Peer review of "On the Topological Structure of Nonlocal Continuum Field Theories"

_foundations, doi:10.3390/foundations2010003_

Round 1
Reviewer 1 Report
The paper "On the Topological Structure of Nonlocal Continuum Field Theories" presents a comprehensive review on the mathematical aspects underlining the approach for studying nonlocal effects.
A so-called "superspace" structure is proposed, and applications are then performed. The paper also brings a nice presentation, well written, and has the advantage of pedagogy.
Sometimes, as a matter of style, the main text appears to be a little repetitive. I do not see this as a problem, nevertheless. This is an important manuscript, whose main goal in my opinion is the effort to be accessible and self-contained, two rare aspects when mathematical physics is being developed/used.
I do recommend this paper for publication as it is.
Reviewer 2 Report
The work proposes a discussion about the topological aspects of theoretical physics. The work is organized in a didactic style, introducing the abstract concepts step-by-step.
The work seems to be interesting, but already in the introduction some statements are misleading or wrong, and since this work has a didactic objective, it must be corrected.
Some troubling aspects of the text are mentioned below.
1) It is mentioned that topological aspects were introduced in Physics by Weyl. But the Classical Mechanics, in the Hamilton and Hamilton-Jacobi formulations, already present many connections between geometry/topology and Physics. Many of the geometrical aspects of modern physics were already present in classical physics.
2) There is a strange discussion about the photon mass that seems to be misleading. There is no doubt that the photon mass is null, so why should one propose measurements to prove it?
3) The existence of multiple scales is not really new. The scaling properties are important in Yang-Mills fields, the non-Abelian field theory, and it has been recently used to propose the presence of fractal structures in the dynamical evolution of the fields.
4) The introduction of nonlocal metamaterials is confusing and must be clarified if the objectives of a didactic work are to be maintained.
5) There is a growing interest in fractional differentiation in Physics, and the author introduces topological differentiation. It would be interesting to check and comment if there is any relationship that can be established between the two differential operators.
Reviewer 3 Report
This paper develops a new mathematical formalism for describing nonlocal phenomena in the electrodynamics of metamaterials, in particular for calculating the response of a material (which can be thought of as consisting of locally homogeneous subdomains) to an excitation field. The author argues that such a formalism must be based on a topology in which the open sets are local microdomains, the sets of points with nonvanishing response to an excitation at some initial point. This formalism is developed in some detail. An example is given to illustrate the usefulness of the formalism.
I would say that these are interesting ideas which, if they could be implemented in practice, might help with practical problems in the calculation of electromagnetic fields in such materials. I think however the author should try harder in making these ideas accessible and interesting to a wide group of readers, which is clearly the goal of the paper.
I think the manuscript is far too long for what it does. Already the introduction takes over 4 pages of two-column text with largely general comments and considerations. It might help to get to the point much quicker. There is also some repetition and the text sometimes reads more like a commentary of what the authors thinks would be interesting about their work, rather than showing some concrete results. Maybe these commentary aspects could be moved to later discussions. Similarly, the concrete example is described over several pages from p.21-25 with intricate detail on how to interpret a particular semiconductor model, which is not really needed for this paper (which was supposed to introduce a new mathematical formalism).
The last point brings me to my main concern which is that I think that the example doesn't really help to support the idea that the mathematical formalism introduced here is useful. The reason is that the example does not seem to fit into the requirements of the formalism: the response function derived in (116) is clearly nonvanishing over all of R^3 and so does not have the "local" property that there is a nonzero response in only a finite ball around a certain point. This is already evident from the fact that it is a continuous function of k when written in Fourier space. The proposal in (122) appears arbitrary and contrary to the requirements defined earlier (e.g. in Definition IV.1). At the radius a_r defined there the response is not in fact zero, but has merely fallen off by a factor 1/e compared to its maximum.
The fact that the example given by the author does not in fact fit into the formalism defined earlier is worrying, since it suggests there are no actual useful applications for the formalism, hence undermining the central message. The author must at least find a better example, and present it in much less space (2-3 pages maximum should be enough).
I have some minor suggestions for improvement too.
In (16)-(17) the author discusses homogeneous materials for which the function K can be defined in Fourier space. On the next page, they suggest that such a form can be generalised "by juxtaposing several subdomains where each subunit is homogeneous", while suggesting that one can still define K(k) in Fourier space for each subdomain. I do not understand mathematically how this is supposed to work, since the Fourier transform is defined globally on R^3. Can the author elaborate on this point? How would I not get a globally homogeneous function in r space from a definition K(k)? Of course this point relates to the issues with the author's example, explained above. In the same paragraph, the author talks about "concatenating multiple regions" again without really explaining what this means.
The expression "phi_i^(-1)(B_(a/3))" in (44) should be defined, since it is not clear what B_(a/3) is (presumably a is indirectly defined through the comment below (39)?)
Figure 8(a) seems to suggest that the function 1/|r-r'| becomes negative, which is odd. Maybe this includes a plane wave modulation, but this should be explained in the caption.
Round 2
Reviewer 2 Report
The revised version is improved with respect to the previous one. The authors included some comments on future perspectives and possible connections with recent advances in the theory. However, they should include citations referring to those advances that are in connection with their work, so readers can appreciate the reach of the theory in development by both the authors of the present work and those that previously contributed to the field. After that, the manuscript can be accepted for publication.
Author Response
Thanks to the Reviewer for his remarks. In the Revised MS, please note the following references to current and future works connected with the proposed theory:
- For topological photonics and topological insulators, please see Refs [10,11]
- Superconductors nonlocality, see Ref. [9]
- Nonlocality in nanostructures, see Ref [49-50]
- For recent advances in nonlocal metamaterials, see Refs [47-48]
- Nonlocality in plasma, see Refs [41]
- Topological methods in condensed matter physics such as Berry phase, etc, see Ref [95]
- Nonlocality in metals, see recent advances in Ref [109]
- Nonlocality in semiconductors, Ref [110,111]
- Energy localization in electromagnetic devices, Ref [195-200]
- Nonperiodic crystals, see Ref [212]
- Nonlocal antennas, see Refs [192]
- Cold plasma in fusion reactions Ref [87]
Thank you again for your time and I am pleased by the positive evaluation by the Reviewer.
Reviewer 3 Report
I appreciate the author's reply and general comments on the philosophy behind their work. Indeed my background is more on the mathematical side. I would argue that if a relatively advanced mathematical formalism is introduced over many pages, an example supposed to support the formalism should at least satisfy the requirements of the formalism itself. I'm not sure I find the author's reply entirely convincing, and the new Remark VII.2, while acknowledging the issue, does not really try to show in detail how the general formalism and the fact that the example doesn't fit into it can be reconciled. Would it be too much to ask that the author would provide an alternative formulation of Definition IV.1, general enough to capture also "physically interesting" cases such as the example in Section VII.C? My feeling is that this might change very little, e.g. if the domain of nonlocality is not defined by the strict compact support of a response function but by the region in which the response function is greater than some epsilon fraction of its value when r = r' ? On the other hand, (34) and (87) would then also only be approximations. This could make the paper more convincing, at least to more "mathematically minded" people.
Author Response
Many thanks to the Reviewer again for the insightful remarks. I agree with his/her opinion. In fact, I am a mathematically-minded person myself. It is a pleasure to find company here since most of my other reviewers on these topics tend to focus on non-mathematical aspects.
Initially I did not want to further elaborate the above issue regarding Def. IV. 1 in noncompact regions because of the other remarks I received regarding the length of the MS so I only added Remark VII.2 in my previous response to the Reviewer. Now in response to this Reviewer’s new request, I composed a new small Appendix J in the Revised MS (blue text). Again, since this new material will go to an Appendix, it will not hinder the flow of the main text. I then inserted a reference in Remark VII. 2 to this new Appendix J (blue text). There (in Appendix J), you will find a simple treatment of the case of the noncompact support domain of the Kernel Response where I truncate the unbounded region by a locally compact one using the physics-motivated decaying exponential e^-1 limit as a practical criterion to “compactify” in applications. Please note that I could have chosen e^-n, where n>1, with little impact on the practical situation. Now, using this effective “spatial cutoff” approximation I evade unnecessary (at least in this MS) mathematical limit-related complications when dealing with non-compact domains. Strictly speaking, nobody wants to build a new topological space where things are not locally compact, at least not in the first proposal. Here I insist on building my superspace using these locally compact topological micro-domains because they are more physically meaningful and much easier to work with. On the other hand, probably only in critical phenomena and phase transitions one would encounter strictly infinite nonlocal domains that are difficult to ignore. However, I am not sure phase transitions belong to the subject of my theory so there is no need to insist on including noncompact microdomains that cannot be “compactified”. In the end, how could a “MICRO-domain” be infinite? So clearly my microdomains should be either exactly locally compact or effectively so as I do in the new Appendix J.
I hope this appendix answers the Reviewer’s requests. Hopefully I can take up this issue in more detail in future publications directed to a more specialized audience.
Thank you again for your time and constructive remarks.